# On the breakdown of dimensional reduction and supersymmetry in random-field models

Gilles Tarjus[*] and Matthieu Tissier[†]

*LPTMC, CNRS-UMR 7600, Sorbonne Université, 4 Pl. Jussieu, 75252 Paris cedex 05, France*

Ivan Balog[‡]

*Institute of Physics, P.O.Box 304, Bijenička cesta 46, HR-10001 Zagreb, Croatia*
(Dated: November 27, 2024)

We discuss the breakdown of the Parisi-Sourlas supersymmetry (SUSY) and of the dimensional-reduction (DR) property in the random field Ising and $O(N)$ models as a function of space dimension $d$ and/or number of components $N$. The functional renormalization group (FRG) predicts that this takes place below a critical line $d_{\mathrm{DR}}(N)$. We revisit the perturbative FRG results for the RFO$(N)$M in $d = 4 + \epsilon$ and carry out a more comprehensive investigation of the nonperturbative FRG approximation for the RFIM. In light of this FRG description, we discuss the perturbative results in $\epsilon = 6 - d$ recently derived for the RFIM by Kaviraj, Rychkov, and Trevisani.[1,2] We stress in particular that the disappearance of the SUSY/DR fixed point below $d_{\mathrm{DR}}$ arises as a consequence of the nonlinearity of the FRG equations and cannot be found via the perturbative expansion in $\epsilon = 6-d$ (nor in $1/N$). We also provide an error bar on the value of the critical dimension $d_{\mathrm{DR}}$ for the RFIM, which we find around $5.11 \pm 0.09$, by studying several successive orders of the nonperturbative FRG approximation scheme.

PACS numbers: 11.10.Hi, 75.40.Cx

## I. INTRODUCTION

Although having been introduced some 50 years ago[3–6], the random-field Ising model (RFIM) and its extension, the random-field $O(N)$ model (RFO$(N)$M), keep stimulating an ongoing research activity and lively debates (for recent papers, see [1,2,7–14]). Besides their relevance as effective theories for a wide range of situations,[12] one of the recurring (and fascinating) questions about the theoretical description of random-field systems is the nature of the mechanism by which dimensional reduction and the underlying supersymmetry[15] are broken as one lowers the dimension below the upper critical dimension $d = 6$. Dimensional reduction (DR) is the property that the critical behavior of the RFIM with the disorder strength as control parameter is the same as that of the pure Ising model with temperature as control parameter in two dimensions less. It is found at all orders of perturbation theory.[4–6] The Parisi-Sourlas supersymmetry (SUSY) is an emergent symmetry associated with the zero-temperature properties of the model[15] and it entails DR, even at a nonperturbative level.[11,16–19] A similar behavior is found in the RFO$(N)$M.[20] As it was proven early on through heuristic[3] and rigorous[21–23] arguments that DR does not hold in dimensions $d = 2$ and $d = 3$, the search for the process leading to the breakdown of DR and SUSY has enjoyed a continuing interest over the last decades.

In a series of papers since 2004 we have proposed a consistent theoretical explanation of DR and SUSY breakdown as one lowers the dimension in the RFIM and RFO$(N)$M through a functional renormalization group (FRG) treatment: for a review, see [12]. The breakdown is associated with the appearance of a nonanalytic dependence on the order-parameter fields (a "cusp", to be described in detail later on) in the cumulants of the renormalized disorder and in the correlation functions at the zero-temperature fixed point that controls the critical behavior of the model.[19,24–27] This cusp in the functional form of the cumulants of the renormalized random field is the consequence of the presence of scale-free avalanches in the ground state of the model under infinitesimal changes of an applied source at criticality,[28] avalanches that are indeed observed in computer simulations of the RFIM at zero temperature.[29–31] The connection between cusp in the functional dependence of the cumulants, avalanches, and breakdown of DR has also been amply demonstrated in another disordered model which describes an elastic manifold pinned in a random medium.[32–38] The functional character of the RG is central in such problems.

There are two different patterns of SUSY and DR breaking in the RFO$(N)$M depending on the values of $N$ and $d$. Near the lower critical dimension of the RFO$(N > 2)$M, in $d = 4 + \epsilon$, the perturbative FRG to 2 loops predicts that the SUSY/DR fixed point which controls the critical point at large $N$ first becomes unstable in $N = 18.3923 \cdots$ when a "cuspy" fixed point becomes stable and then vanishes in $N = 18$ when it collapses with a SUSY/DR unstable fixed point.[27] ("Cuspy" refers to the fact that the cumulants of the renormalized random field have a nonanalytical functional dependence in theform of a cusp: note that the functional character of the RG is crucial to capture this effect, even if the FRG is perturbative in $\epsilon = d - 4$ here.) On the other hand, for the Ising version, $N = 1$, the nonperturbative FRG predicts that the SUSY/DR critical fixed point which is present at the upper critical dimension $d = 6$ disappears around $d_{\mathrm{DR}} \approx 5.1$, even before becoming unstable to a

cuspy perturbation associated with avalanches. There is thus a critical value along the line $d_{\mathrm{DR}}(N)$ at which the SUSY/DR fixed point disappears that separates the two different patterns and that we have estimated around $N_x \approx 14$ and $d_{\mathrm{DR}}(N_x) \approx 4.4$.[27]

The FRG prediction for the breakdown of SUSY and DR in $d_{\mathrm{DR}} \approx 5.1$ for the RFIM is supported by both nonperturbative[10,19,24–26] and $\epsilon = 6 - d$ perturbative calculations,[39] and the results are in good agreement with all simulation results in $d = 3$, 4, and 5.[7,40–45] In particular it describes very well the DR broken results in $d = 4$[41,43] and the weak or negligible breaking of SUSY and DR in $d = 5$.[7,44] Furthermore it has a clear physical interpretation in terms of scale-free avalanches: They are present even in $d \geq d_{\mathrm{DR}}$ but then only have a subdominant effect at the fixed point (the resulting amplitude of the cusp which comes from the second moment of the avalanches is an irrelevant perturbation for $d \geq d_{\mathrm{DR}}$), while they dominate the critical behavior when $d < d_{\mathrm{DR}}$.[28] The mechanism of the collapse of fixed points and the emergence of a new cuspy fixed point below $d_{\mathrm{DR}}$ is very unusual,[27] which explains the corrections to scaling observed in $d = 5$ in large-scale simulations.[7,10] Finally, the FRG prediction is also compatible with the loop expansion around the Bethe lattice[9] and approximate conformal-bootstrap results.[8]

In this paper we revisit this FRG description of SUSY and DR breakdown by first focusing on the recent work of Kaviraj, Rychkov and Trevisani, hereafter denoted KRT.[1,2] In the latter a perturbative (nonfunctional) RG investigation of the RFIM at 2 loops around the Gaussian fixed point in $d = 6 - \epsilon$ suggests that SUSY breaking operators destabilize the SUSY fixed point in $d \approx 4.2 - 4.6$. By building on our previous FRG analysis of both the RFO$(N)$M in $d = 4 + \epsilon$[25,46] and the RFIM in either $d = 6 - \epsilon$[25,39] or nonperturbatively in all dimensions,[10,18,19,25,26,28,47] we show that the scaling dimension of these dangerous operators have a counterpart in the eigenvalues that have already been computed within the FRG around the SUSY/DR fixed point. Crucially, the functional and nonperturbative character of our approach allows us to address questions that cannot be directly answered through the perturbative treatment of KRT. In particular, we show that when the eigenvalue associated with the most dangerous (polynomial) operator destabilizing the SUSY/DR fixed point vanishes at a critical dimension $d_{\mathrm{DR}}$ (the operator then becomes marginal), the SUSY/DR fixed point actually *disappears* instead of just becoming unstable as predicted in [1] and we discuss the mechanism by which this happens. As a result, even by fine-tuning additional control parameters such as the form of the bare random-field distribution there is no way to access any SUSY/DR critical point in dimensions below $d_{\mathrm{DR}} \approx 5.1$. In computer simulations of the RFIM in $d = 4$ or 5 one could therefore only at best observe remnants of SUSY/DR behavior over finite sizes, remnants that would disappear in the asymptotic critical regime if large enough system sizes are accessible.

Finally, we check the robustness of our theoretical prediction that the SUSY/DR fixed point disappears in $d_{\mathrm{DR}} \approx 5.1$ for the RFIM. For this we study different orders of the nonperturbative approximation scheme that we have previously introduced within the FRG formalism. By considering both cruder and improved levels of approximation compared to our previous work,[10,19,26,27] we find that $d_{\mathrm{DR}} \approx 5.11 \pm 0.09$, thereby providing an error bar for our result.

The paper is organized as follows. In Sec. II we summarize the recent work of KRT.[1,2] We next give the key results of our work and put the contribution of Refs. [1,2] in the framework of our FRG approach. Then, in Sec. III we make a detour via the RFO$(N > 2)$M by reanalyzing its perturbative but functional RG description at one- and two-loop orders near the lower critical dimension $d = 4$. Here, the study is performed as a function of the number of field components $N$ instead of the spatial dimension $d$. We present analytical and numerical results unambiguously showing that when the most dangerous operator for destabilizing the SUSY/DR fixed point which is found for large values of $N$ becomes marginal the SUSY/DR fixed point disappears at once by collapsing with another (completely unstable) SUSY/DR fixed point. We next come back to the case of the RFIM in Sec. IV. We discuss the nature and the scaling dimension of the operators that are potentially dangerous for destabilizing the SUSY/DR fixed point as the dimension $d$ is decreased. We do so in the various parametrizations of the replica fields. We also spell out within the nonperturbative FRG the mechanism by which the SUSY/DR fixed point disappears in the critical dimension $d_{\mathrm{DR}}$ where the most dangerous operator becomes marginal. In Sec. V we check the robustness of the nonperturbative FRG predictions by implementing several successive orders of the nonperturbative approximation scheme. The outcome is an apparent rapid convergence for the value of the critical dimension $d_{\mathrm{DR}}$. Finally, we present some concluding remarks in Sec. VI about the physical implications of our findings. In addition, we provide several appendices to discuss more technical aspects of our investigation and to further address some of the comments made by KRT[1,2] on our FRG approach.

## II. THE RECENT WORK OF KRT[1,2] IN LIGHT OF THE FRG APPROACH

### A. Summary of the work in [1,2]

KRT[1,2] start from the replica field-theoretical description of the RFIM in which one considers a bare action for

replica scalar fields $\phi_a$ with $a = 1, \cdots, n$ of the form[48]

$$S[\{\phi_a\}] = \sum_a \int_x \left\{ \frac{1}{2}[\partial\phi_a(x)]^2 + \frac{r}{2}\phi_a(x)^2 + \frac{u}{4!}\phi_a(x)^4 \right\} - \left(\frac{\Delta}{2}\right) \sum_{ab} \int_x \phi_a(x)\phi_b(x),$$

(1)

which is obtained after having introduced $n$ replicas of the system and averaged over a Gaussian (bare) random field of zero mean and variance $\Delta$. (Here, $\int_x \equiv \int d^d x$.) It then makes use of Cardy's linear transformation of the replica fields.[49] In $d = 6$, when dropping from the action terms that are irrelevant by simple power counting and terms that vanish when the number $n$ of replicas go to zero, one ends up with a theory that reproduces the main features of the Parisi-Sourlas SUSY action.[15] Note that once transformed the fields have different scaling dimensions: $\phi = (1/2)[\phi_1 + (\phi_2 + \cdots + \phi_n)/(n-1)]$, which is essentially the physical (order parameter) field, has a canonical dimension $D_\phi = (d-4)/2$, $\hat{\phi} = (1/2)[\phi_1 - (\phi_2 + \cdots + \phi_n)/(n-1)]$, which plays a role similar to the "response" field, has a canonical dimension $D_{\hat{\phi}} = d/2$, and the $(n-2)$ independent "antisymmetric" field combinations $\chi_i$, which somehow mimic the two anticommuting Grassmannian ghost fields $\bar{\psi}$ and $\psi$ of the SUSY formalism, have a canonical dimension $D_\chi = (d-2)/2$. (Cardy's formalism does not explicitly involves a renormalized temperature, whose dimension would be $-\theta = -2$ in the SUSY/DR regime, but the above field dimensions are identical to those obtained introducing a scaling in $1/T$ for $\phi$ and $1/\sqrt{T}$ for the $\chi_i$'s.)

The idea followed by KRT[1] is to study the scaling dimensions of the operators that have not been considered by Cardy[49] and investigate through a perturbative RG calculation at 2 loops in $\epsilon = 6 - d$ if these operators can become relevant for some value of $\epsilon$. The operators are classified into SUSY-writable, SUSY-null, and SUSY-non-writable, only the last two ones being potentially dangerous to destabilize the SUSY fixed point. There are several subtleties that are carefully handled by KRT. First, Cardy's transformation obscures the replica permutational symmetry $S_n$ of the action and one must make sure that only singlets under $S_n$ are retained. Second, these singlets are not eigen-operators of scale transformations and must be decomposed into terms of increasing scaling dimensions, the dominant one (of lowest dimension) being called the "leader".

KRT then find that two (leader) operators, denoted by $(\mathcal{F}_4)_L$ and $(\mathcal{F}_6)_L$, are more dangerous: if extrapolated, their scaling dimensions at 2 loops become relevant and destabilize the SUSY fixed point, in $d \approx 4.6$ for the former and in $d \approx 4.2$ for the latter.[1,2] There are other operators which appear at first sight even more dangerous because the extrapolated 2-loop expression of their scaling dimension crosses $d$ at a higher spatial dimension than those for $(\mathcal{F}_4)_L$ and $(\mathcal{F}_6)_L$. However, the authors argue that this does not take place due to nonperturbative mixing effects that should repel the scaling dimensions of operators belonging to the same symmetry class when they approach each other. The claim is that $(\mathcal{F}_4)_L$ and $(\mathcal{F}_6)_L$ are protected from this effect but not the others which are then conjectured to never become dangerous.

By construction, the operators under consideration are analytical functions of the replica fields (polynomials). The most dangerous ones have already been studied and discussed, albeit in a different framework, by Feldman[50] and later by two of us[25] for both the RFO($N$)M and the RFIM. As will be further developed, they are also accounted for in the nonperturbative FRG.[10,18,19,24–26] These operators are 2-replica functions of the form

$$\mathcal{F}_{2p} = \sum_{a,b} [(\phi_a - \phi_b)^2]^p.$$

(2)

(They are denoted by $A_p$ in Ref. [50].) In the RFO($N > 2$)M near the lower critical dimension of ferromagnetism, $d = 4$, the long-distance physics can be described by a nonlinear sigma model and the above operators can be rewritten as $\sum_{a,b}[1 - \phi_a \cdot \phi_b/(|\phi_a||\phi_b|)]^p$ which can then be represented as linear combinations of random anisotropies.[50]

In the RFIM close to its upper critical dimension $d_{\text{uc}} = 6$, one finds that these operators have a scaling dimension strictly larger than $d$ at tree level but acquire a negative anomalous dimension proportional to $p^2\epsilon^2$ at two loops, so that extrapolations may lead to an intriguing outcome: either, as proposed by Feldman,[50] one takes the limit of large $p$ at any fixed small $\epsilon$ and concludes that the operators always destabilize the SUSY/DR fixed point or, as done by KRT,[1] one considers the situation at a fixed $p$ and extrapolate at large $\epsilon$ to predict that the operators become relevant at some low enough specific dimension. The two scenarios lead to different pictures of SUSY/DR breaking. Our nonperturbative FRG yields yet another picture which dismisses the former scenario and, while having some overlap with the latter, also shows some key differences.

## B. Putting the above results in the framework of our FRG approach

The FRG allows one to derive exact flow equations for the cumulants of the effective average action, or scale-dependent Gibbs free-energy functional. Because the effective action is the generating functional of all 1-particle irreducible (1-PI) correlation functions[51], we will generically call the associated cumulants "1-PI cumulants". For practical purposes the exact FRG equations can be truncated in a nonperturbative approximation scheme. In the case of the RFIM this relies on the combined truncation of the expansions of the effective action in number of field derivatives and order of the cumulants. (A perturbative approximation scheme can of course also be used through an expansion in the $\phi^4$ coupling constant of the first cumulant, which is marginal at the upper dimension $d = 6$,

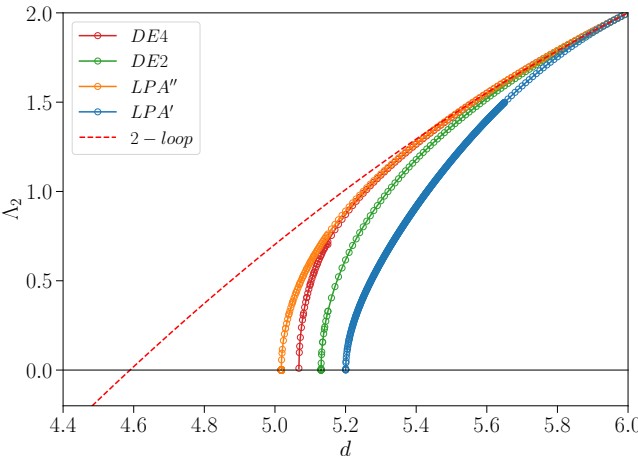

FIG. 1: Eigenvalue $\Lambda_2(d)$ corresponding to the most dangerous analytic perturbations around the SUSY/DR fixed point in the RFIM and associated with Feldman's operator $\mathcal{F}_4$. Dashed line: 2-loop calculation in $d = 6 - \epsilon$, together with a plausible extrapolation (the result coincide with that of KRT[1,2]); full lines: Results of successive levels of the nonperturbative approximation scheme of the FRG (LPA', LPA'', DE2, and DE4), which are discussed in Secs. IV C, V and Appendix E (full lines). Below $d_{\mathrm{DR}} \approx 5.11 \pm 0.09$ at which $\Lambda_2 = 0$ the SUSY/DR fixed point disappears and gives way to a cuspy fixed point at which both SUSY and DR are broken. Note that this disappearance which is associated with a square-root singularity in $\Lambda_2(d)$ is out of reach of the extrapolated perturbative expansion in $\epsilon$ which only suggests that the eigenvalue becomes negative (relevant) below $d \approx 4.6$.

and a subsequent expansion in $\epsilon = 6 - d$.) Contrary to the perturbative treatment in the coupling constant, the nonperturbative scheme provides an account of the functional dependence of the 1-PI cumulants in their field arguments. The truncation is chosen such that it does not explicitly break the symmetries and the Parisi-Sourlas SUSY of the theory. Furthermore, the 1-loop perturbative results are recovered in the vicinity of $d = 6$ (and, for the RFO($N > 2$)M, in the vicinity of the lower critical dimension of ferromagnetism, $d = 4$).

It should be stressed that all of the operators considered in the approach of KRT [1,2] have a counterpart in the FRG description, whether the latter makes use of conventional fields,[24,25] superfields,[18,19,26] or is derived within the dynamical formalism.[47,52] A difference that is worth mentioning is that our FRG formalism deals with 1-PI quantities present in the effective action, hence with somehow averaged operators which are functions of the average (replica) fields. On the other hand the operators studied by KRT are present in the action and involve the fluctuating (replica) fields. In the FRG the "averaged" operators are associated with a Taylor expansion of the functional dependence of the 1-PI cumulants of the renormalized random field. This can be directly seen for Feldman's operators which appear in the expansion of the second 1-PI cumulant as polynomials in the difference

between the two field arguments, $(\phi_a - \phi_b)$ [see Eq. (2)]. (For simplicity we will keep referring to such functions of the average fields appearing in the 1-PI cumulants as "operators" but one should keep in mind that they are not fluctuating quantities.) Most importantly, $\mathcal{F}_4$, which corresponds to the most dangerous perturbation that may destabilize the SUSY/DR fixed point and which is a SUSY-null operator in the formalism of Refs. [1,2] already played a key role in our FRG treatment because it signals when a cuspless, hence SUSY/DR, fixed point can no longer exist. We called the critical spatial dimension at which this happens $d_{\mathrm{DR}}$ and found it to be about 5.1. A check of the robustness of the prediction with an estimate of the error bar is provided below in Sec. V.

A crucial point is that the nonperturbative FRG is able to show the disappearance of the SUSY/DR fixed point altogether when the most dangerous analytic perturbation associated with Feldman's operator $\mathcal{F}_4$ becomes marginal, i.e., when $\Lambda_2 = 0$. The reason is that the nonperturbative FRG in fixed dimension $d < 6$ provides a full characterization of the effective action at the SUSY/DR fixed point and of the spectrum of eigenvalues (or equivalently of scaling dimensions) around this fixed point. Whereas the latter is determined from the linearization of the RG flow equations, the former is obtained via the resolution of fixed-point equations that may be *nonlinear* in some coupling constants (or rather functions). This should be contrasted with the conventional perturbative RG in which the only nonlinearity concerns the coupling constant that is marginal in $d = 6$.[53] Then, both the eigenvalues and the characteristics of the fixed-point effective action are derived as expansions in powers of this coupling constant (eventually turned into an expansion in $\epsilon = 6 - d$). If the fixed point disappears in a given dimension $d_{\mathrm{DR}} < 6$ because the (nonlinear) equation describing a specific coupling constant/function of the fixed-point effective action, in the present case that associated with the operator $\mathcal{F}_4$ which is irrelevant in $d = 6$, has no more solution due to the collapse with another fixed point, the expansion in $\epsilon$ used in [1,2] cannot *per se* capture this phenomenon. We will discuss additional symmetry arguments further below.

We illustrate the outcome of the two frameworks, nonperturbative FRG and conventional perturbative RG, for the eigenvalue $\Lambda_2(d)$ associated with the most dangerous perturbation for the SUSY/DR fixed point in Fig. 1. The perturbative calculation of KRT up to 2-loop order, which in the present case is also reproduced within the FRG (see below) and was already obtained by Feldman,[50] predicts a curve as a function of $\epsilon$ or $d$ that when extrapolated to lower dimension passes through 0 in $d \approx 4.6$ and then becomes negative. On the other hand, the nonperturbative FRG result coincides with the perturbative curve near $d = 6$ but strongly deviates from it as $d$ decreases and go to 0 in $d_{\mathrm{DR}} = 5.11 \pm 0.09$ (depending on the level of the nonperturbative approximation scheme: see Sec. V) with a singular square-root behavior. Below $d_{\mathrm{DR}}$ the SUSY/DR fixed point *no longer exists*. The ex-

trapolation of the perturbative result is of course blind to this feature. As we will also show in the next section, a similar phenomenon takes place in the RFO($N > 2$)M where an expansion in $1/N$ is structurally unable to detect the disappearance of the SUSY/DR fixed point.

As already pointed out in the Introduction, the FRG not only predicts the disappearance of the SUSY/DR fixed point in $d_{DR}$, but it also describes the appearance and the properties of the new fixed point below $d_{DR}$ at which both SUSY and DR are broken. And, much like in the case of elastic manifolds in a random environment, it relates this emergence to the appearance of a linear cusp in the functional dependence of the 1-PI cumulants of the renormalized random field at the zero-temperature fixed point. This is in turn associated with the fact that the long-distance physics at criticality is dominated by avalanches. As also already mentioned but worth stressing again, scale-free avalanches are present in all dimensions at criticality. Their effect is subdominant (the associated "cuspy" perturbation is irrelevant at the SUSY/DR fixed point) when $d \geq d_{DR}$ while they control the long-distance physics when $d < d_{DR}$. However, the disappearance of the SUSY/DR fixed point in $d_{DR}$ is not due to the avalanches and the cuspy perturbation *per se*. The latter is indeed still irrelevant in $d = d_{DR}$,[10,27] and there is a discontinuity in the associated eigenvalue because the nature of the fixed points (characterized by cuspless or cuspy 1-PI cumulants of the renormalized random field) is different above and below $d = d_{DR}$. This is illustrated below in the inset of Fig. 5.

To pinpoint $d = d_{DR}$ more accurately, and we agree on this conclusion with KRT, one should instead focus on Feldman's operator $\mathcal{F}_4$ and its scaling dimension (or, equivalently, the eigenvalue $\Lambda_2$). Yet, one should also consider the associated coupling constant/function at the fixed point. The critical dimension $d_{DR}$ can then be located in two ways: either looking at the appearance of a cusp in the fixed-point function (and a resulting divergence in some properly chosen derivative) as a function of dimension, as we did in our first nonperturbative FRG investigations,[19,24–26] or by studying the vanishing of the eigenvalue $\Lambda_2(d)$, as we did in [10,27] and in the present paper.

Finally, in their work[1,2] KRT also raise concerns about some aspects of our nonperturbative FRG approach. These concerns mostly stem from the unusual mechanism by which the SUSY/DR fixed point disappears in $d = d_{DR}$ to give way to a cuspy, SUSY/DR broken fixed point and from the peculiarities of a zero-temperature critical fixed point. This will be addressed below.

## III. A DETOUR VIA THE RFO($N$)M

### A. SUSY/DR fixed point, nonanalyticities, and dangerous operators

It is instructive to first consider the critical behavior of the RFO($N$)M which also has an underlying SUSY and is naively described by DR.[20] (The RFO($N$)M corresponds to the same replica field-theoretical action as in Eq. (1) with the fields now having $N$ components and the Lagrangian having an O($N$) instead of a $Z_2$ symmetry.) The RFO($N > 2$)M is more directly accessible to an analytical treatment when studied near its lower critical dimension $d = 4$ where it can be described through a nonlinear sigma model. Results have then be obtained both at 1-loop[20,25,27,46,50] and at 2-loop[46,55,56] order, so that all statements can be directly proven analytically and rather easily checked numerically. Note that the perturbative RG in $\epsilon = d - 4$ is *functional*, *i.e.*, deals with functions of the fields in place of coupling constants. Indeed, the scalar product of different replica fields, $\phi_a \cdot \phi_b$, is dimensionless in $d = 4$.

For the RFO($N$)M the study of Feldman's operators ("operator" being used here and below in an abuse of language to denote polynomials of the average replica fields that are the 1-PI counterpart of fluctuating operators, see above) is equivalent to considering the eigenvalues associated with the derivatives of the variance of the renormalized disorder $R(z)$, which corresponds to the second (1-PI) cumulant of the effective action, when the angle between the 2 replica-fields arguments $\phi_1$ and $\phi_2$ goes to 0 and its cosine, $z = \phi_2 \cdot \phi_2/(|\phi_1||\phi_2|)$, goes to 1. The first derivative $R'(z)$ is then the second (1-PI) cumulant of the renormalized random field. The eigenvalues are positive, *i.e.*, irrelevant, at large $N$ and become negative for some value as one decreases $N$. We studied them in detail, as well as the full functional dependence of the cumulant $R(z)$, in the perturbative FRG of the RFO($N > 2$)M in $d = 4 + \epsilon$ up to 2 loops.[25,27,46]

Let $R^{(p)}(1) = \partial_z^p R(z)|_{z=1}$ denote the $p$th derivative of the 1-PI second cumulant evaluated for equal replica fields. As discussed above, the associated Feldman operator is $\sum_{a,b}(1 - z_{ab})^p$, which is indeed of power $2p$ in the fields since $|\phi_a - \phi_b|^{2p} \propto (1 - z_{ab})^p$, where we recall that $z_{ab}$ is the cosine of the angle between the two replica vectors $\phi_a$ and $\phi_b$ which are taken to have (fixed) equal norm. The associated eigenvalue around the SUSY/DR fixed point is given at one loop in $d = 4 + \epsilon$ by[25]

$$\Lambda_{p \geq 2}(N) =$$
$$- \epsilon \Big[ \frac{2p^2 - (N-1)p + N - 2}{N - 2} + p(6p + N - 5)R''_*(1) \Big]$$
$$= -\frac{\epsilon}{N - 2} \Big[ 2p^2 - (N-1)p + N - 2 +$$
$$\frac{p(6p + N - 5)}{2(N + 7)}(N - 8 - \sqrt{(N-2)(N-18)}) \Big],$$

$$(3)$$

where we have used that the fixed point itself is characterized by

$$R'_*(1) = \frac{\epsilon}{(N-2)},$$

$$R''_*(1) = \epsilon \frac{N - 8 - \sqrt{(N-2)(N-18)}}{2(N-2)(N+7)}. \tag{4}$$

The above value of $R'_*(1)$, which determines the critical exponents $\eta$, $\bar{\eta} = \eta$, and $\nu$, together with the existence of a finite $R''_*(1)$ are enough to guarantee that the fixed point indeed satisfies SUSY and DR.[20,25,27,46] Note that $R''_*(1)$ and $\Lambda_{p\geq 2}(N)$ are real for $N \geq 18$ only (we restrict ourselves to the case $N > 2$). The 2-loop calculation leads to similar results,[46] but for our illustration purpose it is sufficient to consider the 1-loop description.

Loosely relating the role of $N$ here with that of $d$ in the RFIM, we observe that the value of $N$ at which the eigenvalue $\Lambda_p$ goes to zero increases with $p$, $i.e.$, Feldman's operators of high $p$ become relevant $before$ those for small $p$ (with $p = 2$ corresponding to $\mathcal{F}_4$), much like what is found for the RFIM at 2 loops below $d = 6$. However, without invoking a nonperturbative mechanism of level repulsion, one finds that the vanishing of an eigenvalue $\Lambda_{p\geq 3}$ around the SUSY/DR fixed point is cured by a change in the functional dependence of the fixed-point cumulant $R_*(z)$, which can then acquire a nonanalytical dependence of the form $(1-z)^{1+\alpha}$, with $\alpha$ real and $> 1$, in the vicinity of $z = 1$.[25,27,46] In the restricted sector associated with $R'_*(1)$ and $R''_*(1)$ the fixed point still shows SUSY/DR. All of this is further discussed below.

We also noticed that the expression in Eq. (3) can be analytically continued to noninteger values of $p$, $p = 1+\alpha$, and that it then controls the RG flow of the amplitude $a_k(\alpha)$ of a nonanalytic term in $(1-z)^{1+\alpha}$ in $R_k(z)$ in the vicinity of $z = 1$:

$$\partial_t a_k(\alpha) = \Lambda_{1+\alpha} a_k(\alpha), \tag{5}$$

where $t = \ln(k/k_{\text{UV}})$ is the RG time with $k$ the running IR cutoff scale and $k_{\text{UV}}$ the initial UV scale. The fixed point with $a_*(\alpha) = 0$ is unstable to a nonanalytical pertubation behaving as $(1-z)^{1+\alpha}$ near $z = 1$ when $\Lambda_{1+\alpha} < 0$ and a new fixed point $R_*(z)$ with $a_*(\alpha) \neq 0$ emerges when $\Lambda_{1+\alpha} = 0$. For a given $N$ this occurs for a specific value $\alpha_\#(N)$ such that $\Lambda_{1+\alpha_\#(N)}(N) = 0$. (Note that due to the simple form of Eq. (5), the value of $a_*(\alpha)$ when $\Lambda_{1+\alpha} = 0$ is determined by requiring not only that the fixed point be stable but that the function $R_*(z)$ be defined over the whole interval of $z$ between $-1$ and $1$; in particular, $R_*(z)$ should be finite when $z = -1$.[56]) From the work of Sakamoto $et$ $al.$ on the allowed nonanalytical perturbations,[56] one can conclude that the only acceptable solutions are with $\alpha = 1/2$, which corresponds to a term in $\sqrt{1-z}$ in the second cumulant of the renormalized random field $\Delta(z) = R'(z)$ and we refer to as a "cusp", and with $\alpha > 1$, which we refer to as "subcusps". We illustrate the appearance of such a subcusp by looking at the SUSY/DR fixed point in $N = 18$. There, $\Lambda_{5/2} = 0$

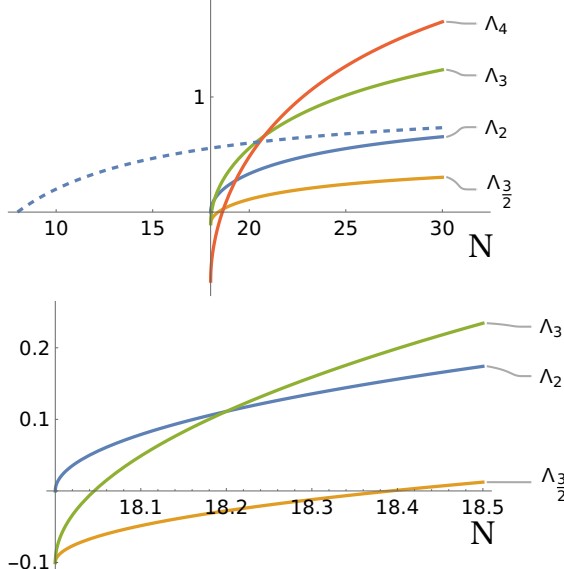

FIG. 2: Stability of the (most stable) SUSY/DR fixed point in the RFO($N$)M at 1-loop in $d = 4 + \epsilon$: Variation with $N$ of the lowest-order eigenvalues $\Lambda_p(N)$; $p = 2,3$ and 4 correspond to Feldman's operators $\mathcal{F}_4$, $\mathcal{F}_6$, and $\mathcal{F}_8$, whereas $p = 3/2$ corresponds to a nonanalytical perturbation (a cusp in $R'(z) = \Delta(z)$). The eigenvalue $\Lambda_{3/2}$ becomes relevant below $N = 2(4 + 3\sqrt{3}) \approx 18.3923\cdots$ and $\Lambda_2$ becomes zero in $N = 18$. The vertical lines indicate the value $N = 18$. Also shown is an extrapolation of the large $N$ expression (up to next-to-leading order in $1/N$). This extrapolation vanishes for $N \sim 8$ and is clearly blind to the disappearance of the SUSY/DR fixed point in $N = 18$. The bottom panel is a zoom in displaying the region near $N = 18$.

and a subcusp in $(1-z)^{3/2}$ is expected in $\Delta_*(z) = R'_*(z)$. The DR fixed point in $N = 18$ is also marginal with respect to $\Delta'(1) = R''(1)$, $i.e.$ $\Lambda_2 = 0$, and unstable with respect to the cuspy fixed point, with $\Lambda_{3/2} = -\epsilon/10$. However, the fixed-point values of $\Delta'(1)$ and $\Delta''(1)$ are exactly known, and one can numerically find the full $z$-dependence of the fixed-point function $\Delta_*(z)$. It unambiguously displays a subcusp in $(1-z)^{3/2}$ when $z \to 1$, as shown in detail in Appendix A.

We also proved the important property that SUSY and DR are $only$ $broken$ $in$ $the$ $presence$ $of$ $a$ $cusp$ in $\Delta_*(z) = R'_*(z)$. Weaker nonanalyticities (subcusps) do not prevent the main scaling behavior from following DR and do not break the SUSY Ward identities. To derive all of these results, which carry over to the 2-loop level,[46,57] the functional nature of the RG is crucial.

We display in Fig. 2 the lowest-order eigenvalues $\Lambda_p(N)$ for $p = 2,3$ and 4 (corresponding to Feldman's operators $\mathcal{F}_4$, $\mathcal{F}_6$, and $\mathcal{F}_8$), for $p = 3/2$ [the "cuspy" perturbation to the second cumulant of the random field $\Delta_*(z) = R'_*(z)$]. One can see that in the present case the amplitude of the cusp, or equivalently of the $(1-z)^{3/2}$ term in $R_*(z)$, becomes relevant when the associated eigenvalue $\Lambda_{3/2}$ changes sign, which takes place

in $N = 2(4 + 3\sqrt{3}) = 18.3923\cdots$. The stable fixed point is then a cuspy one with a $(1 - z)^{3/2}$ nonanalytic dependence in $R_*(z)$ in the vicinity of $z = 1$ [and a cusp in $\Delta_*(z) = R'_*(z)$]. This happens before Feldman's operator $\mathcal{F}_4$ becomes marginal, *i.e.*, before $\Lambda_2 = 0$, which takes place in $N = 18$. The key point, however, is that the SUSY/DR fixed point *no longer exists* below $N = 18$, as is easily seen from the expression of $R''_*(1)$ in Eq. (4). When $N < 18$ it only remains the cuspy fixed point. SUSY and DR are broken at this cuspy fixed point. Note that if one uses a large-$N$ expansion for the eigenvalue $\Lambda_2$, one finds as illustrated in Fig. 2 (top) up to the next-to-leading order in $1/N$, that $\Lambda_2$ can be extrapolated to 0 (here, for $N \approx 8$) but the extrapolated expansion is of course blind to the disappearance of the SUSY/DR fixed point in $N = 18$.

So, as one decreases $N$, the SUSY/DR fixed point first acquires nonanalytic terms that do not break SUSY/DR, then becomes unstable in $N_{\text{cusp}} = 2(4 + 3\sqrt{3})$ to a cuspy fixed point at which SUSY and DR are broken, and it finally disappears in $N = 18$, which is also when $\Lambda_2$ vanishes. This coincidence of the vanishing of $\Lambda_2$ and the disappearance of the SUSY/DR fixed point is a central feature that can only be found by considering a nonperturbative treatment in $N$ (as opposed to an expansion in $1/N$) and by studying the RG equations for the fixed point itself (and not only the linearized version for the determination of the eigenvalue spectrum). It results from an exact property of the FRG flow equations (beyond 1- and 2-loop results): Whereas the flow equations for $R^{(p \geq 3)}(1)$ (associated with $\mathcal{F}_{2p}$) is linear, that for $R''(1)$ is nonlinear. At 1-loop order when keeping $R'(1)$ at its SUSY/DR fixed-point value $\epsilon/(N - 2)$, one for instance finds

$$\partial_t R''_k(1) =$$
$$- (N + 7)R''_k(1)^2 + \frac{\epsilon(N - 8)}{N - 2}R''_k(1) - \frac{\epsilon^2}{(N - 2)^2} \quad (6)$$

and, with $R''_k(1)$ kept fixed at its fixed-point value $R''_*(1)$ given in Eq. (4),

$$\partial_t R^{(p \geq 3)}_k(1) = \Lambda_p(N)R^{(p \geq 3)}_k(1) + \mathcal{G}_{p,k}(N), \quad (7)$$

where the $\Lambda_{p \geq 2}(N)$'s are given by Eq. (3) and the $\mathcal{G}_{p \geq 3,k}(N)$'s are functions of the $R^{(q)}_k(1)$'s with $q \leq p - 1$.[25] (We recall that $t = \ln(k/k_{\text{UV}})$ is the RG time.) At 2-loop order, the equation for $R^{(p \geq 3)}_k(1)$ stays linear and that for $R''_k(1)$ becomes cubic. The degree of the nonlinearity for the latter increases with the loop order, so that at $j$-loop order, $\partial_t R''_k(1) = Q_{N,j}(R''_k(1))$ where $Q_{N,j}$ is a polynomial of degree $j$. This nonlinearity is responsible for the fact that there is no solution for the SUSY/DR fixed point $R''_*(1)$ below the value of $N$ at which $\Lambda_2(N)$ vanishes. Indeed, the fixed-point value is a solution of $Q_{N,j}(R''_*(1)) = 0$ and $\Lambda_2$ is given by $\Lambda_2(N) = Q'_{N,j}(R''_*(1))$, where $Q'_{N,j}$ is the derivative of the polynomial. One expects $\Lambda_2$ to be positive

for $N > N_{\text{DR}}$ (with $N_{\text{DR}} = 18$ at one loop) so that $Q'_{N,j}(R''_*(1)) > 0$. When $N = N_{\text{DR}}$, $\Lambda_2 = Q'_{N,j}(R''_*(1)) = 0$ while $Q_{N,j}(R''_*(1)) = 0$. Generically, and in the absence of an additional symmetry, this corresponds to the collapse of the SUSY/DR solution with another solution, such that the SUSY/DR solution disappears altogether for $N < N_{\text{DR}}$: see Appendix B. Although not a rigorous proof, this strongly supports the fact that *the SUSY/DR fixed point no longer exists below the value $N_{\text{DR}}$ at which the eigenvalue $\Lambda_2$ associated with the operator $\mathcal{F}_4$ vanishes*.

We stress again that the perturbative expansions in $1/N$ around the large-$N$ limit *cannot* capture the annihilation of fixed points even if it predicts through an extrapolation the vanishing of the extrapolated eigenvalue $\Lambda_2(N)$.

## B. More on the SUSY/DR fixed points and their stability

In what follows we study in more detail the issue of the stability of the fixed points and the existence of the putative unstable fixed point that coalesces and annihilates with the SUSY/DR fixed point when $N = N_{\text{DR}}$.

As we have already mentioned, when $N < N_{\text{cusp}} = 2(4 + 3\sqrt{3}) \approx 18.39$ (at 1-loop order), the stable SUSY/DR fixed point becomes unstable to a nonanalytic perturbation associated with a $(1 - z)^{3/2}$ term in $R(z)$ [and a cusp in $\Delta(z) = R'(z)$] near $z = 1$ and characterized by the eigenvalue $\Lambda_{3/2}$. The SUSY/DR fixed point is cuspless but has nonetheless a weak singularity in $(1 - z)^{1+\alpha_*(N)}$ with $\alpha_*(N) \geq 3/2$.

Consider now the putative unstable fixed point that collapses with the cuspless fixed point when $\Lambda_2(N) = 0$. This fixed point is characterized by the same value of $R'_*(1)$ as the SUSY/DR most stable fixed point (*e.g.*, $\epsilon/(N - 2)$ at one loop). It has therefore the same critical exponents $\eta$, $\bar{\eta}$, and $\nu$ because these exponents are obtained from $R'_*(1)$ only.[25,46] At one loop, the fixed point is also specified by $R''_*(1) = \epsilon[N - 8 + \sqrt{(N - 2)(N - 18)}]/[2(N - 2)(N + 7)]$, which is the other solution of Eq. (6), and it is unstable in the direction $R''(1)$. More generally, the eigenvalue associated with $R^{(p)}(1)$ around this unstable SUSY/DR fixed point is given by a simple modification of Eq. (3),

$$\Lambda_{p \geq 2}(N) = - \frac{\epsilon}{N - 2}\Big[2p^2 - (N - 1)p + N - 2 +$$
$$\frac{p(N - 5 + 6p)}{2(N + 7)}(N - 8 + \sqrt{(N - 2)(N - 18)})\Big],$$
$$(8)$$

which can also be extended to real values $p = 1 + \alpha$ associated with a nonanalytic perturbation in $(1 - z)^{1+\alpha}$ near $z = 1$. We note that for $N > 18.001785\cdots$ all eigenvalues $\Lambda_p(N)$ are strictly negative, meaning that all the associated directions are relevant: see Fig. 3(b). For $N > 18.001785\cdots$, provided the eigenperturbations can

be extended over the whole interval $-1 \leq z \leq 1$, this fixed point therefore appears so unstable that a whole function $R_*(z)$ must be fine-tuned at the start of the FRG flow, even if one restricts the initial condition to analytical functions. So, this fixed point is completely unphysical.

The situation is different when $18 \leq N \leq 18.001785\cdots$. In this case, there are two zeros for each $N$, $\alpha_-(N)$ and $\alpha_*(N)$, with $1 < \alpha_-(N) \leq \alpha_*(N) < 3/2$: see Fig. 3(b). As for the other SUSY/DR fixed point, we expect that the (several times) unstable SUSY/DR fixed point acquires a nonanalytical dependence that behaves as $(1-z)^{1+\alpha_-(N)}$ near $z = 1$. (Since one is dealing with full functions one must check that the corresponding $R_*(z)$ is defined over the full interval $-1 \leq z \leq 1$, but this seems possible by forming a linear combination with a solution that behaves as $(1-z)^{1+\alpha_*(N)}$ near $z = 1$.) The pending question is that of the stability of this fixed point in the present domain of $N$. It is unstable in the direction $R''(1)$ and is also unstable in the direction of the cusp ($\alpha = 1/2$), which implies that it is unstable with respect to both the cuspless SUSY/DR more stable fixed point and to the cuspy fixed point that is the most stable for $N < 2(4 + 3\sqrt{3}) \approx 18.39$. In addition, however, all eigenvalues $\Lambda_{1+\alpha}(N)$ with $1 < \alpha \leq \alpha_-$ are also negative, *i.e.*, relevant, and associated with acceptable eigenfunctions when following the same procedure as before.[27,56] Nonetheless, over the very narrow interval of values of $N$, $18 \leq N \leq 18.001785\cdots$, one expects that the fixed point can be found by choosing initial conditions of the FRG flow that are restricted to analytical functions $R_0(z)$ and by fine-tuning the two remaining relevant directions, $R'(1)$ and $R''(1)$, to their fixed-point values.

### C. Recap

To sum up: Around the SUSY/DR critical fixed point in the RFO($N$)M in $d = 4 + \epsilon$, Feldman-like operators of the form $(1-z)^p$ near $z = 1$ become marginal (and potentially relevant) as $N$ decreases from infinity but, provided $p > 2$, this is cured by the fact that the SUSY/DR fixed point acquires a weak nonanalytical behavior in $(1-z)$, which does not break SUSY nor DR in the main sector of the critical behavior. The most dangerous operators are then $\mathcal{F}_4$, which is characterized by the eigenvalue $\Lambda_2(N)$, and the cuspy perturbation in $(1-z)^{3/2}$ near $z = 1$, which is characterized by the eigenvalue $\Lambda_{3/2}(N)$. When $\Lambda_{3/2}(N) = 0$ the SUSY/DR fixed point becomes unstable to a cuspy perturbation and there is an exchange of stability with another fixed point at which the second cumulant of the renormalized random field $\Delta(z) = R'(z)$ has itself a cusp and is therefore associated with a breakdown of SUSY and DR. This takes place for $N_{\text{cusp}} = 2(4 + 3\sqrt{3}) \approx 18.39$ at 1-loop order. Below this value the SUSY/DR fixed point continues to exist down to $N = 18$, at which $\Lambda_2 = 0$ ($\mathcal{F}_4$ becomes marginal) and, at the same time, a coalescence with yet another unsta-

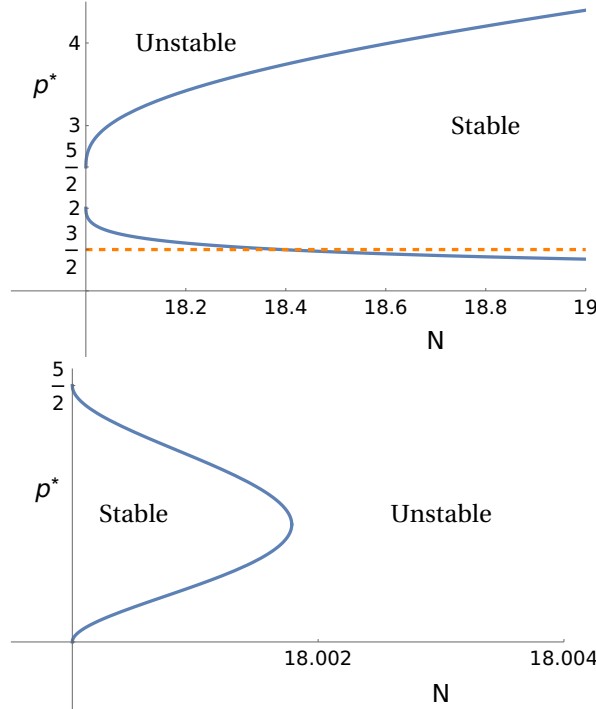

FIG. 3: Stability of the SUSY/DR fixed points in the RFO($N$)M at 1-loop order in $d = 4 + \epsilon$. Top: Zeros of the eigenvalue $\Lambda_p(N)$, where $p = 1 + \alpha$ is extended to real values, for perturbations around the (most) stable fixed point. The dashed line corresponds to $p = 3/2$, *i.e.*, to a cusp in $R'(z) = \Delta(z)$, which is the only acceptable value for $p < 2$.[27,56] The eigenvalue is negative (relevant) above the top (orange) curve $\alpha_*(N)$ and positive (irrelevant) between this curve and the bottom (blue) curve $\alpha_-(N)$. Below $N = 2(4 + 3\sqrt{3}) \approx 18.3923\cdots$, the fixed point is unstable to the cuspy perturbation, *i.e.*, $p = 3/2$ or equivalently $\alpha = 1/2$. Bottom: Zeros of the eigenvalue $\Lambda_{p=1+\alpha}(N)$ for perturbations around the unstable fixed point. To the right of the curves the eigenvalues are relevant and to the left they are irrelevant. The top and bottom curves giving the two zeros, $\alpha_*(N)$ and $\alpha_-(N)$, merge in $N = 18.001785\cdots$.

ble SUSY/DR fixed takes place. Below $N = 18$ there are *no* SUSY/DR fixed points. This phenomenon cannot be found through a perturbative expansion in $1/N$ around the large-$N$ limit.

All of the above is true at both 1- and 2-loop orders, and we have given arguments extending the conclusion to any loop order. One notices from the 2-loop calculation that the two curves associated with $\Lambda_2(N) = 0$ and $\Lambda_{3/2}(N) = 0$ tend to move toward each other as $\epsilon$ increases and, if extrapolated, would cross near $N \sim 14$ and $d \sim 4.4$. At this intersection, $\Lambda_{3/2}$ and $\Lambda_2$ vanish together, and for lower values of $N$, which include the RFIM, one expects that $\Lambda_2$ goes to zero before (*i.e.*, at a higher $d$ than) $\Lambda_{3/2}$. This is sketched in Fig. 4.

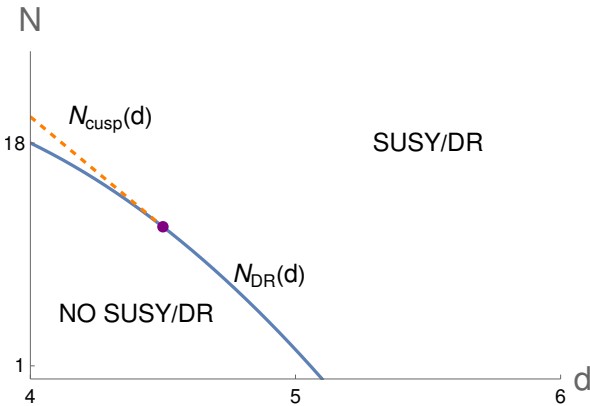

FIG. 4: Schematic phase diagram of the RFO($N$)M in the ($N$, $d$) plane. The full line, $N_{\mathrm{DR}}(d)$, is where the eigenvalue $\Lambda_2$ associated with Feldman's operator $\mathcal{F}_4$ vanishes and the cuspless SUSY/DR fixed point disappears. The dashed line, $N_{\mathrm{cusp}}(d)$, is where the eigenvalue $\Lambda_{3/2}$ associated with a cusp in $R''(z)$ vanishes and the cuspless SUSY/DR fixed point becomes unstable with respect to a cuspy fixed point. The two lines meet around $N_x \approx 14$ and $d_x \approx 4.4$ (estimated from a 2-loop perturbative FRG in $d = 4 + \epsilon^{27,46}$).For $N < N_x$ and $d > d_x$, which includes the RFIM, the SUSY/DR critical fixed point disappears when it is still stable with respect to a cuspy perturbation, i.e., $\Lambda_{3/2} > 0$ when $\Lambda_2 = 0$.

## IV. BACK TO THE RFIM

### A. Dangerous operators and their scaling dimensions around the SUSY/DR fixed point

We first address the issue of the dimension of Feldman's operators [given in Eq. (2)] in the RFIM near $d = 6$. The question of determining the scaling dimensions at the fixed point, even without invoking nonanalytical field dependences, is unexpectedly rather subtle in the RFIM. The problem is found in the usually trivial operation of finding the canonical dimensions of the operators. In the replica approach of KRT,[1,2] the difficulty comes with the limit $n \to 0$ in the number of replicas. Then, as stressed in [2], the 2-point correlation functions of $S_n$ invariant operators vanishes. In principle scaling dimensions can be extracted from the operator product expansion (OPE) and/or by considering higher-order correlation functions with other operators but the procedure is much far from straightforward. In the FRG approach based on a cumulant expansion and the description of the fixed point as being at zero (renormalized dimensionless) temperature,[18,19,24,32,34,37] the difficulty lies in the fact that cumulants of different orders come with different powers of the inverse temperature and that, as a consequence, the limit $T \to 0$ must be performed separately for each order.

To give examples, near $d = 6$, the canonical dimension of the leader of Feldman's operator $\mathcal{F}_4$, which involves products of 4 transformed fields, is taken in [1] as $\Delta_{(\mathcal{F}_4)_L} = 2(d - 2)$ and that of $\mathcal{F}_6$, which involves products of 6 transformed fields, as $\Delta_{(\mathcal{F}_6)_L} = 3(d - 2)$. In the FRG near the zero-temperature fixed point, the counterpart of $\mathcal{F}_4$ is present in the second 1-PI cumulant and involves 4 replica fields and a factor of $1/T^2$. It has a canonical dimension of $2(d - 4) + 4$, where $(d - 4)/2$ is the scaling dimension of the fields and 2 that of the inverse temperature, and this indeed corresponds to $\Delta_{(\mathcal{F}_4)_L}$ as predicted in [1]. On the other hand, the term corresponding to Feldman's $\mathcal{F}_6$ involves products of 6 replica fields and, also being in the second cumulant, a factor of $1/T^2$, so that its canonical dimension is $3(d - 4) + 4$, which disagrees with the prediction of [1].

Accordingly, the eigenvalues of the linearized flow equations around the (Gaussian) fixed point should be $\Lambda_2 = \Delta_{(\mathcal{F}_4)_L} - d = d - 4$ for both [1] and the FRG, and $\Lambda_3 = \Delta_{(\mathcal{F}_6)_L} - d = 2d - 6$ for [1] and a less irrelevant value $\Lambda_3 = 2d - 8$ for the FRG. (Note that in [50] Feldman appears to have used the canonical dimensions for the nonrandom $\phi^4$ theory, i.e., a dimension of $(d - 2)/2$ for the replica fields which however does not predict the right upper critical dimension in the presence of a random field nor the right lower critical dimension for the RFO($N$)M; this choice also gives $\Lambda_3 = 2d - 6$ as in KRT.)

In the FRG approach, the scaling dimensions are fixed by both casting the flow equations in a dimensionless form such that a fixed point can effectively be found and then determining the spectrum of all eigenvalues by solving the linearized FRG equations near this fixed point. In the RFIM where, as stressed above, there is an ambiguity in choosing the canonical dimensions of some of the irrelevant operators, one may fix the ambiguity by considering the loop contributions to the flow equations (or the fixed-point equations) of the coupling constants/functions associated with these operators. By construction, the eigenvalue equations are indeed linear in the coupling constants/functions themselves (which is a reason for the mentioned ambiguity) but the loop contributions to the flow equations are nonlinear in other coupling constants/functions, which helps determining the proper canonical (engineering) dimensions. The latter should then indeed be fixed by finding a consistent matching between the dimensions of the tree-level contributions and those of the loop contributions. We discuss this in detail in Appendix C, where we illustrate the subtleties coming from Cardy's replica field transformation and from sums over replicas. We also point out the disagreements with KRT.[1,2] Further work would nonetheless be needed to settle the issue.

We close this discussion by stressing two points:

- The problem with the canonical scaling dimensions comes only for operators (or their 1-PI counterparts) that involve sums over replicas, such as $\mathcal{F}_{2p}$ in Eq. (2) or the associated leaders given in [1]. For the random-field free scalar field theory, which describes the RFIM at the upper critical dimension and has been for instance recently studied in

[58], there is an agreement between the various formalisms concerning the scaling of the correlations functions of the physical primary (and composite) fluctuating fields. Sums over replicas are what brings an unusual ambiguity.

- Most importantly, there is an agreement among all derivations concerning the canonical dimension of Feldman's operator $\mathcal{F}_4$ which turns out to be the most dangerous for the SUSY/DR fixed point below $d = 6$. The main conclusions of this paper are therefore not affected by the discrepancy about other canonical dimensions, such as those of the $\mathcal{F}_{2p}$'s with $p \geq 3$.

### B. Perturbative calculation at 2-loop order of the scaling dimensions/eigenvalues

When considering the "anomalous" contributions to the scaling dimensions of Feldman's operators at the SUSY/DR fixed point, i.e., the contributions coming from the Feynman diagrams of the loop expansion beyond the tree level, the outcome is the same whether the calculation is performed within the FRG as here and in [50] or with Cardy's transformed fields as in [1].

For computing the scaling dimension of an irrelevant operator in perturbation theory, the general strategy goes as follows. One first lists the operators (call them $\mathcal{O}_i$) with the same canonical dimension and the same symmetries. The bare action is then perturbed by a term of the form $\sum_i x_i \mathcal{O}_i$. The calculation consists in computing the beta functions for the coupling constants, $\beta_{x_i}$, at linear order in $x_j$. (Note then that the canonical dimensions do not enter the calculation of the anomalous contributions to the scaling dimensions.) The scaling dimensions are related to the eigenvalues of the matrix $M_{ij} = \frac{\partial(\beta_{x_i})}{\partial x_j}|_{x_k=0}$. Consider for illustration the eigenvalue $\Lambda_2$ associated with Feldman's operator $\mathcal{F}_4$. A major simplification takes place because $\mathcal{F}_4$ does not mix with the other operators of the same dimension. If we attribute the index 1 to this operator ($\mathcal{O}_1 = \mathcal{F}_4$), it can be checked that the associated coupling constant ($x_1$) does not contribute to the RG flows of the other coupling constants $x_{i>1}$, at least at two-loop order. The matrix $M_{ij}$ is therefore such that $M_{i1} = 0$ for $i > 1$. In this situation, the sought eigenvalue is simply equal to $M_{11}$. In a second step, it is necessary to consider the diagrams built with one operator $\mathcal{F}_4$ which renormalize the coupling constant $x_1$. The one-loop contribution vanishes, as first observed by Feldman,[50] and we obtain the two-loop one as

$$\Lambda_2^{2-\text{loop}}(\epsilon) = -\frac{8}{27}\epsilon^2. \qquad (9)$$

More generally, the output of the calculation for the anomalous contribution is

$$\Lambda_p^{2-\text{loop}}(\epsilon) = -\frac{p(3p-2)}{27}\epsilon^2, \qquad (10)$$

which coincides with the results of Feldman[50] and KRT.[1]

The above results lead to several comments:

- For $p = 2$, and as already stressed, all the derivations agree and one has

$$\Lambda_2(\epsilon) = 2 - \epsilon - \frac{p(3p-2)}{27}\epsilon^2, \qquad (11)$$

which indeed coincides with $\Delta_{(\mathcal{F}_4)_L} - d$ in [1]. This is what is plotted in Fig. 1. For $p > 2$ on the other hand we find

$$\Lambda_p(\epsilon) = 2p - 2 - (p-1)\epsilon - \frac{p(3p-2)}{27}\epsilon^2, \qquad (12)$$

which differs by an additive term $-2(p-2)$ from the result of [1], a discrepancy discussed above.

- Setting $p = 3/2$ in Eq. (12) gives back our previous result for the eigenvalue $\Lambda_{3/2}$ associated with the cuspy perturbation stemming from the presence of scale-free avalanches,[39] $\Lambda_{3/2}(\epsilon) = 1 - \epsilon/2 - (5/36)\epsilon^2$. Note that when extrapolated to finite values of $\epsilon$ or low values of $d$, this eigenvalue $\Lambda_{3/2}(\epsilon)$ vanishes in $d \approx 4.57$ whereas $\Lambda_2(\epsilon)$ which is associated with $\mathcal{F}_4$ vanishes in a slightly higher dimension $d \approx 4.59$, so that the cuspy perturbation is still irrelevant when $\Lambda_2$ vanishes. (Here in the RFIM, we keep the terminology "cusp" and "cuspy" to denote a dependence in the second cumulant of the renormalized random field $\Delta(\phi_1, \phi_2)$ in $|\phi_1 - \phi_2|$ when $\phi_1 \to \phi_2$. More properly, the cusp arises when the cumulant, which is an even function of the field difference, is considered as a function of $(\phi_1 - \phi_2)^2$ but we will keep the terminology for simplicity. In a similar vein, we will use "subcusp" to denote weaker singularities in the field difference.)

### C. Disappearance of the SUSY/DR fixed point for $d < d_{\text{DR}}$: A nonperturbative FRG calculation

As already stressed several times, a main difference of interpretation with KRT[1,2] is that in our nonperturbative FRG calculations the vanishing of $\Lambda_2$ in $d = d_{DR}$ coincides with the disappearance of the SUSY/DR fixed point. This is at odds with their scenario in which the SUSY/DR fixed point becomes unstable but is still present below $d_{\text{DR}}$. We emphasize that the issue cannot be directly probed through the perturbative approach that relies on an expansion in the $\phi^4$ coupling constant and/or in $\epsilon = 6 - d$. It instead requires a nonperturbative and functional RG which allows one to investigate not only the stability of the SUSY/DR fixed point but also its very existence. This is much like the situation in the RFO($N$)M near its lower critical dimension which we have presented in detail above, with the dimension $d$ now playing the role of

the number of components $N$; the shortcomings of the $\epsilon = 6 - d$ expansion are then similar to those of the $1/N$ expansion. Note that symmetry arguments beyond the inconclusive perturbative calculation may also be invoked.[1,2] Indeed, the coalescence of two fixed points when an operator becomes marginal conventionally takes place when the marginal operator does not break the symmetries of the merging fixed points. Whether in the present case the nonperturbative prediction of the coalescence of two SUSY/DR fixed points in $d_{DR}$ stems from the unusual, functional, character of the pattern of fixed-point disappearance/appearance or relates the property that the marginal operator $\mathcal{F}_4$ is SUSY-null instead of strictly SUSY-nonwritable (in the language of KRT) remains to be clarified.

The central quantity of the nonperturbative FRG for disordered systems is the scale-dependent effective action (or Gibbs free-energy functional) $\Gamma_k[\{\phi_a\}]$ which incorporates fluctuations down to some imposed IR cutoff $k$ and represents the generating functional of all 1-PI correlation functions at this (running) scale $k$.[59] The IR cutoff is implemented through the introduction of regulator functions in such a way that they do not explicitly break the Ward identities associated with the Parisi-Sourlas SUSY[18,19,26] (see also below). This disorder-averaged effective average action depends on the average (or background) fields $\phi_a$ in an arbitrarily large number of copies (or replicas) $a = 1, 2, \cdots$ of the original system, which allows one to generate the 1-PI cumulants with their full functional dependence through an expansion in increasing number of free replica sums,[12,18,24]

$$\Gamma_k[\{\phi_a\}] = \sum_a \Gamma_{k1}[\phi_a] - \frac{1}{2}\sum_{a,b}\Gamma_{k2}[\phi_a, \phi_b] + \frac{1}{3!}\sum_{a,b,c}\Gamma_{k3}[\phi_a, \phi_b, \phi_c] + \cdots,$$

(13)

where $\Gamma_{k,p=1}$ is the disorder-averaged Gibbs free energy at scale $k$ and the $\Gamma_{kp}$'s for $p \geq 2$ are essentially the cumulants of the renormalized disorder at the scale $k$. The evolution of the 1-PI cumulants $\Gamma_{kp}$ with the IR scale $k$ down to $k = 0$ is governed by a hierarchy of *exact* functional RG flow equations: see Appendix E. (Note that for avoiding the introduction of too many symbols we have used the same notation for the fields in the bare action of Eq. (1) and for the background fields involved in the scale-dependent effective action, although the latter are the averages of the former at the scale $k$.)

Truncations are however necessary to turn this exact hierarchy into an operational scheme for studying the FRG flows toward fixed points. An efficient ansatz that can capture the long-distance physics, including the influence of avalanches and droplets which we have argued to be central to the critical behavior of the RFIM,[19,25,28,47] consists in truncating the expansion in derivatives of the fields and at the same time truncating the expansion in cumulants. This must be done in such a way that the

Parisi-Sourlas SUSY is not explicitly broken (this is of course true as well for all the other symmetries). In our previous studies of criticality in the RFIM,[10,12,19,26,27] we have considered the second order of the approximation scheme in which we keep the first 1-PI cumulant of at the second order of the derivative expansion, the second 1-PI cumulant at the local potential approximation level, and we neglect higher-order cumulants,

$$\Gamma_{k1}[\phi_1] = \int_x \left[ U_k(\phi_1(x)) + \frac{1}{2}Z_k(\phi_1(x))(\partial_\mu \phi_1(x))^2 \right],$$

$$\Gamma_{k2}[\phi_1, \phi_2] = \int_x V_k(\phi_1(x), \phi_2(x)),$$

$$\Gamma_{kp \geq 3} = 0.$$

(14)

The effective average potential $U_k(\phi_1)$ describes the thermodynamics of the system, $Z_k(\phi_1)$ is the field-renormalization function, and $V_k(\phi_1, \phi_2)$ is the 2-replica effective average potential whose second derivative, $V_k^{(11)}(\phi_1, \phi_2) = \Delta_k(\phi_1, \phi_2)$, is the second cumulant of the renormalized random field at zero momentum. $\Delta_k(\phi_1, \phi_2)$ is the key quantity that tracks the effect of avalanches and droplets through its functional dependence. We refer to this approximation as DE2 because it involves an expansion up to the second order of the derivative expansion (DE) in the first cumulant. We will discuss other levels of the nonperturbative approximation scheme, as well as the issues of robustness and convergence in Sec. V.

Inserting the above ansatz into the exact FRG equations for the cumulants leads to a set of coupled flow equations for 3 functions, $U_k(\phi_1)$, $Z_k(\phi_1)$, and $V_k(\phi_1, \phi_2)$: see Refs. [10,19,26]. Fixed points describing scale invariance and the spectrum of eigenvalues around them can then be found by casting the resulting FRG flow equations in a dimensionless form. As we are searching for zero-temperature fixed points,[48,62,63] we define a dimensionless renormalized temperature $\widetilde{T}_k$ which flows to zero as $k \to 0$ (this is the precise meaning of a "zero-temperature" fixed point) and we introduce scaling dimensions such that the dimensionful quantities scale as

$$x \sim k^{-1}, \; T \sim k^{-\theta}, \; Z_k \sim k^{-\eta}, \; \phi_a \sim k^{\frac{1}{2}(d-4+\bar{\eta})}, \quad (15)$$

with $\theta$ and $\bar{\eta}$ related through $\theta = 2 + \eta - \bar{\eta}$. (Note that, formally and as indicated above, the scaling dimension of the temperature is $D_T = -\theta$ and is such that for a fixed bare temperature $T$ the dimensionless renormalized temperature $\widetilde{T}_k = k^\theta T$ indeed goes to zero as $k \to 0$, provided of course $\theta > 0$.) Moreover,

$$U_k \sim k^{d-\theta}, \; V_k \sim k^{d-2\theta}, \quad (16)$$

so that the second cumulant of the renormalized random field $\Delta_k$ scales as $k^{-(2\eta-\bar{\eta})}$. Contrary to Cardy's transformed fields,[49] all the fields $\phi_a$ have the same scaling dimension $D_\phi = (d - 4 + \bar{\eta})/2$, but there is now in addition a renormalized temperature with its own scaling dimension.

Letting the dimensionless counterparts of $U_k, V_k, \Delta_k, \phi$ be denoted by lower-case letters, $u_k, v_k, \delta_k, \varphi$, the resulting FRG flow equations can be symbolically written as

$$\partial_t u_k'(\varphi) = -\frac{1}{2}(d - 2\eta_k + \bar{\eta}_k)u_k'(\varphi) + \frac{1}{2}(d - 4 + \bar{\eta}_k) \times$$
$$\varphi u_k''(\varphi) + \beta_{u'}(\varphi) \tag{17}$$

$$\partial_t z_k(\varphi) = \eta_k z_k(\varphi) + \frac{1}{2}(d - 4 + \bar{\eta}_k)\varphi z_k'(\varphi) + \beta_z(\varphi) \tag{18}$$

and

$$\partial_t \delta_k(\varphi_1, \varphi_2) = (2\eta_k - \bar{\eta}_k)\delta_k(\varphi_1, \varphi_2) + \frac{1}{2}(d - 4 + \bar{\eta}_k) \times$$
$$(\varphi_1 \partial_{\varphi_1} + \varphi_2 \partial_{\varphi_2})\delta_k(\varphi_1, \varphi_2) + \beta_\delta(\varphi_1, \varphi_2) \tag{19}$$

where $t = \log(k/k_{\mathrm{UV}})$ and $k_{\mathrm{UV}}$ is a UV cutoff associated with the microscopic scale of the model. The beta functions themselves depend on $u_k'$, $z_k$, $\delta_k$ and their derivatives, and they depend as well on the dimensionless IR cutoff function. In addition, the running anomalous dimensions $\eta_k$ and $\bar{\eta}_k$ are fixed by the conditions $z_k(0) = \delta_k(0,0) = 1$. All the expressions are given in Ref. [19]. The above flow equations are written for a zero bare temperature. For a nonzero one there are additional terms proportional to $\widetilde{T}_k$ which are however subdominant as they go to zero as $k^\theta$ when approaching the fixed point.[18,19,25] Note finally that the RG is "functional" as its central objects are functions instead of coupling constants and it is "nonperturbative" as the approximation scheme does not rely on an expansion in some small coupling constant or function.

Fixed points are studied by setting the left-hand sides of the dimensionless FRG equations in Eqs. (17-19) to zero and the spectrum of eigenvalues, or equivalently of scaling dimensions, around a given fixed point can be obtained from the linearization of the equations in the vicinity of this fixed point. The zero-temperature fixed point controlling the critical behavior has been determined in previous investigations:[19,26] Above a critical dimension $d_{\mathrm{DR}}$ close to 5.1, there exists a stable cuspless fixed point (stable, except of course for the relevant direction that corresponds to fine-tuning to the critical point). As already discussed, the presence or absence of a cusp now refers to the dependence of $\delta_k(\varphi_1, \varphi_2)$ on the field difference $\varphi_1 - \varphi_2$ (the cusp being a square-root dependence on the variable $(\varphi_1 - \varphi_2)^2$ when $\varphi_1 \to \varphi_2$). This cuspless fixed point entails SUSY and DR.[19,26]

We stress that the dimensions introduced in Eqs. (15,16) have to be chosen such that a fixed point is indeed found for the whole functions, $u_k(\varphi)$, $\delta_k(\varphi_1, \varphi_2)$, etc. Near $d = 6$ this then fixes the canonical dimensions of all the coupling constants obtained by expanding in powers of the fields: See also the discussion in Sec. IV A and Appendix C.

For addressing the presence or absence of a cuspy behavior, it turns out to be convenient to change variable from $\varphi_1$ and $\varphi_2$ to $\varphi = (\varphi_1 + \varphi_2)/2$ and $\delta\varphi = (\varphi_1 - \varphi_2)/2$. The putative cusp is now in the variable $\delta\varphi$. For $d \geq d_{\mathrm{DR}}$, the (critical) cuspless fixed point is characterized in the limit $\delta\varphi \to 0$ by the expansion

$$\delta_*(\varphi, \delta\varphi) = \delta_{*,0}(\varphi) + \frac{1}{2}\delta_{*,2}(\varphi)\delta\varphi^2 + \mathcal{O}(|\delta\varphi|^3). \tag{20}$$

The signature of the Parisi-Sourlas SUSY is a Ward identity relating the second and the first cumulants,[64]

$$\delta_{*,0}(\varphi) = z_*(\varphi) \tag{21}$$

which is satisfied at the cuspless fixed point and implies DR.[18,19,26] Indeed, the sector with $\delta\varphi = 0$ then decouples, and one finds that in this sector the cuspless fixed point in $d$ dimensions is the same as that of the pure $\phi^4$ theory in dimension $d - 2$ obtained from the same approximation (*i.e.*, the second order of the derivative expansion for the effective average action described by 2 functions, $u'(\varphi)$ and $z(\varphi)$).

We can now fix the functions $u_k'(\varphi)$, $z_k(\varphi)$ and $\delta_{k,0}(\varphi)$ at their SUSY/DR cuspless fixed-point expressions and study what happens at and around this fixed point for the terms of the expansion in $\delta\varphi^2$, which are related to the dangerous operators $\mathcal{F}_{2p}$ pointed out by Feldman[50] [see Eq. (2)]. We focus on the potentially most dangerous one, $\mathcal{F}_4$, which corresponds to $\delta_{k,2}(\varphi)$. Here too, the sector in $\delta\varphi^2$ decouples from higher orders in $\delta\varphi$ (provided one assumes a regular enough behavior), and one can derive a closed FRG equation for $\delta_{k,2}(\varphi)$, which is of the form

$$\partial_t \delta_{k,2}(\varphi) = A_*(\varphi)\delta_{k,2}(\varphi)^2 + L_*(\varphi, \partial_\varphi, \partial_\varphi^2)\delta_{k,2}(\varphi) + B_*(\varphi) \tag{22}$$

where $L_*$ is a linear operator, $L_* = C_*(\varphi) + D_*(\varphi)\partial_\varphi + E_*(\varphi)\partial_\varphi^2$, and the functions $A_*$, $B_*$, $C_*$, $D_*$, $E_*$ are obtained from the known SUSY/DR fixed-point functions and anomalous dimensions, with $A_*(\varphi) \neq 0$. (Note that as a result of SUSY, $\bar{\eta}_* = \eta_*$.) The expressions are given in [27].

From Eq. (22) one can (numerically) obtain the fixed point $\delta_{*,2}(\varphi)$ by setting the left-hand side to 0,

$$0 = A_*(\varphi)\delta_{*,2}(\varphi)^2 + L_*(\varphi, \partial_\varphi, \partial_\varphi^2)\delta_{*,2}(\varphi) + B_*(\varphi), \tag{23}$$

and determine the eigenvalues by introducing $\delta_{k,2}(\varphi) = \delta_{*,2}(\varphi) + k^\lambda f_\lambda(\varphi)$ and linearizing the right-hand side in the function $f$. The eigenvalue $\Lambda_2$ which is associated with $(\mathcal{F}_4)_L$ is then the smallest $\lambda$ that satisfies

$$\lambda f_\lambda(\varphi) = 2A_*(\varphi)\delta_{*,2}(\varphi)f_\lambda(\varphi) + L_*(\varphi, \partial_\varphi, \partial_\varphi^2)f_\lambda(\varphi). \tag{24}$$

The key property of the fixed-point equation for $\delta_{*,2}(\varphi)$ is that it is nonlinear in $\delta_{*,2}(\varphi)$ itself, *e.g.*, quadratic at the present level of approximation. On the other hand, the FRG equations for all the higher terms $\delta_{k,2p}(\varphi)$ with $p > 1$ are linear. As argued in detail in the case of the RFO($N$)M near its lower critical dimension (see Sec. III and Appendix B), *the nonlinearity of the equation is what leads to the disappearance of the fixed point $\delta_{*,2}(\varphi)$ when*

*the eigenvalue* $\Lambda_2$ *vanishes.* This is precisely what we find when solving the two above equations, Eqs. (23) and (24). (In practice, from the knowledge of $u'_*(\varphi)$ and $z_*(\varphi)$, which are obtained from two coupled equations, we first solve the equation for $\delta_{*,2}(\varphi)$ and then use the input to solve Eq. (24); all partial differential equations are numerically integrated on a one-dimensional grid by discretizing the field $\varphi$, and the solution can be studied for any value of $d$.[10,27,28]) The stable fixed-point function $\delta_{*,2}(\varphi)$ collapses with another unstable fixed-point function in the critical dimension $d_{\mathrm{DR}} \approx 5.13$, when the eigenvalue $\Lambda_2 = 0$.

For $d < d_{\mathrm{DR}}$ the SUSY/DR fixed point for the RFIM therefore no longer exists (of course, the DR fixed-point functions $u_*(\varphi)$ and $z_*(\varphi)$ are still well defined, but not the fixed-point functions associated with the cumulants of the renormalized disorder for distinct field arguments) and $\Lambda_2$, whose equation involves $\delta_{*,2}(\varphi)$, is no longer defined. A heuristic analytical argument showing that this is the case in the absence of a nongeneric cancellation (either accidental or due to an additional symmetry) is given in Appendix D. Note that if for $d < d_{\mathrm{DR}}$ one solves the FRG equation for $\delta_{k,2}(\varphi)$ in Eq. (22) starting from cuspless initial conditions, one finds that $\delta_{k,2}(\varphi)$ grows and diverges at a *finite* RG scale (which as alluded to in Sec. II B defines a Larkin length, length that diverges as $d \to d_{\mathrm{DR}}^{-}$[10,27]). Accordingly, the running Ward identity associated with the Parisi-Sourlas SUSY ceases to be valid at this finite Larkin scale, even at zero temperature when starting from a SUSY-compatible initial condition, and SUSY is then broken along the FRG flow.

From the above result it is clear that, despite the fact that the operator $(\mathcal{F}_4)_L$ which is characterized by the eigenvalue $\Lambda_2$ is SUSY-null and not SUSY-nonwritable,[1] the vanishing of $\Lambda_2$ leads to a breakdown of both SUSY and DR. This is due to the coincidence of $\Lambda_2 = 0$ with the disappearance of a fixed point for $\delta_2(\varphi)$, complemented by the fact that $\delta_2(\varphi)$ appears in all other fixed-point equations in the sector of the theory where the replica-field arguments of the renormalized cumulants are not equal. Below the dimension $d_{\mathrm{DR}}$ in which $\Lambda_2 = 0$ there is no more *bona fide* SUSY/DR fixed point because the existence of such a fixed point requires that all dimensionless renormalized 1-PI cumulants be defined, which includes $\delta_2(\varphi)$. As already mentioned, this fixed-point disappearance is different from the situation in the model of an elastic interface in a random environment for which the SUSY/DR fixed point is simply the Gaussian one (*i.e.*, with $\delta_2(\varphi) = 0$) which, although unstable, can be continued below $d_{\mathrm{DR}} = 4$.[34,65]

Starting from the FRG flow equations for the higher-order terms $\delta_{k,2p}(\varphi)$ of the expansion of $\delta(\varphi, \delta\varphi)$ in $\delta\varphi$, one can obtain the eigenvalues $\Lambda_p(d)$ introduced above. This requires to first find the fixed-point value $\delta_{*,2p}(\varphi)$ and then to solve the linearized eigenvalue equation around this value. The quality of the result for a given nonperturbative approximation, *e.g.*, DE2, is expected to deteriorate as $p$ increases. We have therefore not com-

puted the eigenvalues beyond $\Lambda_3$ which is associated to the term in $\delta\varphi^4$ in $\delta(\varphi, \delta\varphi)$ and corresponds to Feldman's operator $\mathcal{F}_6$. The family of eigenvalues can also be extended to noninteger values of $p$ and we have calculated the eigenvalue $\Lambda_{3/2}$ which is associated with a cuspy perturbation in $\sqrt{\delta\varphi^2}$ in the second 1-PI cumulant of the renormalized random field.

We plot in Fig. 5 the eigenvalues $\Lambda_2(d)$, $\Lambda_3(d)$, and $\Lambda_{3/2}(d)$, as computed both from the perturbative result of Sec. IV B and from our nonperturbative FRG calculation obtained at DE2. The nonperturbative calculations are detailed above and in Appendix E. The 2-loop perturbative calculation for $\Lambda_2$ coincides with that of [1] and, as explained in Secs. IV A and IV B, the result for $\Lambda_3$ is shifted from that of [1] by $-2$. In the present approximation scheme, $\Lambda_3$ vanishes at a slightly higher $d$ than $\Lambda_2$. As also found for the RFO($N > 2$)M and discussed in Sec. III, this does not break SUSY/DR but leads to the appearance of a subcusp in $\delta(\varphi, \delta\varphi)$.

As already stressed in [10,27,28], $\Lambda_{3/2}(d)$ is irrelevant when $\Lambda_2(d)$ vanishes. Physically, this means that the presence of scale-free avalanches at criticality does induce cusps in the cumulants of the renormalized random field but that the amplitude(s) of these cusps go to zero at the fixed point. The disappearance of the SUSY/DR cuspless fixed point is therefore not due to the effect of the avalanches *per se*. (On the other hand, scale-free avalanches do control the critical behavior described by the cuspy fixed point below $d_{\mathrm{DR}}$.) The SUSY/DR cuspless fixed point disappears for $d < d_{\mathrm{DR}}$, where $d_{DR}$ coincides with the location of $\Lambda_2(d) = 0$.

Note finally that, as discussed in Sec. III C, the fact that $\Lambda_{3/2} > 0$ when $\Lambda_2 = 0$ is in contrast with what is observed for the RFO($N$)M near $d = 4$ (compare with Fig. 2). In the latter case, $d_{\mathrm{cusp}}(N)$, or equivalently $N_{\mathrm{cusp}}(d)$, is given by the location of $\Lambda_{3/2}(d) = 0$ which takes place while the SUSY/DR fixed point becomes unstable but still exists (with $\Lambda_2 > 0$): see the sketch in Fig. 4 for an illustration.

We conclude this section by a few additional comments:

- As already mentioned, our calculation is functional and nonperturbative, and valid for any $d$, but is approximate. It exactly recovers the 1-loop results in $d = 6 - \epsilon$ but is not exact at 2-loop order. We compare the approximate results obtained by fitting the data to a quadratic polynomial in $\epsilon$ near of $d = 6$ (at different orders of the nonperturbative approximation scheme) to the exact perturbative result at 2-loop in Sec. V C below.

- A cruder approximation than the DE2 discussed above is obtained by neglecting the variation on $\varphi$ of $z_k(\varphi)$ and $\delta_k(\varphi, \delta\varphi)$ and fixing $\varphi$ to the value of the minimum of the effective potential $u_k(\varphi)$, *i.e.*, $z_k(\varphi) = z_k(\varphi_{\mathrm{min,k}})$ and $\delta_k(\varphi, \delta\varphi) = \delta_k(\varphi_{\mathrm{min,k}}, \delta\varphi)$

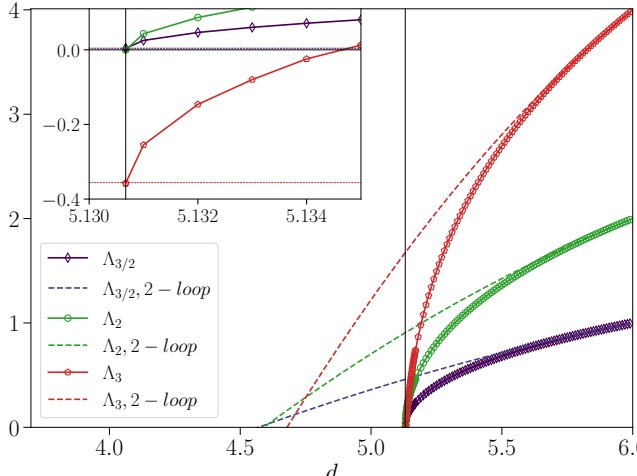

FIG. 5: Eigenvalues $\Lambda_2(d)$, $\Lambda_3(d)$, and $\Lambda_{3/2}(d)$ of the most dangerous perturbations around the (most stable) SUSY/DR fixed point in the RFIM as calculated from the nonperturbative truncation DE2 discussed in Secs. IV C. The dashed lines represent the 2-loop calculation in $d = 6 - \epsilon$, together with plausible extrapolations. Inset: Zoom in on the region around $d_{\mathrm{DR}}$. Note that the cuspy perturbation associated with $\Lambda_{3/2}$ is small but irrelevant when $\Lambda_2$ vanishes in $d_{\mathrm{DR}}$; on the other hand, $\Lambda_3$ vanishes at a slightly higher $d$ than $\Lambda_2$ (as also found in the RFO$(N)$M: see Fig. 2). The dashed horizontal lines indicate the values of the three eigenvalues when $d = d_{\mathrm{DR}}$.

with $u'_k(\varphi_{\mathrm{min,k}}) = 0$. We refer to this approximation as the LPA'. It, too, does not explicitly break SUSY so that the cuspless fixed point exactly satisfies dimensional reduction with respect to its LPA' counterpart in the pure $\phi^4$ theory in $d - 2$. The equations for $\delta_{*,2}$ and $\Lambda_2$ [see Eqs. (23) and (24)] become

$$0 = A_*(\varphi_{\mathrm{min,*}})\delta_{*,2}^2 + C_*(\varphi_{\mathrm{min,*}})\delta_{*,2} + B_*(\varphi_{\mathrm{min,*}}) \quad (25)$$

and

$$\Lambda_2 = 2A_*(\varphi_{\mathrm{min,*}})\delta_{*,2} + C_*(\varphi_{\mathrm{min,*}}) \quad (26)$$

with $A_* > 0$ and $B_* \geq 0$. The situation is now very similar to that encountered in the the FRG of the RFO$(N)$M in $d = 4 + \epsilon$ at 1-loop (see Sec. III and Appendix B), except that all quantities depend on $d$ instead of $N$. One expects a regular behavior with $d$ of the coefficients $A_*$, $B_*$, and $C_*$ in the vicinity of $d_{\mathrm{DR}}$, so that, barring the occurrence of a nongeneric cancellation, e.g., due to an additional symmetry, which makes $B_*$ vanish in $d = d_{\mathrm{DR}}$, there are two solutions for $\delta_{*,2}$ above $d_{\mathrm{DR}}$: one is stable, with an eigenvalue $\Lambda_2 > 0$, and one is unstable, with an eigenvalue $\Lambda'_2 < 0$; the two coalesce in $d_{\mathrm{DR}}$ where $\Lambda_2(d_{\mathrm{DR}}) = \Lambda'_2(d_{\mathrm{DR}}) = 0$, and there are no real solutions for $d < d_{\mathrm{DR}}$. This is indeed what is found numerically and is illustrated in the LPA' result of Fig. 6.

• The validity of the truncation beyond the perturbative regime can be assessed by studying different levels of the nonperturbative approximation scheme. This is what we do and detail in the next section. There, we show that the apparent convergence of the approximation scheme is actually fast.

## V. RFIM: ROBUSTNESS AND ACCURACY OF THE NONPERTURBATIVE FRG APPROXIMATION SCHEME

### A. Goal

Our goal is to check the robustness of the nonperturbative FRG predictions for the explanantion of the SUSY/DR breakdown as a function of space dimension. In particular, we focus on the critical dimension $d_{\mathrm{DR}}$ at which the SUSY/DR fixed point is predicted to disappear and we provide error bars on its value by studying different levels of the nonperturbative approximation scheme.[12,19,26] We also want to assess the results for the eigenvalues around the SUSY/DR cuspless fixed point, especially the eigenvalue $\Lambda_2$ which in the work of KRT,[1,2] is associated to the leader $(\mathcal{F}_4)_L$ of the most dangerous Feldman operator that can destabilize the SUSY/DR fixed point.

To obtain $d_{\mathrm{DR}}$ and compute the eigenvalues we consider the domain of spatial dimension $d$ in which SUSY and DR are valid at the fixed point. The main simplification in the FRG treatment is that the calculation only requires the determination of functions entering the 1-PI cumulants when all replica-field arguments are equal. Working with functions of only one field is then much more tractable than the determination of the full functional dependence which is needed when $d < d_{\mathrm{DR}}$ to capture the cuspy fixed point.[10,12,19,24,25]

In our previous papers,[10,19,26–28] we have studied the second order of the approximation scheme in which we keep the first 1-PI cumulant of at the second order of the derivative expansion, the second 1-PI cumulant at the local potential approximation level, and we neglect higher-order cumulants. This is described in Sec. IV C where it is denoted DE2 approximation.

Two cruder approximations can be considered that both avoid an explicit breaking of SUSY: The simplest is LPA' in which one freezes the dependence of $Z_k(\phi)$ on $\phi$ and that of $\Delta_k(\phi_1, \phi_2)$ on $\phi = (\phi_1 + \phi_2)/2$ (the other independent field variable $\delta\phi = (\phi_1 - \phi_2)/2$ still being free) to the value at the minimum of the effective average potential, $\phi = \phi_{\mathrm{min,k}}$ with $U'_k(\phi_{\mathrm{min,k}}) = 0$ (see also above in Sec. IV C). An improved approximation, which we call LPA", consists in still choosing $Z_k(\phi) = Z_k(\phi_{\mathrm{min,k}})$ and $\Delta_k(\phi, \phi) = \Delta_k(\phi_{\mathrm{min,k}}, \phi_{\mathrm{min,k}})$ but letting both $\phi_1$ and $\phi_2$ unconstrained in $\Delta_k(\phi_1, \phi_2) - \Delta(\phi, \phi)$. More importantly, we also investigate the next level of the nonperturbative approximation scheme beyond DE2, which is what we present below.

## B. Level DE4 of the nonperturbative FRG approximation scheme

We now study the next level of approximation, which we call DE4: we keep the first three 1-PI cumulants, the first one being considered at the 4th order of the derivative expansion (hence the acronym DE4), the second one at the second order of the derivative expansion, and the third one at the LPA, *i.e.*, explicitly,

$$\Gamma_{k1}[\phi_1] =$$
$$\int_x \left[ U_k(\phi_1(x)) + \frac{1}{2} Z_k(\phi_1(x))(\partial_\mu \phi_1(x))^2 + \frac{1}{2} W_{a;k}(\phi_1(x)) \right.$$
$$\times (\partial_\mu \partial_\nu \phi_1(x))^2 + \frac{1}{2} W_{b;k}(\phi_1(x)) \partial_\mu \partial_\nu \phi_1(x) \partial_\mu \phi_1(x) \times$$
$$\left. \partial_\nu \phi_1(x) + \frac{1}{8} W_{c;k}(\phi_1(x))(\partial_\mu \phi_1(x))^2 (\partial_\nu \phi_1(x))^2 \right],$$
$$(27)$$

$$\Gamma_{k2}[\phi_1, \phi_2] =$$
$$\int_x \left[ V_k(\phi_1(x), \phi_2(x)) + X_{a;k}(\phi_1(x), \phi_2(x)) \partial_\mu \phi_1(x) \partial_\mu \phi_2(x) \right.$$
$$+ \frac{1}{2} X_{b;k}(\phi_1(x), \phi_2(x))[(\partial_\mu \phi_1(x))^2 + (\partial_\mu \phi_2(x))^2] +$$
$$\left. \frac{1}{2} X_{c;k}(\phi_1(x), \phi_2(x))[(\partial_\mu \phi_1(x))^2 - (\partial_\mu \phi_2(x))^2] \right],$$
$$(28)$$

$$\Gamma_{k3} = \int_x V_{3k}(\phi_1(x), \phi_2(x), \phi_3(x)), \quad (29)$$

and

$$\Gamma_{kp \geq 4} = 0, \quad (30)$$

where the functions $V_k$, $X_{a,k}$, $X_{b,k}$, and $V_{3k}$ are symmetric in the permutations of the arguments while the function $X_{c,k}$ is antisymmetric. Recall that the effective average potential $U_k(\phi_1)$ describes the thermodynamics of the system and $Z_k(\phi_1)$ is the field-renormalization function. The function $V_k(\phi_1, \phi_2)$ is the 2-replica effective average potential; its second derivative, $V_k^{(11)}(\phi_1, \phi_2) = \Delta_k(\phi_1, \phi_2)$, is the second cumulant of the renormalized random field at zero momentum and is a key quantity that tracks avalanches and droplets through its functional dependence in $(\phi_1 - \phi_2)$. Similarly, $V_{3k}(\phi_1, \phi_2, \phi_3)$ is the 3-replica effective average potential whose third derivative, $V_{3k}^{(111)}(\phi_1, \phi_2, \phi_3) = S_k(\phi_1, \phi_2, \phi_3)$, is the third cumulant of the renormalized random field at zero momentum. The other functions describe the higher-order momentum dependence of the first cumulant and the momentum dependence of the second cumulant.

Inserting the above ansatz into the *exact* FRG equations for the cumulants leads to a set of coupled flow equations for 5 functions of one field $U_k$, $Z_k$, and $W_{a,b,c;k}$, 4 functions of two fields, $V_k$, $X_{a,b,c;k}$, and 1 function of three fields, $V_{3k}$. It turns out to be more convenient to work with the functions $\Delta_k(\phi_1, \phi_2) =$

$V_k^{(11)}(\phi_1, \phi_2)$, $X_{a;k}(\phi_1, \phi_2)$, $X_{b;k}^{(10)}(\phi_1, \phi_2) - X_{c;k}^{(10)}(\phi_1, \phi_2)$, $X_{b;k}^{(11)}(\phi_1, \phi_2) + X_{a;k}^{(11)}(\phi_1, \phi_2)$, and $S_k(\phi_1, \phi_2, \phi_3) = V_{3k}^{(111)}(\phi_1, \phi_2, \phi_3)$, which in conjunction with the 5 functions of one field present in the first cumulant leads to a closed set of FRG equations. Note that we could have alternatively derived these equations by using the FRG in a superfield formalism as we did in [18,19,26]; the advantage of the latter is to make explicit the Ward identities associated with the Parisi-Sourlas SUSY (superrotations) and the mechanism by which they may break and cease to apply. However, once this information is available, it is easier (and fully equivalent) to work with the FRG with conventional replica fields as we do here.

Finding fixed points that describe scale invariance as well as the spectrum of eigenvalues around them requires casting the resulting FRG flow equations in a dimensionless form, as already discussed in Sec. IV C. More details are given in Appendix E. The full-blown numerical resolution of the resulting extensive set of coupled partial differential equations is intractable at the present time. However, for determining the value of the critical dimension $d_{DR}$ at which DR is broken (and below which SUSY is broken along the FRG flow) and to find the spectrum of eigenvalues around the SUSY/DR fixed point, we can restrict ourselves to considering the SUSY/DR fixed point and its vicinity when $d \geq d_{DR}$, with $d_{DR}$ yet to be determined.

We have previously shown that the appearance of cusps in the functional dependence of the cumulants of the renormalized random field entails the breakdown of SUSY and DR. The cusps appear in the difference between replica fields so that it is convenient to introduce a linear change of field arguments: in the second (dimensionless) cumulant, $\varphi = (\varphi_1 + \varphi_2)/2$ and $\delta\varphi = (\varphi_1 - \varphi_2)/2$ (see Sec. IV C), and in the third (dimensionless) cumulant,

$$\varphi = \frac{\varphi_1 + \varphi_2 + \varphi_3}{3},$$
$$y = \frac{\varphi_1 - \varphi_2}{\sqrt{2}}, \quad (31)$$
$$z = \frac{\varphi_1 + \varphi_2 - 2\varphi_3}{\sqrt{6}}.$$

An important property of the hierarchy of flow equations for the cumulants of the renormalized random field ($\Gamma_{k2}^{(11)}[\phi_1, \phi_2]$, $\Gamma_{k3}^{(111)}[\phi_1, \phi_2, \phi_3]$, etc., and their dimensionless counterparts) when expanded in the differences between replica fields is that the sector of equal fields, $\varphi_1 = \varphi_2 = \varphi_3 = \varphi$, $\delta\varphi = y = z = 0$, decouples from the sector with nonzero field differences, provided that the functional dependence of the cumulants on the latter is regular enough (no cusps). In addition, all functions in the sector of equal replica fields can then be related to the functions of the first cumulant as a result of the SUSY Ward identities.[19] As already stressed, the latter are indeed preserved by the present truncation scheme with a choice of IR cutoff functions that satisfy their own SUSY

Ward identity. In particular, we obtain that if SUSY is valid at the scale $k$,

$$\delta_k(\varphi, \varphi) = z_k(\varphi)$$
$$x_{a;k}(\varphi, \varphi) = 2w_{a;k}(\varphi)$$
$$x_{b;k}^{(10)}(\varphi, \varphi) - x_{c;k}^{(10)}(\varphi, \varphi) = w_{b;k}(\varphi) - w'_{a;k}(\varphi)$$
$$x_{b;k}^{(11)}(\varphi, \varphi) + x_{a;k}^{(11)}(\varphi, \varphi) = \frac{1}{2}w_{c;k}(\varphi) \qquad (32)$$
$$s_k(\varphi, \varphi, \varphi) = \frac{3}{2}[w_{b;k}(\varphi) - w'_{a;k}(\varphi)].$$

With the assumption of a regular enough field dependence, the dimensionless functions can be expanded in the differences between replica fields. For the second cumulant, taking into account the symmetries (in particular, $\delta\varphi \to -\delta\varphi$ at constant $\varphi$) and the above SUSY related identities, one has

$$\delta_k(\varphi + \delta\varphi, \varphi - \delta\varphi) = z_k(\varphi) + \frac{1}{2}\delta_{k,2}(\varphi)\delta\varphi^2 + \cdots,$$

$$x_{a;k}(\varphi + \delta\varphi, \varphi - \delta\varphi) = 2w_{a;k}(\varphi) + \frac{1}{2}x_{a;k,2}(\varphi)\delta\varphi^2 + \cdots,$$

$$\frac{1}{2}[x_{b;k}^{(10)}(\varphi + \delta\varphi, \varphi - \delta\varphi) + x_{b;k}^{(01)}(\varphi + \delta\varphi, \varphi - \delta\varphi)] -$$
$$\frac{1}{2}[x_{c;k}^{(10)}(\varphi + \delta\varphi, \varphi - \delta\varphi) - x_{c;k}^{(01)}(\varphi + \delta\varphi, \varphi - \delta\varphi)] =$$
$$w_{b;k}(\varphi) - w'_{a;k}(\varphi) + \frac{1}{2}x_{e;k,2}(\varphi)\delta\varphi^2 + \cdots,$$

$$(33)$$

and

$$x_{b;k}^{(11)}(\varphi + \delta\varphi, \varphi - \delta\varphi) + x_{a;k}^{(11)}(\varphi + \delta\varphi, \varphi - \delta\varphi) =$$
$$\frac{1}{2}w_{c;k}(\varphi) + \frac{1}{2}x_{f;k,2}(\varphi)\delta\varphi^2 + \cdots, \qquad (34)$$

when $\delta\varphi \to 0$, where all the functions of $\varphi$ in the right-hand sides are even. For the third cumulant of the random field, after taking into account the symmetries (in particular, $y \to -y/2 - \sqrt{3}z/2, z \to \sqrt{3}y/2 - z/2$ at constant $\varphi$, $y \to -y$ at constant $\varphi, z$, and the property that $s_k$ is odd under the global inversion $\varphi \to -\varphi, y \to -y, z \to -z$), one finds

$$s_k(\varphi + y/\sqrt{2} + z/\sqrt{6}, \varphi - y/\sqrt{2} + z/\sqrt{6}, \varphi - 2z/\sqrt{6}) =$$
$$\frac{3}{2}[w_{b;k}(\varphi) - w'_{a;k}(\varphi)] + \frac{1}{2}s_{k,2}(\varphi)(y^2 + z^2) +$$
$$\frac{1}{3!}s_{k,3}(\varphi)(z^3 - 3y^2z) + \cdots, \qquad (35)$$

when $y, z \to 0$, where $s_{k,2}$ is an odd function of $\varphi$ and $s_{k,3}(\varphi)$ is even.

The structure of the FRG equations for the cumulants of the random field is such that the sector which is quadratic in the field differences, *i.e.*, involving the functions $\delta_{k,2}(\varphi)$, $x_{a,e,f;k,2}(\varphi)$, and $s_{k,2}(x = \sqrt{3}\varphi)$, when complemented with the cubic term $s_{k,3}(\varphi)$, also decouples from higher orders in field differences. This

triangular-like structure of the system of equations likely carries over to all orders of the approximation scheme.

The procedure is then to fix all functions in the sector of equal fields to their fixed-point values. Through the SUSY Ward identities these values can all be expressed in terms of the 5 functions, $u_*(\varphi), z_*(\varphi), w_{a,b,c;*}(\varphi)$, which, through the ensuing DR, coincide with those computed at the DE4 for the pure $\phi^4$ theory in dimension $d-2$. We next consider the set of coupled flow equations for the 6 functions $\delta_{k,2}(\varphi), x_{a,e,f;k,2}(\varphi), s_{k,2}(\varphi)$, and $s_{k,3}(\varphi)$:

$$\partial_t\delta_{k,2}(\varphi) = (d - 4 + 2\eta_*)\delta_{k,2}(\varphi)$$
$$+ \frac{1}{2}(d - 4 + \eta_*)\varphi\delta'_{k,2}(\varphi) + \beta_{\delta_2}(\varphi), \qquad (36)$$

$$\partial_t x_{a;k,2}(\varphi) = (d - 2 + 2\eta_*)x_{a;k,2}(\varphi)$$
$$+ \frac{1}{2}(d - 4 + \eta_*)\varphi x'_{a;k,2}(\varphi) + \beta_{x_{a;2}}(\varphi), \qquad (37)$$

$$\partial_t x_{e;k,2}(\varphi) = \frac{1}{2}(3d - 8 + 5\eta_*)x_{e;k,2}(\varphi)$$
$$+ \frac{1}{2}(d - 4 + \eta_*)\varphi x'_{e;k,2}(\varphi) + \beta_{x_{e;2}}(\varphi), \qquad (38)$$

$$\partial_t x_{f;k,2}(\varphi) = (2d - 6 + 3\eta_*)x_{f;k,2}(\varphi)$$
$$+ \frac{1}{2}(d - 4 + \eta_*)\varphi x'_{f;k,2}(\varphi) + \beta_{x_{f;3}}(\varphi), \qquad (39)$$

$$\partial_t s_{k,2}(\varphi) = \frac{1}{2}(3d - 8 + 5\eta_*)s_{k,2}(\varphi)$$
$$+ \frac{1}{2}(d - 4 + \eta_*)\varphi s'_{k,2}(\varphi) + \beta_{s_2}(\varphi), \qquad (40)$$

$$\partial_t s_{k,3}(\varphi) = (2d - 6 + 3\eta_*)s_{k,3}(\varphi)$$
$$+ \frac{1}{2}(d - 4 + \eta_*)\varphi s'_{k,3}(\varphi) + \beta_{s_3}(\varphi), \qquad (41)$$

where the beta functions themselves depend on $\delta_{k,2}(\varphi)$, $x_{a,e,f;k,2}(\varphi)$, $s_{k,2}(\varphi)$, $s_{k,3}(\varphi)$, their derivatives, and on the DR fixed-point functions. Through this dependence, the above equations are coupled nonlinear second-order partial differential equations with a nonlinearity that can be up to cubic in the functions. The equations given above correspond to a zero bare temperature for which the subdominant terms in $O(\widetilde{T}_k)$ are absent. (The expressions for the beta functions are too long to be shown here but they can be systematically and straightforwardly derived with the help of Mathematica.)

As done at the DE2 level (see Sec. IV C), we consider both the fixed-point equations obtained by setting the left-hand sides of the FRG flow equations to zero and the eigenvalue equations obtained by linearizing the flow equations. Again, we stress that the validity of the scaling dimensions used to cast the FRG equations in

a dimensionless form is guaranteed by the fact that a *bona fide* fixed point can actually be found with, in particular, fixed-point solutions for the functions $\delta_{k,2}(\varphi)$, $x_{a;k,2}(\varphi)$,..., $s_{k,3}(\varphi)$. Within the spectrum of eigenvalues, $\Lambda_2$ should be the one that starts in $2 - \epsilon$ around the upper critical dimension and $\Lambda_3$ that starting in $4 - 2\epsilon$: see Sec. IV B. We have also computed the eigenvalue $\Lambda_{3/2}$ associated with cuspy perturbations in the second and the third 1-PI cumulants of the renormalized random field.

Finally, as in [10,27] the critical dimension $d_{\mathrm{DR}}$ is determined by looking at the vanishing of the eigenvalue $\Lambda_2(d)$. It vanishes with a square-root behavior and collapses with the eigenvalue found for a SUSY/DR unstable fixed point. This allows a crisp determination of $d_{\mathrm{DR}}$: see Fig. 6. Alternatively, $d_{\mathrm{DR}}$ can be located as the dimension below which, *e.g.*, $\delta_{k,2}(\varphi)$ diverges in a finite RG time. As this RG time, which is associated with the Larkin length discussed in Sec. IV C, diverges when $d \to d_{\mathrm{DR}}-$, this procedure is however less accurate than that using the vanishing of $\Lambda_2(d)$.

A heuristic argument extending to the present DE4 approximation why the cuspless SUSY/DR fixed point disappears below the dimension at which $\Lambda_2$ vanishes as a result of the nonlinearity of the flow equations in Eqs. (36-41) is given in Appendix F. We stress again that the DR fixed point restricted to the 1-replica (first cumulant) sector is of course always present because it corresponds to the Wilson-Fisher fixed point in dimension $d - 2$ and can be continued below $d_{\mathrm{DR}}$. However, for the RFIM at criticality, all the cumulants with their full functional dependence must reach a fixed point. The global RFIM SUSY/DR fixed point then disappears below $d_{\mathrm{DR}}$ because it no longer exists in the sector associated with the cumulants of order 2 and higher and for field arguments that do not coincide. As already discussed, this property is missed by the conventional perturbation approach of KRT.[1,2]

### C. Results

We combine the results already obtained at the DE2 level with those that we have presently calculated for the cruder approximation LPA' discussed in Sec. IV C and for the improved truncation DE4 introduced above. We also consider the improved LPA' approximation, which we have called LPA" (see Sec. V A), where we recall that $z_k(\varphi)$ and $\delta_k(\varphi, \varphi)$ are fixed at their value in the running minimum of the potential, $\varphi_{\mathrm{min},k}$, but the $\varphi$-dependence of $\delta(\varphi + \delta\varphi, \varphi - \delta\varphi) - \delta_k(\varphi, \varphi)$ is not constrained. So defined, this LPA" does not explicitly break SUSY so that, as the other approximation levels, it leads to DR so long as the cuspless fixed point exists. We have already shown the data for the eigenvalue $\Lambda_2(d)$ corresponding to all the studied levels of the approximation scheme in Fig. 1(a).

As explained before, we determine the critical dimen-

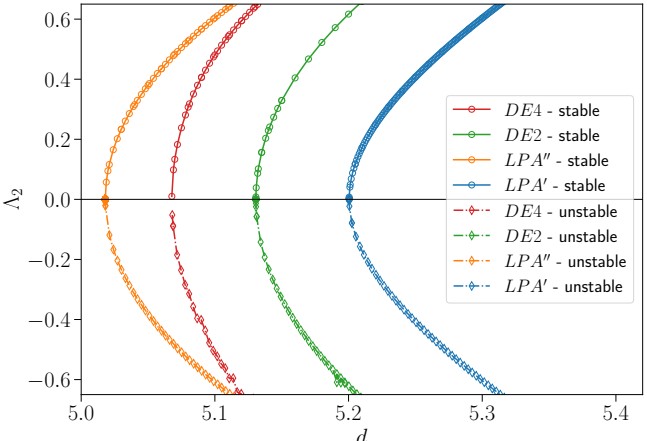

FIG. 6: Determination of the critical dimension $d_{\mathrm{DR}}$ in the RFIM from the vanishing of the eigenvalue $\Lambda_2(d)$ associated with the operator $\mathcal{F}_4$ for the successive nonperturbative FRG approximations LPA', LPA", DE2, and DE4. The (positive) eigenvalue $\Lambda_2(d)$ vanishes as a square-root and collapses with the (negative) eigenvalue $\Lambda_2'(d)$ associated with an unstable SUSY/DR fixed point. Below $d_{\mathrm{DR}}$, there are no SUSY/DR fixed points. Note that as shown in Fig. 1 all the curves for $\Lambda_2$ converge to the same value with the same slope when $d \to 6$.

sion $d_{\mathrm{DR}}$ by looking at the location where the most dangerous eigenvalue $\Lambda_2(d)$ vanishes. The latter does so with a square-root singularity which is associated with the collapse of the stable SUSY/DR fixed point with another unstable SUSY/DR fixed point (see above) which we have been able to track in the vicinity of $d_{\mathrm{DR}}$. The results are displayed in Fig. 6. This allows us to extract the value of $d_{\mathrm{DR}}$. We find $d_{\mathrm{DR}} \approx 5.2005$ for the lowest order approximation LPA', $d_{\mathrm{DR}} \approx 5.0180$ for the LPA", and $d_{\mathrm{DR}} \approx 5.0678$ for the highest order DE4.

For the previously studied DE2 level of the approximation scheme,[10] we obtain $d_{\mathrm{DR}} \approx 5.1307$ through the same procedure. We can therefore conclude that the results are robust with respect to the approximation order. We are also able to provide an estimate of $d_{\mathrm{DR}}$ with, for the first time, an error bar accounting for the 4 levels of approximation:

$$d_{\mathrm{DR}} \approx 5.11 \pm 0.09. \tag{42}$$

In addition, it should be noted that the values obtained by increasing the order of the approximation scheme appear *to oscillate around* 5.11.[66]

Note that the above values obtained for $d_{\mathrm{DR}}$ at the different levels of the nonperturbative approximation scheme are determined with a high precision (at least 5 digits) as the location where the eigenvalue $\Lambda_2$ vanishes: see Fig. 6. On the other hand, the solution of the FRG equations at each level of approximation somewhat depends on the detailed form of the (dimensionless) regulator functions that are introduced to implement the IR cutoff on the functional RG flows: see Sec. IV C and Refs. [60,61]. In all our calculations we have used an ex-

TABLE I: $\epsilon^2$ coefficient of the eigenvalues $\Lambda_2$, $\Lambda_3$ and $\Lambda_{3/2}$ from successive orders of the nonperturbative FRG approximation scheme, together with the exact 2-loop result.

| Order | $\Lambda_2$ | $\Lambda_3$ | $\Lambda_{3/2}$ |
|-------|-------------|-------------|-----------------|
| LPA'  | $-1.03 \pm 0.07$ | | |
| LPA"  | $-0.26 \pm 0.01$ | $-0.69 \pm 0.09$ | |
| DE2   | $-0.40 \pm 0.02$ | $-0.99 \pm 0.03$ | $-0.19 \pm 0.01$ |
| DE4   | $-0.25 \pm 0.02$ | $-0.65 \pm 0.02$ | $-0.10 \pm 0.01$ |
| Exact | $-8/27 \approx 0.30$ | $-7/9 \approx 0.78$ | $-5/36 \approx 0.14$ |

ponential regulator with a prefactor that is optimized in a dimension near but strictly above $d_{\mathrm{DR}}$ (separately at each approximation level) according to the principle of minimum sensitivity.[61] From the detailed work of [67] we expect that the effect of the regulator at each approximation level is within the global error bar obtained from comparing different levels [given in Eq. (42)].

As already mentioned, the present calculations are nonperturbative but approximate. The robustness and apparent convergence of the predictions of the successive orders of the approximation scheme is a strong support for the approach. We have also compared the results obtained in the vicinity of $d = 6$ to the exact 2-loop calculation. We have fitted our numerical data for $\Lambda_2$, $\Lambda_3$, and $\Lambda_{3/2}$ to a quadratic polynomial in $\epsilon = 6 - d$ near $d = 6$:

$$\Lambda_2(d) = 2 - \epsilon - a_2\epsilon^2 + \mathrm{O}(\epsilon^3)$$
$$\Lambda_3(d) = 4 - 2\epsilon - a_3\epsilon^2 + \mathrm{O}(\epsilon^3) \qquad (43)$$
$$\Lambda_{3/2}(d) = 1 - \epsilon/2 - a_{3/2}\epsilon^2 + \mathrm{O}(\epsilon^3),$$

where the $\epsilon^2$ coefficients $a_2$, $a_3$ and $a_{3/2}$ are given in Table I with error bars due to the fitting procedure. We find that the numerical values approach the exact ones as the approximation order increases and that they oscillate around the latter (as for the value of the critical dimension $d_{\mathrm{DR}}$). The relative error at DE4 is less than 20% for $\Lambda_2$ and $\Lambda_3$ and about 30% for $\Lambda_{3/2}$.

## VI. CONCLUSION

By first revisiting the perturbative FRG results for the RFO($N$)M in $d = 4+\epsilon$ and then carrying out a more comprehensive investigation of the nonperturbative approximation scheme to the FRG of the RFIM, we have put the perturbative results recently derived by KRT,[1,2] which involve a comprehensive and more rigorous development of Feldman's ideas[50] recast within Cardy's parametrization of the RFIM field theory,[16] in light of our 20-year-old FRG description of the breakdown of SUSY and dimensional reduction (DR) in random-field systems.[12,24]

There are two main differences with the treatment of KRT which illustrate the power of the nonperturbative

FRG. First, the latter is able to describe what happens when SUSY and DR are broken. It indeed predicts a non-SUSY fixed point at which the 1-PI cumulants of the renormalized random field display a nonanalytical ("cuspy") dependence on their field arguments and it provides a physical picture emphasizing the role of scale-free collective phenomena that appear in the form of avalanches (at zero temperature) and droplets (at nonzero temperature) at criticality. All of this is well supported by state-of-the-art computer simulations. Second, the nonperturbative calculations show that in the critical dimension where the eigenvalue associated with the (Feldman) operator which is most dangerous for destabilizing the SUSY/DR fixed point vanishes, there is an annihilation of fixed points that leads to the disappearance of the SUSY/DR fixed point below this dimension. This disappearance stems from the nonlinear nature of the associated fixed-point equation and is not captured through the perturbative RG and the $\epsilon = 6 - d$ expansion.

As the critical change of behavior is predicted to take place near $d_{\mathrm{DR}} \approx 5.1$, what are the observable consequences of the different scenarios beyond a general compatibility with the main critical behavior obtained in simulation results in $d = 4, 5, 6$?

Within the FRG approach of the RFIM we predict that scale-free avalanches are present but have a subdominant effect for $d \geq d_{\mathrm{DR}}$ and become central to the critical behavior for $d < d_{\mathrm{DR}}$. A physical argument relies on comparing the fractal dimension $d_f$ of the largest system-spanning avalanches at criticality (in a large but finite system) and the scaling dimension of the spontaneous magnetization (times the volume of the system), i.e., $d - (d - 4 + \bar{\eta})/2 = (d + 4 - \bar{\eta})/2$.[28] Our prediction, which is substantiated by the nonperturbative FRG calculations,[10,27,28,47] is that $(d + 4 - \bar{\eta})/2 - d_f = 0$ when $d < d_{DR}$ and $(d + 4 - \bar{\eta})/2 - d_f > 0$, which explains the subdominant effect of the avalanches, when $d > d_{DR}$. The difference between $(d + 4 - \bar{\eta})/2$ and $d_f$ in the latter case is due to the fact that the number of system-spanning critical avalanches scales with the system size as $L^\lambda$ with $\lambda = (d + 4 - \bar{\eta})/2 - d_f$.[28] The exponent $\lambda$ precisely coincides with the eigenvalue associated with a cuspy perturbation around the SUSY fixed point, $\lambda \equiv \Lambda_{3/2}$.[10,27,28,39,47] From the solution of the mean-field RFIM, one finds that $d_f = 4$ and $\lambda = 1$,[12,28] and one expects that the same values are derived from the field-theoretical version of the RFIM at the upper critical dimension $d_{\mathrm{uc}} = 6$. Our suggestion is then to carry out a simulation of the ground state of the RFIM in $d = 5$ and $d = 6$ in the presence of an applied field, as in [29–31], and measure the statistics of the system-spanning avalanches near criticality. Doing this, one has access to the fractal dimension $d_f$ and, with more difficulty, to the exponent $\lambda = \Lambda_{3/2}$ characterizing the number of these avalanches. This type of determination has been for instance attempted for the athermally driven RFIM near

its (out-of-equilibrium) critical point.[68]

On the other hand, KRT have suggested that the SUSY fixed point could be reached below the dimension where it becomes unstable, say, in $d = 4$, by fine-tuning the distribution of the random field.[1,2] In contrast, we have shown that, as a consequence of the nonlinear nature of the fixed-point equation associated with the dangerous operator $\mathcal{F}_4$, the SUSY/DR fixed point disappears exactly when it becomes unstable. No SUSY/DR fixed point should therefore be found in $d = 4$ (nor in $d = 5$ but this is too close to $d_{\mathrm{DR}}$ to bring any decisive conclusion.[10]) This issue is nonetheless hard to settle in computer simulations due to the finite-size effects. Indeed, for $d < d_{\mathrm{DR}} \approx 5.1$, we find that, even if one starts the FRG flow with initial conditions that are compatible with a SUSY/DR fixed point (in a restricted sector of the theory), conditions that involve cuspless cumulants of the random field, a cusp must appear along the flow at a scale which by analogy with random manifolds in a disordered environment we associated with a "Larkin length".[38] This length diverges rapidly as $d \to d_{\mathrm{DR}}^{-}$ [10,27] but is finite in $d = 4$. It is clear that by fiddling with the initial distribution of the quenched disorder one can vary the Larkin length and make it larger, which would allow the RG flow to first describe a behavior resembling that predicted by DR while eventually evolving toward the proper cuspy (SUSY- and DR-broken) fixed point. The system sizes required for reaching the asymptotic critical behavior may however be out of reach of present-day simulations. (A different issue is whether there exists below $d_{\mathrm{DR}}$ another, unstable, "cuspy" fixed point, at which both SUSY and DR break down, on top of the one that we have already found numerically and that describes the critical behavior of the RFIM; we have not carried out extensive computations to search for it and we therefore cannot a priori exclude its presence.)

Finally, as we have previously advocated,[27,47,69,70] it would be interesting to study by computer simulation the long-range RFIM because one may have a direct access to the critical change of behavior between SUSY/DR and non SUSY/DR fixed points in $d = 3$ [27,69] or between cuspless and cuspy fixed points in $d = 1$ [70] by continuously varying the range of the interactions and of the bare random-field correlations. The disappearance or not of the SUSY/DR or cuspless fixed point when it is predicted to become unstable could then be more crisply probed.

**Acknowledgments**

We thank A. Kaviraj, S. Rychkov, and E. Trevisani for useful exchanges and discussions. IB acknowledges the support of the Croatian Scientific Foundation grant HRZZ-IP-2022-9423.

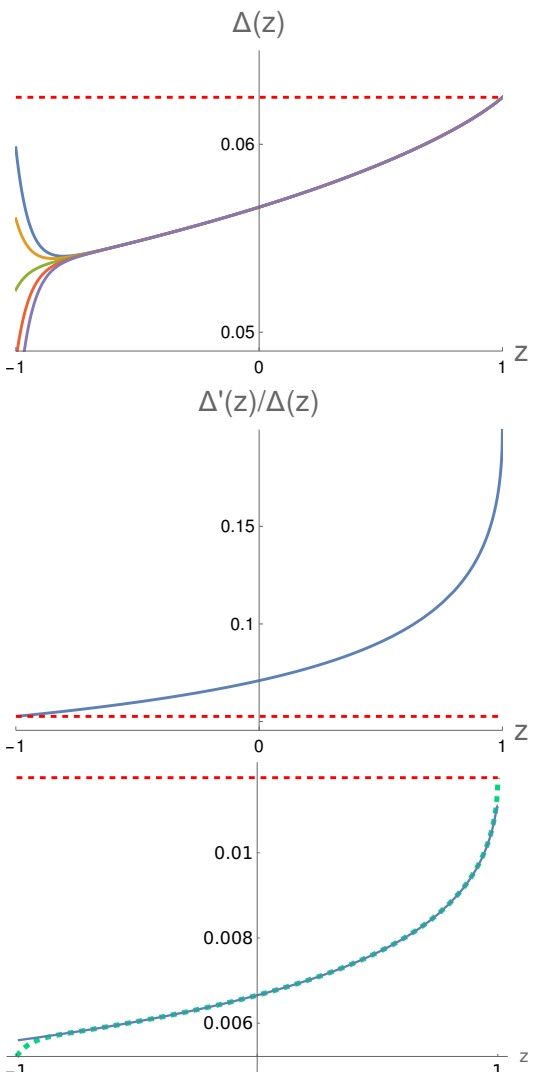

FIG. 7: Evidence for a subcusp in the SUSY/DR fixed point $\Delta_*(z)$ of the RFO($N$)M at one loop for $N = 18$. Top: Function $\Delta(z)$ obtained from a Taylor expansion up to $(1-z)^n$ for $n$ between 30 and 40 by steps of 2 (from bottom to top near $z = -1$). The value $f(1) = 0.011763907083777499$ is optimized for the best apparent convergence near $z = -1$ for $n = 34$. Middle: $\Delta'_*(z)/\Delta_*(z)$ over the interval $[-1, 1]$ as obtained from the expansion up to $n = 40$ with the same value of $f(1)$ as in (a). The horizontal line is the exact value of the ratio $\Delta'_*(-1)/\Delta_*(-1) = 1/19$. The curve appears to converge to the exact value but deviates and sharply drops as one approaches $z = -1$ because the fine-tuning of $f(1)$ is not precise enough at the level of the 15th digit. Bottom: $(\Delta_*(z) - [\frac{1}{16} - \frac{1}{80}(1-z)])/(1-z)^{3/2}$ versus $\sqrt{1-z}$ over the whole interval $-1 \le z \le 1$. The (red) horizontal dashed line is equal to $f(1) = 0.01176\cdots$, which confirms the presence of a subcusp. The dashed green curve is the result of the expansion (up to order 40) about $z = 1$: It coincides with the numerical solution (full line) except when approaching $z = -1$.

## Appendix A: Illustration of the presence of a subcusp at the SUSY/DR fixed point of the RFO($N$)M at one loop

The simplest illustration is to look at the SUSY/DR fixed point in $N = 18$. There, the expected subcusp in the cumulant of the renormalized random field $\Delta_*(z) = R'_*(z)$ is in $(1-z)^{3/2}$ when $z \to 1$ (as $\Lambda_{5/2} = 0$). We work at 1-loop order and we set $\epsilon \equiv 1$. In $N = 18$ the two SUSY/DR fixed points coincide ($\Lambda_2 = 0$) and are unstable with respect to a cuspy perturbation ($\Lambda_{3/2} = -1/10$). However, this does not prevent us from finding the fixed point as we fix $\Delta_k(1)$ and $\Delta'_k(1)$ to their known DR fixed-point values, $i.e.$, for $N = 18$ and $\epsilon = 1$, $\Delta_k(1) = 1/16$ and $\Delta'_k(1) = 1/80$. We therefore look for a solution of the form

$$\Delta_*(z) = \frac{1}{16} - \frac{1}{80}(1-z) + (1-z)^{3/2}f(z) + (1-z)^2 g(z) \tag{A1}$$

where $f(z)$ and $g(z)$ are regular function of $z$ in the vicinity of $z = 1$ [they have a Taylor expansion in powers of $(1-z)$]. The value of $f(1)$ is not fixed and should be determined by requiring that the functions $f(z)$ and $g(z)$ are regular over the whole interval of definition $[-1,1]$; in particular, the functions should be finite in $z = -1$. As we will see this requirement indeed selects a unique solution.

We study the fixed-point equation in two stages:

First, we solve the fixed-point equations for $f(z)$ and $g(z)$ in an expansion around $z = 1$ to rather high orders of $n$ up to 40. Solving the set of equations at order $n$ gives expressions of all derivatives up to $f^{(n+1)}(1)$ and $g^{(n)}(1)$. They are polynomials of $f(1)$ which is the only unknown. The polynomial for $f^{(n+1)}$ is of degree $2(n+1)+1$ and that for $g^{(n)}(1)$ of degree $2(n+1)$. We observe that to keep the polynomials "small enough" so that the expansions of $f(z)$ and $g(z)$ in powers of $(1-z)$ have a chance to converge over a large enough interval one needs to choose the value of $f(1)$ in a very narrow range that drastically shrinks as $n$ increases. For $30 \leq n \leq 40$ we find that the expansions have apparently converged to unique curves for $z \gtrsim -0.8$ but the last segment down to $z = -1$ is sensitive to the 15th significant digit of $f(1) = 0.01176390708377\cdots$, as illustrated in Fig. 7(a). To improve the results one can also include information coming from the exact behavior of $\Delta_*(z)$ near $z = -1$. The value $\Delta_*(-1)$ itself is not analytically known but one for instance finds that $\Delta'_*(-1)/\Delta_*(-1) = 1/19$. Fig. 7(b) shows how the approximation of $\Delta'_*(z)/\Delta_*(z)$ for $n = 40$ behaves as one approaches $z = -1$ and finally deviates from the exact value. To do better one would need to fine-tune the value of $f(1)$ even more precisely.

Next, we numerically solve the fixed-point equation for $\Delta_*(z)$ which is a second-order differential equation. The resolution is performed with Mathematica with a working precision of 30 digits. For boundary values, it is better not to use conditions in $z = 1$ which have a peculiar character, and we instead consider conditions in

$z = 0$. We choose $\Delta_*(0)$ and $\Delta'_*(0)$ within $10^{-7} - 10^{-9}$ of the values obtained through the previous procedure ($z = 0$ is well within the region where the expansion has apparently converged) and we further fine-tune them so that the function $\Delta_*(z)$ is well-behaved down to $z = -1$. The outcome has a clear subcusp in $(1-z)^{3/2}$, as shown in Fig. 7(c).

## Appendix B: Disappearance of the SUSY/DR fixed point in the RFO($N$)M

We consider the mechanism by which the SUSY/DR fixed point in the RFO($N$)M near $d = 4$ disappears (or not) at any given loop order at the value $N_{\mathrm{DR}}$ for which the eigenvalue $\Lambda_2$ associated with Feldman's operator $\mathcal{F}_4$ vanishes. (At 2 loops in $d = 4+\epsilon$, $N_{\mathrm{DR}} = 18 - (49/5)\epsilon$.[46]) We thereby complement the discussion of Sec. III A below Eq. (6).

We simplify the notations by replacing $R''(1)$ by $X$ and the polynomial $Q_{N,k}$ by $Q_N$. The fixed-point value $X_*(N)$ is then solution of $Q_N(X_*(N)) = 0$ and the eigenvalue controlling the stability is $\Lambda_2(N) = Q'_N(X_*(N))$. We are interested by the behavior in the vicinity of $N_{\mathrm{DR}}$, defined by $\Lambda_2(N_{\mathrm{DR}}) = 0$, when $\delta N = N - N_{\mathrm{DR}} \to 0$.

The coefficients of the polynomial $Q_N(X)$, whose degree depends on the number of loops in the expansion in $\epsilon = 4 - d$, are expected to be regular in $\delta N$ around $\delta N = 0$ (this is for instance verified at 1- and 2-loop levels), with

$$Q_N(X) = Q_{N_{\mathrm{DR}}}(X) + R_{N_{\mathrm{DR}}}(X)\delta N + \mathrm{O}(\delta N^2), \tag{B1}$$

Furthermore, $Q_N(X)$ can be Taylor expanded as well around $X_{*0} = X_*(N_{\mathrm{DR}})$,

$$Q_N(X) = Q_N(X_{*0}) + Q'_N(X_{*0})(X - X_{*0}) + \frac{1}{2}Q''_N(X_{*0})(X - X_{*0})^2 + \mathrm{O}((X - X_{*0})^3). \tag{B2}$$

When $\delta N \to 0$, after defining $\delta X_* = X_*(N) - X_{*0}$ and using that by construction $Q_{N_{\mathrm{DR}}}(X_{*0}) = Q'_{N_{\mathrm{DR}}}(X_{*0}) = 0$, we find from the two above expansions that

$$Q_N(X_*(N)) = 0 = R_{N_{\mathrm{DR}}}(X_{*0})\delta N + R'_{N_{\mathrm{DR}}}(X_{*0})\delta N \delta X_* + \frac{1}{2}Q''_{N_{\mathrm{DR}}}(X_{*0})\delta X_*^2 + \mathrm{O}(\delta N^2, \delta N \delta X_*^2). \tag{B3}$$

For a given (small) $\delta N$ the above equation has two real solutions $\delta X_*(\delta N)$ provided the discriminant

$$D = R'_{N_{\mathrm{DR}}}(X_{*0})^2 \delta N^2 - 2R_{N_{\mathrm{DR}}}(X_{*0})Q''_{N_{\mathrm{DR}}}(X_{*0})\delta N \tag{B4}$$

is positive. The two solutions merge and $D = 0$ when $\delta N = 0$. When $\delta N \to 0$ the sign of $D$ is generically given by the second term of the right-hand side which then changes sign when $\delta N$ changes sign. This implies that the 2 real solutions of Eq. (B3) annihilate in

$N = N_{\mathrm{DR}}$ and that there are no real solutions for $N < N_{\mathrm{DR}}$. For this not to happen, one must have $R_{N_{\mathrm{DR}}}(X_{*0})Q''_{N_{\mathrm{DR}}}(X_{*0}) = 0$, and this may then correspond to a crossing of real solutions (depending on the higher-order terms in $\delta N$ and $\delta X_*$) and an exchange of stability of the associated fixed points. The condition, however, has no reason to be satisfied in the absence of an additional symmetry. *Mutatis mutandis* (replacing $N$ by $d$), such a symmetry exists for instance at the SUSY/DR Gaussian fixed point of an elastic interface in a random environment (which then corresponds to $X_{*0} = 0$). This fixed point then crosses with another fixed point at the upper critical dimension $d = 4$ but is still present albeit unstable below. However, this nongeneric phenomenon is absent in the RFO($N$)M in $d = 4 + \epsilon$. One can easily check that $R_{N_{\mathrm{DR}}}(X_{*0})Q''_{N_{\mathrm{DR}}}(X_{*0}) \neq 0$ at both 1-loop and 2-loop orders.

## Appendix C: Multi-copy formalism and relation to the operator classification in [1]

### 1. Averaging over disorder

There are several ways to handle the quenched disorder in the RFIM in order to generate an effective disorder-averaged theory. One is the conventional replica formalism in which one introduces $n$ replicas of the original system, averages over the disorder, and takes the limit $n \to 0$. Another one is the Parisi-Sourlas SUSY method that starts from the functional minimization equation describing the ground state of the system at zero temperature. (Both methods have their limitations, *e.g.*, if replica permutational symmetry or SUSY are broken.) There are also ways to access the cumulants of the random free-energy functionals by introducing replicas or copies of the original system which are coupled to different applied sources (replica symmetry is then explicitly broken). This can be combined with the Boltzmann-Gibbs distribution at equilibrium in a replica field theory,[24,25] a minimization equation at zero temperature in a superfield theory,[18,19,26] and a Langevin equation describing the dynamics in a dynamical field theory.[47,52] In the following, we discuss the replica field formalism (which we abbreviate as "rep-f" below), the Parisi-Sourlas SUSY formalism (abbreviated simply as "SUSY") and the replica superfield theory (in the limit which was called "Grassmannian ultralocality" in [18,19,26]) which we denote by "rep-sf".

For the bare $\phi^4$ RFIM theory, the corresponding actions read

$$S_{\mathrm{rep-f}} = \sum_a \int_x \Big[\frac{1}{2}(\partial\phi_a)^2 + \frac{1}{2}r\phi_a^2 + \frac{1}{4!}g\phi_a^4\Big] \\ - \frac{1}{2}\Delta\sum_{ab}\int_x \phi_a\phi_b, \tag{C1}$$

where $\phi_a$ is a (replica) field depending on the space co-ordinate $x$, and

$$S_{\mathrm{SUSY}} = \int_{x\theta\bar\theta}\Big[-\frac{1}{2}\Phi\Delta_{\mathrm{SUSY}}\Phi + \frac{1}{2}r\phi^2 + \frac{1}{4!}g\Phi^4\Big], \tag{C2}$$

$$S_{\mathrm{rep-sf}} = \sum_a \int_{x\theta_a\bar\theta_a}\Big[\frac{1}{2}(\partial_\mu\Phi_a)^2 + \frac{1}{2}r\Phi_a^2 + \frac{1}{4!}g\Phi_a^4\Big] \\ - \frac{\Delta}{2}\sum_{ab}\int_{x\theta_a\bar\theta_a\theta_b\bar\theta_b}\Phi_a\Phi_b, \tag{C3}$$

where $\Phi$ is a superfield that depends on $x$ and two Grassmann coordinates $\theta$ and $\bar\theta$ and $\Phi_a$ is a (replica) superfield that depends on $x$ and two Grassmann coordinates $\theta_a$ and $\bar\theta_a$.

The superfields can be expanded in their Grassmann coordinates,

$$\Phi = \phi(x) + \bar\theta\psi(x) + \bar\psi(x)\theta + \bar\theta\theta\hat\phi(x) \\ \Phi_a = \phi_a(x) + \bar\theta_a\psi_a(x) + \bar\psi_a(x)\theta_a + \bar\theta_a\theta_a\hat\phi_a(x). \tag{C4}$$

Finally, the supersymmetric Laplacian $\Delta_{\mathrm{SUSY}} = \partial_\mu^2 + \Delta\partial_\theta\partial_{\bar\theta}$ involves the standard Euclidean Laplacian and derivatives with respect to the Grassmann coordinates.

Starting from the replica field approach, Cardy found a linear transformation of the fields that allows a diagonalization of the quadratic part in the limit $n \to 0$. The new fields introduced by Cardy are

$$\hat\phi = \frac{1}{2}\Big(\phi_1 + \frac{1}{1-n}\sum_{a=2}^n \phi_a\Big), \\ \phi = \frac{1}{2}\Big(\phi_1 - \frac{1}{1-n}\sum_{a=2}^n \phi_a\Big), \tag{C5}$$

as well as $(n-2)$ fields $\chi_i$, $i = 3, \cdots, n$, which have no component along the $\phi_1$ field and which are orthogonal to $\phi$ and $\hat\phi$. In order to simplify the final expressions, it is often convenient to introduce an extra field $\chi_2$ (so that we now have $(n-1)$ $\chi_i$ field) which satisfies $\sum_{i=2}^n \chi_i = 0$. The sum over the $(n-1)$ indices $i$ is denoted by $\sum'_i = \sum_{i=2}^n$.

In this new set of variables, the relation with the Parisi-Sourlas approach is striking: The fields $\phi$ and $\hat\phi$ closely resemble the fields $\phi$ and $\hat\phi$ of the SUSY formalism and, in the limit $n \to 0$, a loop of the $(n-2)$ bosonic fields $\chi_i$ has the same contribution as a pair of Grassmann ghost fields $\psi$ and $\bar\psi$. We stress that Cardy's approach is just a rewriting of the replica formalism in the limit $n \to 0$, so that the outcome should be the same as with a conventional definition of the replica fields.

### 2. Dimensionless quantities

We now discuss the different ways of introducing dimensionless quantities. For simplicity, but without altering the main point to be made, we do not consider anomalous dimensions.

### a. Parisi-Sourlas SUSY

The coordinates are rescaled as

$$x = k^{-1}\tilde{x} \qquad \theta = k^{-1}\tilde{\theta} \qquad \bar{\theta} = k^{-1}\tilde{\bar{\theta}}, \qquad \text{(C6)}$$

so that the measure of integration and the Laplacian change to

$$\int_{x\theta\bar{\theta}} = k^{-d+2} \int_{\tilde{x}\tilde{\theta}\tilde{\bar{\theta}}} \qquad \text{(C7)}$$
$$\Delta_{\text{SUSY}} = k^2 \tilde{\Delta}_{\text{SUSY}}.$$

The scaling dimension of the field is then fixed by requiring that the kinetic term remains equal to unity, which imposes

$$\Phi = k^{\frac{d-4}{2}} \tilde{\Phi}. \qquad \text{(C8)}$$

This implies that the various fields appearing in the decomposition of Eq. (C4) transform as

$$\phi = k^{\frac{d-4}{2}}\tilde{\phi} \qquad \psi = k^{\frac{d-2}{2}}\tilde{\psi} \qquad \bar{\psi} = k^{\frac{d-2}{2}}\tilde{\bar{\psi}} \qquad \hat{\phi} = k^{\frac{d}{2}}\tilde{\hat{\phi}}. \qquad \text{(C9)}$$

The rescaling of the coupling constants therefore comes as

$$\tilde{r} = rk^{-2}, \qquad \tilde{g} = gk^{d-6}. \qquad \text{(C10)}$$

Note that the variance of the random field, $\Delta \equiv 1$, is unchanged under rescaling.

### b. Replica superfields

Not much needs to be said in this case. The rescaling of the coordinates and fields is the same as in the Parisi-Sourlas SUSY formalism. We observe that the variance of the random field comes with 2 integrals over Grassmann coordinates and 2 Grassmann derivatives in the SUSY formalism, while it appears with 4 Grassmann integrals in the replica superfield formalism. But, scalingwise, this is the same.

### c. Replica fields

In the replica field formalism, it was understood a long time ago that there exists a dangerously irrelevant variable. The most convenient way to account for this is to rewrite the action by multiplying the 1-replica part by one power of an inverse temperature $\beta$, the 2-replica part by two powers, etc.,

$$S_{\text{rep-f}} = \beta \sum_a \int_x \left[\frac{1}{2}(\partial\phi_a)^2 + \frac{1}{2}r\phi_a^2 + \frac{1}{4!}g\phi_a^4\right]$$
$$- \frac{\beta^2}{2}\Delta \sum_{ab} \int_x \phi_a\phi_b. \qquad \text{(C11)}$$

In the vicinity of the fixed point, the running dimensionless inverse temperature $\tilde{\beta}_k$ behaves as $k^{-2}$, *i.e.*, the inverse temperature has scaling dimension $D_\beta = 2$ (and temperature a dimension of $D_T = -2$). It is then necessary to rescale the field with powers of the (inverse) temperature. The fields transform as

$$\phi_a = k^{\frac{d-2}{2}}\sqrt{\tilde{\beta}_k}\,\tilde{\phi}_a, \qquad \text{(C12)}$$

which yields

$$\phi_a = k^{\frac{d-4}{2}}\tilde{\phi}_a. \qquad \text{(C13)}$$

This coincides with the scaling dimension of the fields $\phi$ in the SUSY and the replica superfield approaches. Comparing with the latter formalisms, the inverse temperature plays the role of two Grassmann integrations or two Grassmann derivatives.

### d. Cardy's parametrization

The quadratic part of the action now reads

$$\int_x 2\partial_\mu\hat{\varphi}\partial_\mu\varphi + \frac{1}{2}\sum_i'(\partial_\mu\chi_i)^2 - 2\Delta\hat{\varphi}^2 + r[2\hat{\varphi}\varphi + \frac{1}{2}\sum_i'\chi_i^2] \qquad \text{(C14)}$$

where we recall that $\sum_i' = \sum_{i=2}^n$. This form enables one to determine the scaling dimension of the fields as

$$\varphi = k^{\frac{d-4}{2}}\tilde{\varphi} \qquad \chi = k^{\frac{d-2}{2}}\tilde{\chi} \qquad \hat{\varphi} = k^{\frac{d}{2}}\tilde{\hat{\varphi}}. \qquad \text{(C15)}$$

These dimensions coincide with those found in the SUSY formalism. Similarly, one can rewrite the $\phi_a^4$ interaction in terms of Cardy's fields and one finds

$$\sum_a \phi_a^4 = 8\hat{\varphi}\varphi(\varphi^2 + \hat{\varphi}^2) + 6(\varphi - \hat{\varphi})^2\sum_i'\chi_i^2$$
$$- 4(\varphi - \hat{\varphi})\sum_i'\chi_i^3 + 2\sum_i'\chi_i^4. \qquad \text{(C16)}$$

Not all terms have the same dimension but consideration of the leading ones indicates that one must rescale the coupling constant according to $\tilde{g} = gk^{d-6}$.

### 3. Feldman's operators

### a. Microscopic realization

For any integer $p > 0$, we now consider a coupling to quenched disorder that, in the action, takes the form

$$S_{\text{dis}} = -\sum_{i=1}^{2p}\int_x \sigma_i(x)\phi^i(x) \qquad \text{(C17)}$$

where the $\sigma_i$'s are Gaussian-distributed quenched random variables with zero mean and variances given by

$$\overline{\sigma_i(x)\sigma_j(y)} = \Delta_{2p}(-1)^i C_{2p}^i \delta(x-y)\delta_{i+j,2p}. \quad \text{(C18)}$$

Repeating the construction of the replica action leads to terms in the 2-replica part of the form

$$S_{2p} = -\frac{1}{2}\Delta_{2p}\sum_{ab}\int_x (\phi_a - \phi_b)^{2p}, \quad \text{(C19)}$$

which are precisely the operators considered by Feldman:[50] see Eq. (2). (Note that so defined the variance matrix has a positive determinant but is not positive definite. In a proper treatment this should be corrected by higher-order terms.)

If we implement the Parisi-Sourlas construction in this case, we find that no terms are generated, except for $p = 1$ which corresponds to the random field (up to a coupling to a random temperature which is here of no interest to us). The terms with $p > 1$ can thus be associated with the SUSY null and SUSY nonwritable terms of [1]. In what follows, we focus on these terms. Before averaging over disorder, one has contributions proportional to the auxiliary fields that read

$$e^{\hat{\phi}\sum_{i=0}^{2p} \; i\sigma_i\phi^{i-1} + \sum_{i=1}^{2p} \; i(i-1)\bar{\psi}\psi\sigma_i\phi^{i-2}}. \quad \text{(C20)}$$

After averaging, one obtains for $p > 1$ terms proportional to $\hat{\phi}^2$,

$$\hat{\phi}^2\sum_{i=1}^{2p-1} i(2p-i)\phi^{i-1}\phi^{2p-i-1}(-1)^i C_{2p}^i$$

$$= \hat{\phi}^2\phi^{2p-2}2p(2p-1)\sum_{j=0}^{2p-2}(-1)^{j+1}C_{2p-2}^j \quad \text{(C21)}$$

$$= -\hat{\phi}^2\phi^{2p-2}2p(2p-1)(1-1)^{2p-2} = 0.$$

There is also a term in $\hat{\phi}\bar{\psi}\psi$ which vanishes for the same reason and a term in $(\bar{\psi}\psi)^2$ which is zero due to the anticommuting property.

In the replica superfield construction, we obtain

$$S_{2p} = -\frac{1}{2}\Delta_{2p}\int_{x\theta_a\bar{\theta}_a\theta_b\bar{\theta}_b}\sum_{ab}(\Phi_a - \Phi_b)^{2p}. \quad \text{(C22)}$$

Performing the Grassmann integrals then yields

$$S_{2p} = -p(2p-1)\Delta_{2p}(\phi_a - \phi_b)^{2p-4}\int_x\sum_{ab}\Big[-\hat{\phi}_a\hat{\phi}_b \times$$

$$(\phi_a - \phi_b)^2 + 2(p-1)\bar{\psi}_a\psi_a\hat{\phi}_b(\phi_a - \phi_b) - 2(p-1)\bar{\psi}_b\psi_b\hat{\phi}_a$$

$$\times (\phi_a - \phi_b) + 2(p-1)(2p-3)\bar{\psi}_a\psi_a\bar{\psi}_b\psi_b\Big]. \quad \text{(C23)}$$

We observe that if we put all replica fields $\phi_a$ equal and all replica fields $\psi_a$ equal, the expression in Eq. (C23) vanishes. This is why the Parisi-Sourlas SUSY construction cannot describe such contributions.

Finally, in terms of Cardy's fields, one finds contributions that start with the $2p$th powers of $\chi$. In particular,

$$S_4 = -\frac{\Delta_4}{2}\int_x\Big[6(\sum_i{}'\chi_i^2)^2 - 16\hat{\varphi}\sum_i{}'\chi_i^3 + 48\hat{\varphi}^2\sum_i{}'\chi_i^2$$

$$-32\hat{\varphi}^4\Big] \quad \text{(C24)}$$

and

$$S_6 = -\frac{\Delta_6}{2}\int_x\Big[30(\sum_i{}'\chi_i^4)(\sum_i{}'\chi_i^2) - 20(\sum_i{}'\chi_i^3)^2$$

$$-24\hat{\varphi}\sum_i{}'\chi_i^5 + 120\hat{\varphi}^2\sum_i{}'\chi_i^4 - 320\hat{\varphi}^3\sum_i{}'\chi_i^3$$

$$+480\hat{\varphi}^4\sum_i{}'\chi_i^2 - 128\hat{\varphi}^6\Big]. \quad \text{(C25)}$$

### b. Dimensionless coupling constants $\tilde{\Delta}_p$

In the replica field formalism, $\Delta_{2p}$ appears at the 2-replica level, i.e., the second cumulant, which comes with a factor $\beta^2$. We deduce that

$$\tilde{\Delta}_{2p} = \frac{\Delta_{2p}}{\tilde{\beta}_k^2}k^{-d}\Big[k^{\frac{d-4}{2}}\Big]^{2p} \quad \text{(C26)}$$

$$= \Delta_{2p}k^{(d-4)(p-1)}. \quad \text{(C27)}$$

In the replica superfield formalism, we define the rescaled variable as

$$\tilde{\Delta}_{2p} = \Delta_{2p}k^{4-d}k^{p(d-4)} \quad \text{(C28)}$$

$$= \Delta_{2p}k^{(d-4)(p-1)}, \quad \text{(C29)}$$

which is compatible with the previous result.

Note that in the two above methods, all terms multiplying the coupling constant $\Delta_{2p}$ have the same dimension. This is no longer true in Cardy's field parametrization. Keeping only the leading term in $p > 1$, we conclude that

$$\tilde{\Delta}_{2p} = \Delta_{2p}k^{-d}k^{(d-2)p} \quad \text{(C30)}$$

$$= \Delta_{2p}k^{d(p-1)-2p}, \quad \text{(C31)}$$

which does not coincide with the previous result, except for $p = 2$. This is surprising because Cardy's formalism is just a rewriting of the replica action. As already pointed out, we observe that, in the replica superfield formalism, the terms with $p \geq 2$ vanish if we put all the replica fields equal, which is, in some sense what is done in Cardy's formalism.

#### 4.  3-replica operator

For later use, it is interesting to consider a generalization of Feldman's operators that involves 3 replicas,

$$\tilde{S}_6 = w_6 \sum_{abc} \int_x (\phi_a - \phi_b)^2 (\phi_b - \phi_c)^2 (\phi_c - \phi_a)^2, \quad (C32)$$

which can be rewriten in Cardy's formalism as

$$\tilde{S}_6 = -6w_6 \int_x \Big[ (\sum_i{}' \chi_i^2)^3 - 4\hat{\varphi}(\sum_i{}' \chi_i^3)(\sum_i{}' \chi_i^2)$$
$$+ 4\hat{\varphi}^2 \big[ \sum_i{}' \chi_i^4 + 3(\sum_i{}' \chi_i^2)^2 \big] - 16\hat{\varphi}^3 \sum_i{}' \chi_i^3 + 16\hat{\varphi}^4 \sum_i{}' \chi_i^2 \Big].$$
$$(C33)$$

When introducing dimensionless coupling constants in the replica field formalism, we must do it such that

$$\tilde{w}_6 = k^{2d-6} w_6. \quad (C34)$$

We then obtain exactly the same scaling in Cardy's approach. It is interesting to observe that, in the latter, $\Delta_6$ and $w_6$ have the same dimension because the leaders have both 6 powers of $\chi$ while, in the replica field approach, they differ by a power of $k^2$ because one is a 2-replica interaction and the other a 3-replica one.

#### 5.  Examples of Feynman diagrams

We now discuss a consistency check for determining scaling dimensions through the Feynman diagrams. To illustrate our argument, we look at Feynman diagrams that contribute to the renormalization of $\Delta_6$. Of course, $\Delta_6$ is an irrelevant coupling constant that does not need to be renormalized in order to make the theory finite. Our point is to show how different formalisms (replica field and Cardy) treat them.

In the replica field theory, one can build a diagram with three 4-point vertices of the 1-replica action (the term proportional to $g$). This diagram contributes to the renormalization of $\Delta_6$ if one uses two disconnected propagators and one connected one.

This diagram when evaluated in zero external momenta reads

$$g^3 \int_q \frac{1}{(q^2 + r)^5}. \quad (C35)$$

Note that the integral is both IR and UV finite. After introducing dimensionless variables, one finds that this diagram scales as $k^{3(6-d)}k^d k^{-10}$. The first contribution corresponds to the scaling of the 3 powers of $g$, the second to the scaling of the integration measure, and the last to the propagators. One therefore obtains a factor of $k^{2(4-d)}$ which coincides with the scaling found in the replica field approach for the operator coupling constant $\Delta_6$, see Eq. (C27).

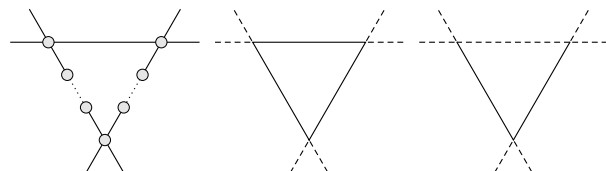

FIG. 8: One-loop Feynman diagrams with three 4-point vertices possibly contributing to the renormalization of $\Delta_6$ in different formalisms. Left: Our calculation with explicit replica symmetry breaking which scales as $k^{8-2d}$; the full line is a connected propagator and the two other edges involving a dashed line and two full lines correspond to disconnected propagators. Middle: The calculation of KRT with Cardy's fields [1]; the incoming dashed segments correspond to $\chi$ fields and a full line correspond to a $\langle \varphi\varphi \rangle$ propagator; this diagram scales as $k^{6-2d}$ but actually does not contribute to the renormalization of $\Delta_6$ but to that of $w_6$ which is a 3-replica and not a 2-replica quantity. Right: The proper diagram expressed with Cardy's fields that contributes to the renormalization of $\Delta_6$; it has two $\langle \varphi\varphi \rangle$ propagators and one $\langle \chi_i\chi_j \rangle$ propagator and scales as $k^{8-2d}$, as the diagram on the left. (Because $\langle \chi_i\chi_j \rangle$ is not purely diagonal, the diagram also contributes to the renormalization of $w_6$ but it is subdominant compared to the middle diagram.)

We now reproduce the calculation with Cardy's fields. In order to renormalize $\Delta_6$, one needs to draw a Feynman diagram with only $\chi$ external legs. The leading terms in the 1-replica 4-point interaction comes with two powers of $\chi$ and 2 powers of $\varphi$. We therefore expect that the relevant diagram has three $\langle \varphi\varphi \rangle$ propagators. In total, once dimensionless coupling constants are introduced, the diagram scales as $k^{3(6-d)}k^d k^{-12} = k^{6-2d}$, which corresponds to the scaling in Eq. (C31).

The previous argument, however, has a flaw because, in the Feynman diagram considered above, all external $\chi$ legs arising from the different 4-point interactions have independent indices. Stated otherwise, this particular diagram renormalizes something proportional to $(\sum_i{}' \chi_i^2)^3$, which corresponds to $w_6$ and not to $\Delta_6$. To renormalize the Feldman operator, one needs to connect the indices of the external $\chi$ legs. This is possible if one uses subleading terms in Eq. (C16), and more specifically the term in $\varphi \sum_i{}' \chi_i^3$, in two of the three interaction terms appearing in the diagram. For the third interaction term one can use the leading contribution, $\varphi^2 \sum_i{}' \chi_i^2$. The diagram now has one $\langle \chi_i\chi_j \rangle$ propagator, which contains a piece with $\delta_{ij}$ and therefore renormalizes $\Delta_6$, and two $\langle \varphi\varphi \rangle$ propagators. After introducing the dimensionless variables, the diagram is found to scale as $k^{3(6-d)}k^d k^{-10} = k^{4-2d}$, which now coincides with the scaling of Eq. (C27) in the replica field approach.

This is illustrated in Fig. 8.

The above development shows that care should be exerted when dealing with operators involving sums over replicas (and their associated coupling constants). The scaling obtained in the FRG, with the replica field or superfield approach and the introduction of a running

dimensionless temperature that flows to zero, for casting the functional flow equations in a dimensionless form appears fully consistent, either nonperturbatively or via an expansion in Feynman diagrams. Is it possible, on the other hand, that the use of Cardy's field parametrization with no explicit reference to a renormalized temperature might run into difficulties? The jury is still out, as there may be subtleties coming from accidental cancellations, the mixing of leaders and followers, and the possible difficulty to directly compare results obtained in the 1-PI formalism with those obtained at the level of the renormalized action.

## Appendix D: Disappearance of the SUSY/DR fixed point in the RFIM in the NP-FRG approximation DE2

We start from the fixed-point equation for $\delta_{*,2}$ in Eq. (22) which we reproduce below, making explicit the dependence on $\epsilon = d - d_{\mathrm{DR}}$. We assume that there is a $d_{\mathrm{DR}}$ in which $\Lambda_2 = 0$, so that

$$0 = A_*(\varphi;\epsilon)\delta_{*,2}(\varphi;\epsilon)^2 + L_*(\varphi,\partial_\varphi,\partial_\varphi^2;\epsilon)\delta_{*,2}(\varphi;\epsilon) + B_*(\varphi;\epsilon),$$
(D1)

where we recall that the functions $u_k''(\varphi)$, $z_k(\varphi)$, $\delta_{k,0}(\varphi)$, and the anomalous dimensions are fixed at their SUSY/DR fixed-point expressions.

We also consider the eigenvalue equation, Eq. (24), in $\epsilon = 0$:

$$\lambda_0 f_{\lambda_0}(\varphi) = \\ 2A_*(\varphi;0)\delta_{*,2}(\varphi;0)f_{\lambda_0}(\varphi) + L_*(\varphi,\partial_\varphi,\partial_\varphi^2;0)f_{\lambda_0}(\varphi).$$
(D2)

By construction, all eigenvalues are $> 0$ (irrelevant) except one, equal to $\Lambda_2$ that vanishes; the corresponding eigenfunction is then denoted $f_0(\varphi)$. Because $\delta_2(\varphi)$ is an even function we restrict ourselves to even eigenfunctions.

As mentioned in the main text, $A_*(\varphi;\epsilon)$, $B_*(\varphi;\epsilon)$, and the linear operator $L_*(\varphi,\partial_\varphi,\partial_\varphi^2;\epsilon)$ are regular function of $\epsilon$,

$$A_*(\varphi;\epsilon) = A_*(\varphi;0) + \epsilon \dot{A}_*(\varphi;0) + \mathrm{O}(\epsilon^2), \qquad \text{(D3)}$$

etc., where a dot denotes a derivative with respect to $\epsilon$. Note that $A_* \neq 0$.

We now expand the fixed-point function $\delta_{k,2}(\varphi;\epsilon)$ around $\delta_{k,2}(\varphi;0)$. We do so by using the basis formed by the eigenfunctions in $\epsilon = 0$, i.e.,

$$\delta_{*,2}(\varphi;\epsilon) = \delta_{*,2}(\varphi;0) + c_0(\epsilon)f_0(\varphi) + \sum_{\lambda_0>0} c_{\lambda_0}(\epsilon)f_{\lambda_0}(\varphi),$$
(D4)

where $c_0(\epsilon) \to 0$ and $c_{\lambda_0}(\epsilon) \to 0$ as $\epsilon \to 0$. It is expected that the coefficients associated with the irrelevant directions in $\epsilon = 0$ are regular at small $\epsilon$, $c_{\lambda_0}(\epsilon) = \epsilon \tilde{c}_{\lambda_0} + \cdots$ for $\lambda_0 > 0$, while the coefficient along the zero mode may behave in a singular way as $\epsilon \to 0$.

Inserting the above results and definitions into Eq. (D3) leads to

$$-\epsilon F(\varphi) = \epsilon \sum_{\lambda_0>0} \lambda_0 \tilde{c}_{\lambda_0} f_{\lambda_0}(\varphi) + A_*(\varphi;0)c_0(\epsilon)^2 f_0(\varphi)^2 \\ + \mathrm{O}(\epsilon^2, \epsilon c_0(\epsilon)).$$
(D5)

where the function $F(\varphi) = \dot{A}_*(\varphi;0)\delta_{*,2}(\varphi;0)^2 + \dot{L}_*(\varphi,\partial_\varphi,\partial_\varphi^2;0)\delta_{*,2}(\varphi;0) + \dot{B}_*(\varphi;0)$ is supposed to be known and $A_*(\varphi;0) \neq 0$. Generically, one expects the solution of the above equation for $\epsilon > 0$, i.e., $d < d_{\mathrm{DR}}$, to have $c_0(\epsilon)^2 \propto \epsilon$, which implies a square-root behavior. This behavior however cannot carry over to $\epsilon < 0$ and, $c_0$ being continuous in $\epsilon$, the solution therefore disappears when $d < d_{\mathrm{DR}}$.

The above argument is not rigorous but is indicative of what the generic behavior should be.

## Appendix E: NP-FRG flow equations at DE4 approximation level

We start from the exact FRG flow equations for the cumulants which are for instance given in Appendix C of [19] and we insert the DE4 truncation given in Eqs. (27-30). This provides a closed set of partial differential equations for the functions $U_k(\phi_1)$, $Z_k(\phi_1)$, $W_{a,b,c;k}(\phi_1)$, $V_k(\phi_1,\phi_2)$, $X_{a,b,c;k}(\phi_1,\phi_2)$, and $V_{3k}(\phi_1,\phi_2,\phi_3)$. It is actually more convenient to work with the cumulants of the renormalized random field, $\Delta_k(\phi_1,\phi_2) = V_k^{(11)}(\phi_1,\phi_2)$ and $S_k(\phi_1,\phi_2,\phi_3) = V_{3k}^{(111)}(\phi_1,\phi_2,\phi_3)$, which also lead to a closed set of flow equations.

The next step is to introduce scaling dimensions and dimensionless quantities, along the procedure described in and around Eqs. (15) and (16). The resulting dimensionless FRG flow equations can then be symbolically written as

$$\partial_t u_k'(\varphi) = -\frac{1}{2}(d - 2\eta_k + \bar{\eta}_k)u_k'(\varphi) \\ + \frac{1}{2}(d - 4 + \bar{\eta}_k)\varphi u_k''(\varphi) + \beta_{u'}(\varphi) + \mathrm{O}(\widetilde{T}_k),$$
(E1)

$$\partial_t z_k(\varphi) = \eta_k z_k(\varphi) \\ + \frac{1}{2}(d - 4 + \bar{\eta}_k)\varphi z_k'(\varphi) + \beta_z(\varphi) + \mathrm{O}(\widetilde{T}_k),$$
(E2)

$$\partial_t w_{a;k}(\varphi) = (2 + \eta_k)w_{a;k}(\varphi) \\ + \frac{1}{2}(d - 4 + \bar{\eta}_k)\varphi w_{a;k}'(\varphi) + \beta_{w_a}(\varphi) + \mathrm{O}(\widetilde{T}_k),$$
(E3)

$$\partial_t w_{b;k}(\varphi) = \frac{1}{2}(d + 2\eta_k + \bar{\eta}_k)w_{b;k}(\varphi) \\ + \frac{1}{2}(d - 4 + \bar{\eta}_k)\varphi w_{b;k}'(\varphi) + \beta_{w_b}(\varphi) + \mathrm{O}(\widetilde{T}_k),$$
(E4)

$$\partial_t w_{c;k}(\varphi) = (d - 2 + \eta_k + \bar{\eta}_k)w_{c;k}(\varphi)$$
$$+ \frac{1}{2}(d - 4 + \bar{\eta}_k)\varphi w'_{c;k}(\varphi) + \beta_{w_c}(\varphi) + \mathrm{O}(\widetilde{T}_k),$$
$$\text{(E5)}$$

$$\partial_t \delta_k(\varphi_1, \varphi_2) = (2\eta_k - \bar{\eta}_k)\delta_k(\varphi_1, \varphi_2) + \frac{1}{2}(d - 4 + \bar{\eta}_k)$$
$$\times (\varphi_1\partial_{\varphi_1} + \varphi_2\partial_{\varphi_2})\delta_k(\varphi_1, \varphi_2) + \beta_\delta(\varphi_1, \varphi_2) + \mathrm{O}(\widetilde{T}_k),$$
$$\text{(E6)}$$

$$\partial_t x_{a,b,c;k}(\varphi_1, \varphi_2) = (2 + 2\eta_k - \bar{\eta}_k)x_{a,b,c;k}(\varphi_1, \varphi_2)$$
$$+ \frac{1}{2}(d - 4 + \bar{\eta}_k)(\varphi_1\partial_{\varphi_1} + \varphi_2\partial_{\varphi_2})x_{a,b,c;k}(\varphi_1, \varphi_2) \quad \text{(E7)}$$
$$+ \beta_{x_{a,b,c}}(\varphi_1, \varphi_2) + \mathrm{O}(\widetilde{T}_k),$$

and

$$\partial_t s_k(\varphi_1, \varphi_2, \varphi_3) = \frac{1}{2}(d + 6\eta_k - 3\bar{\eta}_k)s_k(\varphi_1, \varphi_2, \varphi_3)$$
$$+ \frac{1}{2}(d - 4 + \bar{\eta}_k)(\varphi_1\partial_{\varphi_1} + \varphi_2\partial_{\varphi_2} + \varphi_3\partial_{\varphi_3})s_k(\varphi_1, \varphi_2, \varphi_3)$$
$$+ \beta_s(\varphi_1, \varphi_2, \varphi_3) + \mathrm{O}(\widetilde{T}_k),$$
$$\text{(E8)}$$

where, we recall, $t = \log(k/k_{\mathrm{UV}})$ and $k_{\mathrm{UV}}$ is a UV cutoff associated with the microscopic scale of the model. The beta functions themselves depend on $u'_k$, $z_k$, $w_{a,b,c;k}$, $\delta_k$, $x_{a,b,c;k}$, $s_k$ and their derivatives, and they depend as well on the (dimensionless) regulator functions that are introduced to implement the IR cutoff on the functional RG flows. In addition, the running anomalous dimensions $\eta_k$ and $\bar{\eta}_k$ are fixed by the conditions $z_k(0) = \delta_k(0,0) = 1$. The $\mathrm{O}(\widetilde{T}_k)$ terms are subdominant when one approaches the fixed point as $\widetilde{T}_k$ goes to zero as $k^\theta$. They are also exactly zero when the bare temperature $T$ is set to zero. The resulting zero-temperature beta functions are sufficient to study the fixed point and the spectrum of eigenvalues around it. The expressions for these beta functions, and if needed for the subdominant terms proportional to $\widetilde{T}_k$, are obtained via Mathematica. They are too long to be displayed in this appendix but a Mathematica notebook can be made available for anyone interested in using the equations.

Note also that the SUSY Ward identities for equal replica fields that are discussed in the main text are preserved by the above flow equations, provided (i) they are satisfied in the initial condition, (ii) one works at zero bare temperature, and (iii) the 6 functions whose flows are given in Eqs. (36-41) indeed reach a fixed point.

### Appendix F: Disappearance of the SUSY/DR fixed point in the RFIM at the DE4 NP-FRG approximation level

We group the 6 functions $\delta_{k,2}(\varphi)$, $x_{a;k,2}(\varphi)$, $x_{e;k,2}(\varphi)$, $x_{f;k,2}(\varphi)$, $s_{k,2}(\varphi)$, and $s_{k,3}(\varphi)$ in a vector $\mathbf{X}_k(\varphi)$ and we fix all the functions $u''_k(\varphi)$, $z_k(\varphi)$, $w_{a,b,c;k}(\varphi)$, and, via

SUSY [see Eq. (32)], $\delta_k(\varphi, \varphi)$, $x_{a;k}(\varphi, \varphi)$, $x_{b;k}^{(10)}(\varphi, \varphi) - x_{c;k}^{(10)}(\varphi, \varphi)$, $x_{b;k}^{(11)}(\varphi, \varphi) + x_{a;k}^{(11)}(\varphi, \varphi)$, and $s_k(\varphi, \varphi, \varphi)$, as well as the anomalous dimensions of the field, to their SUSY/DR fixed-point expressions. Then, making explicit the linear and nonlinear parts of Eqs. (36-41), one can rewrite the flow equations in the vicinity of the putative critical dimension $d_{\mathrm{DR}}$ as

$$\partial_t X_{\alpha,k}(\varphi; \epsilon) = C_\alpha(\varphi; \epsilon) + L_{\alpha\beta}(\varphi, \partial_\varphi, \partial_\varphi^2; \epsilon)X_{\beta,k}(\varphi; \epsilon) +$$
$$A_{\alpha\beta\gamma}(\varphi; \epsilon)X_{\beta,k}(\varphi; \epsilon)X_{\gamma,k}(\varphi; \epsilon) + B_{\alpha\beta\gamma\delta}(\varphi; \epsilon)X_{\beta,k}(\varphi; \epsilon)\times$$
$$X_{\gamma,k}(\varphi; \epsilon)X_{\delta,k}(\varphi; \epsilon),$$
$$\text{(F1)}$$

where $\alpha = 1, \cdots, 6$, summation over repeated indices is implied, $\epsilon = d - d_{\mathrm{DR}}$, $L_{\alpha\beta}(\varphi, \partial_\varphi, \partial_\varphi^2; \epsilon)$ is a linear operator, and $A_{\alpha\beta\gamma}(\varphi; \epsilon) > 0$. The cubic term actually only appears in the equation for $X_6 \equiv s_3$.

From Eq. (F1) one obtains the fixed-point equations by setting the left-hand side to zero and the eigenvalue equations by linearizing the equations for a small perturbation around the fixed point. When $\epsilon = 0$ and with $X_{\alpha,k}(\varphi; 0) = X_{\alpha,*}(\varphi; 0) + k^{\lambda_0}f_{\lambda_0,\alpha}(\varphi; 0)$, this leads to

$$\lambda f_{\lambda_0,\alpha}(\varphi) = L_{\alpha\beta}(\varphi, \partial_\varphi, \partial_\varphi^2; 0)f_{\lambda_0,\beta}(\varphi) + [A_{\alpha\beta\gamma}(\varphi; 0) +$$
$$A_{\alpha\gamma\beta}(\varphi; 0)]X_{\beta,*}(\varphi; 0)f_{\lambda_0,\beta}(\varphi) + [B_{\alpha\beta\gamma\delta}(\varphi; 0) + B_{\alpha\beta\delta\gamma}(\varphi; 0)$$
$$+ B_{\alpha\delta\gamma\beta}(\varphi; 0)]X_{\beta,*}(\varphi; 0)X_{\gamma,*}(\varphi; 0)f_{\lambda_0,\delta}(\varphi),$$
$$\text{(F2)}$$

where by definition of $d_{\mathrm{DR}}$ (*i.e.*, $\epsilon = 0$), one eigenvalue is zero and the others are strictly positive.

We work in the limit of vanishingly small $\epsilon$. As for the DE2 case (see Appendix D) we expand the difference between the fixed-point functions in $\epsilon$ and those in $\epsilon = 0$ in the basis formed by the eigenfunctions of the linearized equations in $\epsilon = 0$. We expect that $L_{\alpha\beta}$, $A_{\alpha\beta\gamma}$, $B_{\alpha\beta\gamma\delta}$, $C_\alpha$ are regular functions of $\epsilon$ around $\epsilon = 0$, with $L_{\alpha\beta} = L_{\alpha\beta}(\epsilon = 0) + \epsilon\dot{L}_{\alpha\beta}(\epsilon = 0) + \cdots$, and similarly for the other functions. We also expect that the coefficients of the expansion of $\mathbf{X}_*(\varphi; \epsilon) - \mathbf{X}_*(\varphi; 0)$ along the eigenfunctions with strictly positive (irrelevant) eigenvalues are also regular in $\epsilon$, *i.e.*,

$$X_{\alpha,*}(\varphi; \epsilon) - X_{\alpha,*}(\varphi; 0) =$$
$$\epsilon \sum_{\lambda_0 > 0} \tilde{c}_{\lambda_0,\alpha\beta}f_{\lambda_0,\beta}(\varphi) + c_{0,\alpha\beta}(\epsilon)f_{0,\beta}(\varphi), \quad \text{(F3)}$$

where the $c_{0,\alpha\beta}$'s go to 0 in a possibly singular way as $\epsilon \to 0$.

The equations for the fixed point then become, for $\alpha = 1, \cdots, 6$,

$$-\epsilon\big[F_\alpha(\varphi) - \sum_{\lambda_0 > 0} \lambda_0\tilde{c}_{\lambda_0,\alpha\beta}f_{\lambda_0,\beta}(\varphi)\big] =$$
$$A_{\alpha\beta\gamma}(\varphi; 0)c_{0,\beta\beta'}(\epsilon)c_{0,\gamma\gamma'}(\epsilon)f_{0,\beta'}(\varphi)f_{0,\gamma'}(\varphi)$$
$$\text{(F4)}$$

up to a $O(\epsilon c_0(\epsilon), c_0(\epsilon)^3)$, with

$$F_\alpha(\varphi) =$$
$$\dot{L}_{\alpha\beta}(\varphi, \partial_\varphi, \partial_\varphi^2; 0) X_{\beta,*}(\varphi; 0) + \dot{A}_{\alpha\beta\gamma}(\varphi; 0) X_{\beta,*}(\varphi; 0)$$
$$\times X_{\gamma,*}(\varphi; 0) + \dot{B}_{\alpha\beta\gamma\delta}(\varphi; 0) X_{\beta,*}(\varphi; 0) X_{\gamma,*}(\varphi; 0) X_{\delta,*}(\varphi; 0)$$
$$+ \dot{C}_\alpha(\varphi; 0)$$

(F5)

a known function. Generically, in the absence of accidental (or symmetry-induced) cancellation, one expects that the solution of the above set of equations which is supposed to exist for $\epsilon \geq 0$ entails that the $c_{0,\alpha\beta}(\epsilon)$'s behave as $\sqrt{\epsilon}$ when $\epsilon \to 0$. This however cannot apply when $\epsilon < 0$ and the solution should then generally disappear.

## Appendix G: Digressions on the peculiarities of a zero-temperature critical fixed point in the presence of disorder and on the unusual mechanism by which the SUSY/DR fixed point disappears

### 1. On nonanalytic operators at a zero-temperature fixed point with disorder

In their paper KRT[1] raise a criticism to the work in which we study the relevance of the avalanches at the critical point of the RFIM by a 2-loop perturbative FRG in $\epsilon = 6 - d$[39]. To do so we considered a functional *nonanalytic* perturbation around the SUSY (zero-temperature) fixed-point effective action and computed its dimension in perturbation. This perturbation is generated by the presence of scale-free avalanches and creates a cusp in the second 1-PI cumulant of the renormalized random field. Because we work with the effective action, *i.e.*, the 1-PI generating functional, what we actually computed is the eigenvalue $\Lambda_{3/2}$ associated with the perturbation around the fixed point. The odd feature pointed out by KRT is that, contrary to what is usually found for *analytic* fixed-point theories (see, *e.g.*, [48]), the two-point correlation function of the nonanalytic (fluctuating) operator $\mathcal{O}_{3/2}(x)$ *a priori* corresponding to the perturbation does not show the power-law spatial dependence at long distance with an exponent $2(d - \Lambda_{3/2})$, *i.e.*, $\langle \mathcal{O}_{3/2}(x) \mathcal{O}_{3/2}(y) \rangle_c \sim 1/|x - y|^{2(d-\Lambda_{3/2})}$, which one would anticipate.

As we have already stressed, the connection between eigenvalues around the fixed point, scaling dimensions, and large-distance spatial of correlation functions is rather subtle in the RFIM at criticality. In our approach, this is due to the zero-temperature nature of the fixed point. Difficulties already arise in the treatment of analytical (polynomial) operators, as discussed in Sec. IV A, but it appears even more striking when considering non-analytical perturbations of the SUSY/DR fixed point. More explicitly, in our perturbative FRG treatment in $d = 6 - \epsilon$, we considered a perturbation in the renormalized 1-PI second cumulant of the form

$$-\frac{w_k}{4} \int_x \sum_{a,b} \varphi_a(x) \varphi_b(x) |\varphi_a(x) - \varphi_b(x)|,$$

which is associated with a cusp in the second cumulant of the renormalized random field (obtained by deriving with respect to $\varphi_a$ and $\varphi_b$). We showed through a 2-loop calculation that $w_k$ is an irrelevant coupling constant going as $k^{\Lambda_{3/2}}$ when $k \to 0$ with $\Lambda_{3/2} = 1 - \epsilon/2 - 5\epsilon^2/36 + O(\epsilon^3)$ (see Sec. IV B and Fig. 5). However, the pair correlation function of the corresponding fluctuating operator, $\langle \sum_{a,b} |\varphi_a(x) - \varphi_b(x)|^3 \sum_{c,d} |\varphi_c(y) - \varphi_d(y)|^3 \rangle_c$, even after having taken care of the factors of $n$ coming from the number of replicas, does not seem to have a leading behavior at large separation in $1/|x - y|^{2(d-\Lambda_{3/2})}$ as is usually found from the relation between scaling dimension and eigenvalue.[48] This was illustrated by KRT on a toy model.[1] We have repeated the calculation around the Gaussian fixed point (at one loop so that there is no anomalous dimension of the field) by using a different treatment of the nonanalyticity at zero temperature that was introduced by Chauve *et al.* for the FRG of a disordered elastic manifold.[34] We have obtained the very same result as KRT.[1] The nonanalytical operator seems to be decomposed at long distance into powers of analytical operators with no sign of the anomalous eigenvalue $\Lambda_{3/2}$. The conditions for relating the eigenvalue around the fixed point with the power-law spatial decay as in the usual (heuristic) derivation of [48] are manifestly not satisfied in the presence of nonanalyticities in a zero-temperature fixed-point theory.

This observation points to the rather unique features associated with such nonanalyticities. The peculiar behavior already appears at the UV/Gaussian level: One cannot derive an effective action with a nonanalyticity such as that in the above expression of the 1-PI second cumulant by simply adding to the bare action a non-Gaussian term with a nonanalytic functional dependence and doing a perturbation expansion around the Gaussian theory. Any naive approach of this kind gives back the DR Gaussian theory. This points to a nontrivial connection between 1-PI quantities in the effective action and fluctuating operators in the action whenever nonanalyticities are present in the former.

These specificities have been discussed in great detail by Chauve *et al.* in the context of the FRG for a disordered elastic manifold.[34] This is clearly unusual compared to more standard cases such as the pure $\phi^4$ theory and comes from the fact that the long-distance physics of disordered systems such as the RFIM and random elastic manifolds is affected by the presence of singular collective phenomena in the form of avalanches.

## 2. Unusual mechanism of the change of fixed point at $d_{\mathrm{DR}}$

In his lectures on the RFIM[2] Rychkov raises additional concerns about our nonperturbative FRG approach.

One question is about the unusual mechanism by which the new cuspy fixed point emerges from the two merging DR/SUSY fixed points when $d < d_{\mathrm{DR}}$. For the lowest eigenvalue governing the stability of the fixed point, this leads to a square-root singularity in $\sqrt{|d_{\mathrm{DR}} - d|}$ associated with a discontinuity at $d_{\mathrm{DR}}$. This peculiar behavior is indeed unseen in other models. It results from the intrinsically functional nature of the RG description of the RFIM at criticality and from the fact that the limit $d \to d_{\mathrm{DR}}^{-}$ is nonuniform in the field dependence of the FRG functions associated with the second and higher-order cumulants of the renormalized disorder. The new fixed point actually appears through a boundary-layer mechanism in these functions. This is unconventional (as is the zero-temperature and functional nature of the RFIM fixed points[48,62,63]), but the phenomenon has been well characterized in our previous papers: see [10,27].

A related issue is the existence and the nature of the unstable SUSY/DR fixed point which we predict to exist above $d_{DR}$ and which then coalesces with the stable SUSY/DR fixed point in $d = d_{DR}$, leading to its disappearance. The main critical scaling behavior, which corresponds to the SUSY/DR sector for equal replica fields, is identical for the stable and the unstable fixed points. But the two fixed points differ when considering the functions that appear in the second and higher-order cumulants of the renormalized random field when the replica-field arguments are different, e.g., $\delta_{*,2}(\varphi)$ in Eq. (20). From the comparison with the pattern as a function of $N$ in the RFO($N$)M in $d = 4 + \epsilon$ (see Fig. 3 bottom), we expect that the unstable fixed point is unobservable because somehow "infinitely unstable" close to $d = 6$. This would explain why a previous numerical search for finding this fixed point directly in $d = 6$ was unsuccessful.[39] (Note also that the fixed point is not perturbatively reachable in $d = 6$.) It only becomes "accessibly unstable" when approaching $d_{\mathrm{DR}}$. It remains that this, admittedly physically odd, unstable fixed point is mathematically well defined as another solution of the fixed-point equations near $d_{\mathrm{DR}}$ when $\Lambda_2 \to 0$: see Fig. 6.

[*] Electronic address: tarjus@lptmc.jussieu.fr

[†] Electronic address: tissier@lptmc.jussieu.fr

[‡] Electronic address: balog@ifs.hr

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
