# Peer review of "On the breakdown of dimensional reduction and supersymmetry in random-field models"

_SciPost Physics_

## Round 2 · Referee Report · Anonymous (Referee 1) · 2025-1-25

Report

The critical point of random field Ising model (RFIM) is believed to have supersymmetry (SUSY) and the property of dimensional reduction (DR) thanks to a famous conjecture by Parisi and Sourlas. In the submitted paper the authors use the method of functional renormalization group (FRG) to discuss the failure of the SUSY/DR properties in various integer dimensions ($d=2,3,4$) below the upper critical dimension ($d=6$).

The paper revisits the FRG analysis on this topic, supported by new results/updates and compares it to the work of Kaviraj, Rychkov and Trevisani (KRT). The latter had predicted, in the framework of non-functional perturbative RG with replica fields, the presence of special SUSY-breaking (Feldman) operators that become relevant in low enough dimensions making the SUSY fixed point unstable. In the FRG framework the Feldman operators are argued to be analogous to the eigenvalues associated with the flow of derivatives of a function of fields $R(z)$, which describe the cumulants of the disorder interaction. The eigenvalue associated with the second derivative corresponds to the most dangerous Feldman operator in KRT analysis. However in FRG when this eigenvalue becomes marginal the SUSY fixed point ceases to exist due to nonlinearity of the flow equations, instead of being unstable as argued by KRT. There is a new “cuspy” fixed point that corresponds to the deformation by a nonanalytic component in the cumulant. The FRG mechanism can be demonstrated in two different examples: the RF O(N) model close to $d=4$ where SUSY/DR fails for $N<=18$, and the RFIM in where SUSY/DR fails for $d<=5$. The authors argue that the nonlinearity and cuspy features showing up in FRG are invisible in usual perturbative RG. To sum up, although the FRG method agrees with some of the results and predictions in KRT, there are still loose ends, since the two approaches do not agree on whether SUSY/DR fixed point is unstable or non-existent below a critical dimension (estimated around $d\approx 5.1$) and emergence of non-analytic operators in RG. These need to be addressed in the future.

I have a few questions that I request the authors to clarify: 

  1. The main analysis of section III and IV shows that the number $p$, for derivatives of cumulant, can be analytically continued to non-integer values, which becomes relevant for $p=3/2$. Why should such a deformation (corresponding to non-integer p) be expected in RG? In particular, can they be related to a local operator in usual Wilsonian way? Or do they correspond to nonlocal deformations? 

  2. Is there a selection rule that always lets us map a given FRG eigenvalue to a specific composite operator of replica fields? For instance, why should $\Lambda_p$ (say for $p=3$) correspond exactly to the $\mathcal{F}_6$ and not a derivative of $\mathcal{F}_4$. 

(This is also related to the apparent mismatch of scaling dimensions of Feldman operators with the corresponding FRG term, for $p \neq 2$.)

  3. In section III, the RF O(N) model is studied at $d=4-\epsilon$ dimension. From the conclusion it seems there is a stable SUSY fixed point for $N \ge 19$ for any $\epsilon$. Does this mean the RF O(N) models, for high enough $N$, have a SUSY/DR fixed point at $d=5$?

In addition, I think it would be useful if the following points are clarified further in the paper:

A. In section III A. It would be nice to have some equations showing the action of the model under consideration, the definition of $R(z)$ from it, and some illustrative examples of how derivatives of $R(z)$ are equivalent to deformation by some replica field operator.

B. In section IV, I found it hard to follow the logic of dimension counting in FRG close to free theory, that is based on powers of inverse temperatures associated with cumulants. It would be useful to clarify the general formula with some equation(s).

These point may have been explained in previous papers of the authors. But since the current paper has a broader take on the RG mechanism, there is a need for completeness especially for an interested reader who may not be fully familiar with the FRG developments.

The RFIM and related models have an interesting physics and its SUSY/DR properties is a fascinating topic. The intriguing connection between two quite different renormalization proposals for the model: the FRG and a perturbative Wilsonian approach, have been nicely discussed in the paper. This definitely clarifies a lot of points in the mysterious RG flow of the model. However, I request the author to address my above queries and requests. I would then recommend the paper for publication.

Recommendation

Ask for minor revision

  • validity: -
  • significance: -
  • originality: -
  • clarity: -
  • formatting: -
  • grammar: -

Author:  Gilles Tarjus  on 2025-05-14  [id 5480]

(in reply to Report 1 on 2025-01-25)

We thank the referee for his/her positive appreciation of our work and we answer his/her questions below:

  1. “The main analysis of section III and IV shows that the number p, for derivatives of cumulant, can be analytically continued to non-integer values, which becomes relevant for p=3/2. Why should such a deformation (corresponding to non-integer p) be expected in RG? In particular, can they be related to a local operator in usual Wilsonian way? Or do they correspond to nonlocal deformations?”

First, let us mention that, as stressed in the manuscript, the eigenvalue corresponding to p=3/2 is irrelevant in the whole region d \geq d_{DR} where dimensional reduction (DR) and SUSY hold and is not the reason why the SUSY/DR fixed point disappears.

The correspondence between eigenvalues around the fixed-point effective action and local or nonlocal operators in the Wilsonian picture is far from obvious and remains an open problem. Unusual difficulties can be tracked down to the zero-temperature nature of the FRG fixed point which entails nonanalytic behavior of the renormalized effective action (behavior which is physically related to the presence of scale-free avalanches at criticality in the RFIM, as well documented theoretically and in computer simulations). This is for instance discussed at length for the case of an elastic manifold in a random environment in Ref. [34] by Le Doussal et al. (in particular, their Appendix B). We have discussed some of it in Appendix G and in a disclaimer paragraph about the difference between the FRG effective-action framework and the operators in the Wilsonian picture in Section II-B. At this stage, and in spite of several fruitful discussions with Kaviraj, Trevisani and Rychkov, as well as others, we cannot offer a more direct equivalence.

  1. “Is there a selection rule that always lets us map a given FRG eigenvalue to a specific composite operator of replica fields? For instance, why should \Lambda_p (say for p=3) correspond exactly to the F6 and not a derivative of F4. (This is also related to the apparent mismatch of scaling dimensions of Feldman operators with the corresponding FRG term, for p≠2.)”

As we have just stated above, there is no straightforward mapping between the FRG eigenvalues computed around the fixed-point effective action and composite operators in the Wilsonian sense. The former are associated with eigenfunctions that depend on the average replica fields (as a result of the Legendre transform) whereas the latter are built with the fluctuating replica fields. Close to the upper critical dimension, the correspondence is more direct, and Feldman showed that F4, F6, and other similar operators, do not mix with other operators having the same dimension up to the second order in \epsilon^2 (Ref. [50]).

In our study of \Lambda_{p=3} associated with Feldman’s F6, we explicitly consider the Feynman diagrams associated with F6 and not with derivatives of F4. This is discussed in Appendix D.

  1. “In section III, the RFO(N) model is studied at d=4−\epsilon dimension. From the conclusion it seems there is a stable SUSY fixed point for N≥19 for any \epsilon. Does this mean the RFO(N) models, for high enough N, have a SUSY/DR fixed point at d=5?”

Yes indeed, for N large enough, there is a SUSY/DR fixed point in all dimensions d≥4. This is clearly illustrated in the N-d diagram in Fig. 4.

Finally, as suggested by the referee:

A. “ In section III A. It would be nice to have some equations showing the action of the model under consideration, the definition of R(z) from it, and some illustrative examples of how derivatives of R(z) are equivalent to deformation by some replica field operator.”

In Sec. III-A we have added equations (and text) introducing the action in the nonlinear sigma model for the RFO(N)M and the function R(z). We have also discussed the derivatives of R(z) in z=1 and the relation with Feldman’s operators. This is on pages 5 and 6:

"To recall, one starts from an action involving only the unit vectors (...)

(...) which also corresponds to the second (1-PI) cumulant of the effective action, in place of coupling constants."

B. “In section IV, I found it hard to follow the logic of dimension counting in FRG close to free theory, that is based on powers of inverse temperatures associated with cumulants. It would be useful to clarify the general formula with some equation(s).These point may have been explained in previous papers of the authors. But since the current paper has a broader take on the RG mechanism, there is a need for completeness especially for an interested reader who may not be fully familiar with the FRG developments.”

It was hard to put the necessary info directly in Sec. IV but we have added a whole new Appendix providing a recap of the FRG for the RFIM and addressing the question of cumulants and powers of temperature. We have referred to this new Appendix C in several places in Sec. IV. The new Appendix is on page 21:

"Appendix C: Brief recap of the FRG formalism for the RFIM

The starting point of the FRG formalism for the RFIM is an exact RG equation (...)

(...) hence in the number of free replica sums involved, and the power of the replica fields or replica field differences with dimension $(d-4)/2$."

---

## Round 2 · Referee Report · Anonymous (Referee 2) · 2025-3-13

Strengths

(i) important connection between the work of different authors
(ii) pedagogic
(iii) good and exhaustive review of existing literature
(iv) among top 20% for this journal

Weaknesses

minor, see report

Report

First of all, there is already one very pertinent report, which I explicitly endorse; I will not repeat the points made there.

To summarize: The story of a field theoretical (RG) treatment for the RF Ising model started with claims on dimensional reduction and supersymmetry. We know today that this does not hold in physically relevant dimensions. The field theoretic mechanism for that was identified by D. Fisher, followed by Feldman. They showed that in order to treat the disordered phase, the effective action, or more specifically the force-force correlator, acquires a cusp. This implies that a functional RG (FRG) is needed. Following Fisher and Feldman, this FRG was developed by Tarjus and Tissier (TT), now joined by Balog (BTT) who focus on RF Ising, and Le Doussal and Wiese (with a focus on disordered elastic manifolds). More recently, the problem was reanalized by Kaviraj, Rychkov and Trevisani (KRT), with a focus on what the upper critical dimension of the RF Ising model is. The latter authors do not use FRG, but (standard) perturbative RG. Another difference in their treatment is the use of Cardy-variables instead of bosonic and fermionic degrees of freedom, which contain more degrees of freedom, resulting in "leaders" and "followers" in their language.

The work under review by BTT carefully compares the approach and results by KRT, to the results of FRG (mostly by TT). This is important as a reader of the literature wonders what "leaders" and "followers" mean physically, and whether the two formalisms are consistent or contradictory.
These questions are well answered by BTT. They show that all operators present in BTT have a counterpart in the FRG. They also give a careful analysis for the estimate of the upper critical dimension in the two approaches, showing that any discrepancies are due to different extrapolations away from d=6. As a result of their refined analysis, using different NPRG schemes, they now assert that the upper critical dimension below which dimensional reduction is broken is d_DR=5.1\pm 0.1. Considering the O(N) generalization of RF Ising allows the authors to complete the picture.

I have some questions, and would be glad if the authors could clarify them:
(i) In his original work, Dima Feldman estimated the critical exponents of the 3D Ising model from the non-linear sigma model. Is this a decent approximation (despite the evident weakness of the theoretical underpinning)?
(ii) When one uses the standard susy approach a la Parisi-Sourlas, as compared to the "leaders" and "followers" approach of KRT, what is one missing, if any?
(iii) there is a physical argument by Wiese (see the cited review, section 9.2 on RF magnets) that d_DR>=5. Any comment?

Requested changes

(i) please answer my questions above, as far as possible.
(ii) Ref. [57] is a comment in PRL. Please cite (and if appropriate) discuss the reply.

Recommendation

Publish (easily meets expectations and criteria for this Journal; among top 50%)

  • validity: high
  • significance: high
  • originality: good
  • clarity: high
  • formatting: excellent
  • grammar: excellent

Author:  Gilles Tarjus  on 2025-05-14  [id 5481]

(in reply to Report 2 on 2025-03-13)

We thank the referee for his/her positive appreciation of our work and we answer his/her questions below:

(i) "In his original work, Dima Feldman estimated the critical exponents of the 3D Ising model from the non-linear sigma model. Is this a decent approximation (despite the evident weakness of the theoretical underpinning)?"

As far as we know, and after having studied his papers again, there is no estimate of the exponents of the 3D RFIM from the nonlinear sigma model in Feldman’s work. He gives the values of the exponents of the RFO(N)M for N= 3, 4 and 5 at order \epsilon in d=4+\epsilon, but no estimate or extrapolation to N=1 in D=3.

(ii) "When one uses the standard susy approach a la Parisi-Sourlas, as compared to the "leaders" and "followers" approach of KRT, what is one missing, if any?"

When one uses the standard SUSY approach of Parisi and Sourlas, only operators evaluated for the same replica field, i.e., for a single replica, are accounted for. One then misses operators involving several different replicas which appear in the second and higher cumulants of the renormalized random field. These operators in Cardy’s parametrization correspond to SUSY-null and non-SUSY-writable operators in KRT. In the functional RG they correspond to expansions of the cumulants in differences in replica fields. This is discussed in Sections II-A and B and Section IV-A. Leaders and followers just refer to the fact that when considering a composite operator (here formed with replica fields) it is usually not an eigen-operator of scale transformation and can further be decomposed into terms of increasing scaling dimensions. This feature is not specific to the problem at hand.

(iii) " There is a physical argument by Wiese (see the cited review, section 9.2 on RF magnets) that d_DR>=5. Any comment?"

We did not find the physical argument mentioned by the referee in Wiese’s review. We directly contacted K. J. Wiese who gave us the following argument orally: In d<5 anti-periodic boundary conditions produce an interface in the RFIM which, as proven by a series of work on the random manifold model, is described by a nonanalytic fixed-point theory and shows scale-free avalanches, all features that cannot be described by a simple SUSY/DR analytic theory.
In the manuscript we have already discussed a similar argument that was put forward in earlier papers by Tissier-Tarjus: Scale-free avalanches are present at the mean-field level and in the whole region above d_{DR} \approx 5.1 where SUSY/DR holds; this implies that a cuspy component appears in the renormalized 1-PI cumulant of the random-field but this component is subdominant as one approaches the fixed point and corresponds to an irrelevant eigenvalue (denoted by \Lambda_{3/2} in the manuscript). The theory describing the RFIM in d>d_{DR} is therefore more complicated than the pure \phi4 theory in two dimensions less. The latter describes the first cumulant and the 1-PI cumulants of the renormalized random field evaluated for equal replica fields, which is sufficient to determine the main critical scaling exponents. However, it does not describe the cumulants for distinct replica fields, which, despite the fact that they do not influence the SUSY/DR sector at the fixed point, are necessary to account for the characteristics of the scale-free avalanches for instance.

This is discussed in Secs. II-B and IV-B, but we have also added a footnote stressing the above in the second paragraph of the right column on page 4 (footnote in now Ref. [53]):

Footnote in [53]:
"It should be stressed that, even in the domain of dimensions where SUSY and DR apply, the RFIM fixed point and its vicinity contain more than just the fixed point of the pure $\phi^4$ theory in 2 dimensions less. Indeed, the latter corresponds to a sector of the full RFIM fixed-point effective action in which the 1-PI cumulants are restricted to equal field arguments (which can also be thought of as associated with a single replica). Cumulants of order 2 and higher when considered for nonequal field arguments should also reach a fixed point, and the approach to this fixed point then involve eigenperturbations that do not appear in the spectrum of the pure $\phi^4$ theory in 2 dimensions less. In particular, scale-free avalanches are present in the RFIM at criticality for $d\geq d_{\rm DR}$, a feature that does not alter the SUSY/DR sector but is not described by it. The fixed point which disappears in $d<d_{\rm DR}$ is the full RFIM fixed point and this is due to the sector associated with nonequal field arguments in the cumulants; the fixed point of the pure $\phi^4$ theory in 2 dimensions less of course continues to exist."

Note that the issue is also stressed in Section III-A and Appendix A about the RFO(N)M. We indeed show in the RFO(N)M near d=4 that weak nonanalyticities in the form of subcusps in the 1-PI cumulants may be present in the region where the fixed point and the main scaling behavior is SUSY/DR. These subcusps are clearly not described by the SUSY/DR theory alone and require considering an extended theory that fully accounts for the disorder cumulants. However, these additional nontrivial features do not affect the SUSY/DR sector when N is large enough.

(iv) "Ref. [57] is a comment in PRL. Please cite (and if appropriate) discuss the reply."

We have cited on page 7 the reply to the comment to PRL in what is now Ref. [59].

---

## Round 3 · Referee Report · Anonymous (Referee 1) · 2025-5-31

Report

I thank the authors for addressing all my queries and requests, and making the appropriate changes. The paper has now significantly improved and is easier to follow. This is an important work towards understanding renormalization in random field systems. I am happy to recommend it for publication.

Requested changes

None

Recommendation

Publish (easily meets expectations and criteria for this Journal; among top 50%)

---

## Round 3 · Author Response

Dear Editor,

we thank both referees for their positive reviews and their comments. We have addressed them in the response and in the new version (posted on arXiv).

We hope that our revised manuscript can now be published in SciPost Physics.

Sincerely yours, The authors

Detailed response to the referees.

Referee #1

We thank the referee for his/her positive appreciation of our work and we answer his/her questions below:

  1. “The main analysis of section III and IV shows that the number p, for derivatives of cumulant, can be analytically continued to non-integer values, which becomes relevant for p=3/2. Why should such a deformation (corresponding to non-integer p) be expected in RG? In particular, can they be related to a local operator in usual Wilsonian way? Or do they correspond to nonlocal deformations?”

First, let us mention that, as stressed in the manuscript, the eigenvalue corresponding to p=3/2 is irrelevant in the whole region d \geq d_{DR} where dimensional reduction (DR) and SUSY hold and is not the reason why the SUSY/DR fixed point disappears.

The correspondence between eigenvalues around the fixed-point effective action and local or nonlocal operators in the Wilsonian picture is far from obvious and remains an open problem. Unusual difficulties can be tracked down to the zero-temperature nature of the FRG fixed point which entails nonanalytic behavior of the renormalized effective action (behavior which is physically related to the presence of scale-free avalanches at criticality in the RFIM, as well documented theoretically and in computer simulations). This is for instance discussed at length for the case of an elastic manifold in a random environment in Ref. [34] by Le Doussal et al. (in particular, their Appendix B). We have discussed some of it in Appendix G and in a disclaimer paragraph about the difference between the FRG effective-action framework and the operators in the Wilsonian picture in Section II-B. At this stage, and in spite of several fruitful discussions with Kaviraj, Trevisani and Rychkov, as well as others, we cannot offer a more direct equivalence.

  1. “Is there a selection rule that always lets us map a given FRG eigenvalue to a specific composite operator of replica fields? For instance, why should \Lambda_p (say for p=3) correspond exactly to the F6 and not a derivative of F4. (This is also related to the apparent mismatch of scaling dimensions of Feldman operators with the corresponding FRG term, for p≠2.)”

As we have just stated above, there is no straightforward mapping between the FRG eigenvalues computed around the fixed-point effective action and composite operators in the Wilsonian sense. The former are associated with eigenfunctions that depend on the average replica fields (as a result of the Legendre transform) whereas the latter are built with the fluctuating replica fields. Close to the upper critical dimension, the correspondence is more direct, and Feldman showed that F4, F6, and other similar operators, do not mix with other operators having the same dimension up to the second order in \epsilon^2 (Ref. [50]).

In our study of \Lambda_{p=3} associated with Feldman’s F6, we explicitly consider the Feynman diagrams associated with F6 and not with derivatives of F4. This is discussed in Appendix D.

  1. “In section III, the RFO(N) model is studied at d=4−\epsilon dimension. From the conclusion it seems there is a stable SUSY fixed point for N≥19 for any \epsilon. Does this mean the RFO(N) models, for high enough N, have a SUSY/DR fixed point at d=5?”

Yes indeed, for N large enough, there is a SUSY/DR fixed point in all dimensions d≥4. This is clearly illustrated in the N-d diagram in Fig. 4.

Finally, as suggested by the referee:

A. “ In section III A. It would be nice to have some equations showing the action of the model under consideration, the definition of R(z) from it, and some illustrative examples of how derivatives of R(z) are equivalent to deformation by some replica field operator.”

In Sec. III-A we have added equations (and text) introducing the action in the nonlinear sigma model for the RFO(N)M and the function R(z). We have also discussed the derivatives of R(z) in z=1 and the relation with Feldman’s operators. This is in a new text on pages 5 and 6 that is detailed below.

B. “In section IV, I found it hard to follow the logic of dimension counting in FRG close to free theory, that is based on powers of inverse temperatures associated with cumulants. It would be useful to clarify the general formula with some equation(s).These point may have been explained in previous papers of the authors. But since the current paper has a broader take on the RG mechanism, there is a need for completeness especially for an interested reader who may not be fully familiar with the FRG developments.”

It was hard to put the necessary info directly in Sec. IV but we have added a whole new Appendix providing a recap of the FRG for the RFIM and addressing the question of cumulants and powers of temperature. We have referred to this new Appendix C in several places in Sec. IV. The new Appendix C is reproduced below.

Referee #2

We also thank the referee for his/her positive appreciation of our work and we answer his/her questions below:

(i) As far as we know, and after having studied his papers again, there is no estimate of the exponents of the 3D RFIM from the nonlinear sigma model in Feldman’s work. He gives the values of the exponents of the RFO(N)M for N= 3, 4 and 5 at order \epsilon in d=4+\epsilon, but no estimate or extrapolation to N=1 in D=3.

(ii) When one uses the standard SUSY approach of Parisi and Sourlas, only operators evaluated for the same replica field, i.e., for a single replica, are accounted for. One then misses operators involving several different replicas which appear in the second and higher cumulants of the renormalized random field. These operators in Cardy’s parametrization correspond to SUSY-null and non-SUSY-writable operators in KRT. In the functional RG they correspond to expansions of the cumulants in differences in replica fields. This is discussed in Sections II-A and B and Section IV-A. Leaders and followers just refer to the fact that when considering a composite operator (here formed with replica fields) it is usually not an eigen-operator of scale transformation and can further be decomposed into terms of increasing scaling dimensions. This feature is not specific to the problem at hand.

(iii) We did not find the physical argument mentioned by the referee in Wiese’s review. We directly contacted K. J. Wiese who gave us the following argument orally: In d<5 anti-periodic boundary conditions produce an interface in the RFIM which, as proven by a series of work on the random manifold model, is described by a nonanalytic fixed-point theory and shows scale-free avalanches, all features that cannot be described by a simple SUSY/DR analytic theory. In the manuscript we have already discussed a similar argument that was put forward in earlier papers by Tissier-Tarjus: Scale-free avalanches are present at the mean-field level and in the whole region above d_{DR} \approx 5.1 where SUSY/DR holds; this implies that a cuspy component appears in the renormalized 1-PI cumulant of the random-field but this component is subdominant as one approaches the fixed point and corresponds to an irrelevant eigenvalue (denoted by \Lambda_{3/2} in the manuscript). The theory describing the RFIM in d>d_{DR} is therefore more complicated than the pure \phi4 theory in two dimensions less. The latter describes the first cumulant and the 1-PI cumulants of the renormalized random field evaluated for equal replica fields, which is sufficient to determine the main critical scaling exponents. However, it does not describe the cumulants for distinct replica fields, which, despite the fact that they do not influence the SUSY/DR sector at the fixed point, are necessary to account for the characteristics of the scale-free avalanches for instance.

This is discussed in Secs. II-B and IV-B, but we have also added a footnote stressing the above in the second paragraph of the right column on page 4. The footnote in now Ref. [53] and is reproduced below.

Note that the issue is also stressed in Section III-A and Appendix A about the RFO(N)M. We indeed show in the RFO(N)M near d=4 that weak nonanalyticities in the form of subcusps in the 1-PI cumulants may be present in the region where the fixed point and the main scaling behavior is SUSY/DR. These subcusps are clearly not described by the SUSY/DR theory alone and require considering an extended theory that fully accounts for the disorder cumulants. However, these additional nontrivial features do not affect the SUSY/DR sector when N is large enough.

(iv) We have cited on page 7 the reply to the comment to PRL in what is now Ref. [59].

---

## Round 3 · List of Changes

In response to Referee's 1 comments (see above):

- New text and equations on pages 5 and 6:

To recall, one starts from an action involving only the unit vectors $\mathbf{n}_a(x)$ describing the orientation of the $N$-component replica fields $\boldsymbol{\phi}_a(x)=\phi_0 \mathbf{n}_a(x)$ when their amplitude $\phi_0$ is frozen to a nonzero constant,\cite{fisher85,feldman02,tissier06,ledoussal06}
\begin{equation}
\begin{aligned}
S[\{\mathbf{n}_a\}]= &\int_x \bigg\{\frac{1}{2T}\sum_a \partial_\mu \mathbf{n}_a(x)\partial_\mu \mathbf{n}_a(x)- \\&
\frac 1{2T^2}\sum_{a,b} R(\mathbf{n}_a(x)\cdot\mathbf{n}_b(x)) +\cdots\bigg\}
\end{aligned}
\end{equation}
where the dependence on $\phi_0^2$ has been omitted (one can choose \phi_0^2\equiv 1$) and the ellipses denote irrelevant terms near $d=4$. The function $R(z)$ is the variance of the disorder. At the bare level it is simply equal to $R_B(z)=\Delta z$, which corresponds to the random-field disorder in Eq.~(\ref{eq_bare-action}) with the dependence on temperature made explicit. The scalar product of different replica fields, $z=\mathbf{n}_a\cdot\mathbf{n}_b$, is dimensionless in $d=4$, so that the whole function $R(z)$ is marginal. One can then proceed to a perturbative computation of the effective action in powers of the disorder variance and perform an RG treatment in $d=4+\epsilon$. Note that this perturbative RG is {\it functional}: it deals with a function of the fields, the renormalized variance of the disorder $R(z)$ which also corresponds to the second (1-PI) cumulant of the effective action, in place of coupling constants.

- New Appendix C providing a recap of the FRG for the RFIM and addressing the question of cumulants and powers of temperature and references to it on pages 9-11:

Appendix C: Brief recap of the FRG formalism for the RFIM}

The starting point of the FRG formalism for the RFIM is an exact RG equation for the scale-dependent effective action (or Gibbs free-energy functional) $\Gamma_k[\{\phi_a\}]$ which generates the 1-PI correlation functions when fluctuations are incorporated from the microscopic (UV) scale down to an IR cutoff $k$. This equation reads\cite{wetterich93},
\begin{equation}
\begin{aligned}
\partial_k\Gamma_k\left[\{ \phi_a\}\right ]= \frac{1}{2}\sum_{a,b} \int_q \partial_k R_{k,ab}(q^2) \big (\big[ \bm \Gamma _k^{(2)}+ \bm R_k\big]^{-1}\big)_{q,- q}^{ab},
\end{aligned}
\end{equation}
where $\bm \Gamma_k^{(2)}$ is the matrix formed by the second functional derivatives of $\Gamma_k$ with respect to the replica fields and the operator $\bm P_k[\{ \phi_a\}] \equiv [ \bm\Gamma _k^{(2)}+ \bm R_k]^{-1}$ is the exact propagator at the scale $k$. In the case of the RFIM,\cite{tissier11,tissier12a,tissier12b} the IR regulator $R_{k,ab}(q^2)$ is taken as $R_{k,ab}(q^2)=\widehat{R}_k(q^2)\delta_{ab} +\widetilde{R}_k(q^2)$. The functions $\widehat{R}_k(q^2)$ and $\widetilde{R}_k(q^2)$ are chosen to provide an IR cutoff on the fluctuations, which enforces a decoupling of the low- and high-momentum modes at the scale $k$. The function $\widehat{R}_k(q^2)$ adds a mass $\sim k^{2-\eta}$ to replica-field modes with $q^2<k^2$ and decays rapidly to zero for $q^2>k^2$, whereas the function $\widetilde{R}_k(q^2)$ (which must be chosen proportional to
$\partial_{q^2}\widehat{R}_k(q^2)$ to avoid an explicit SUSY breaking \cite{tissier11,tissier12a,tissier12b}) reduces the fluctuations of the bare random field.

The 1-PI cumulants $\Gamma_{kp}[\phi_{a1},\cdots,\phi_{ap}]$ are generated by the expansion of $\Gamma_k[\{\phi_a\}]$ in increasing number of free replica sums which is given in Eq.~(\ref{eq_expansion_cumulants}). When inserted in Eq.~(\ref{eq_erge}) this provides an infinite hierarchy of exact functional RG equations for the 1-PI cumulants, whose first equations are for instance given in Appendix C of [\onlinecite{tissier12b}] and not reproduced here.

To study scale invariance at criticality and describe the associated fixed point one needs to introduce scaling dimensions and dimensionless quantities. As stressed in several places in this manuscript, the RFIM fixed point is at zero dimensionless renormalized
temperature, {\it i.e.}, the latter flows to zero as one approaches the fixed point:
$\widetilde T_k\sim k^\theta T$ with $\theta>0$ as $k\to 0$ and $T$ the bare temperature. Temperature is thus irrelevant, albeit dangerously so, because one cannot simply set it to zero in the effective action. Each 1-PI cumulant indeed scales differently with temperature,
\begin{equation}
\Gamma_{kp}[\phi_{a1},\cdots,\phi_{ap}]=\frac 1{\widetilde T_k^p} \bigg[\gamma_{kp}[\varphi_{a1},\cdots,\varphi_{ap}] +
{\rm O}(\widetilde T_k)\bigg ] ,
\end{equation}
where the $\gamma_{kp}$'s have a finite limit when $\widetilde T_k \to 0$. In the above expression the replica fields are rescaled as
$\varphi_a(\tilde x)=k^{-(d-2+\eta)/2}\widetilde T_k^{1/2}\phi_a(x)$ $\forall a$, with $\tilde x=k x$ [see Eq.~(\ref{eq_dim1})]. One can check that this leads to a consistent hierarchy of dimensionless exact FRG equations for the 1-PI cumulants
$\gamma_{kp}[\varphi_{a1},\cdots,\varphi_{ap}]$ that allows one to describe the zero-temperature fixed point of the critical RFIM.

As discussed in Sec.~\ref{sub_dangerous}, near the upper critical dimension $d=6$, the canonical dimensions of the coupling constants and the associated operators are obtained by first expanding in number of free replica sums [Eq.~(\ref{eq_expansion_cumulants})] and then expanding the resulting cumulants in powers of the replica fields or of the difference between replica fields as in Feldman's operators. The canonical dimensions follow from the above, with the anomalous dimensions $\eta=\bar\eta=0$ and, as a result, $\theta=2$. One counts the power of the inverse temperature with dimension $\theta=2$, which depends on the cumulant that is considered [see Eq.~(\ref{eq_rescaled-cumulant})], hence in the number of free replica sums involved, and the power of the replica fields or replica field differences with dimension $(d-4)/2$.

In response to referee's 2 comments:

- Footnote in Ref. [53]:

It should be stressed that, even in the domain of dimensions where SUSY and DR apply, the RFIM fixed point and its vicinity contain more than just the fixed point of the pure $\phi^4$ theory in 2 dimensions less. Indeed, the latter corresponds to a sector of the full RFIM fixed-point effective action in which the 1-PI cumulants are restricted to equal field arguments (which can also be thought of as associated with a single replica). Cumulants of order 2 and higher when considered for nonequal field arguments should also reach a fixed point, and the approach to this fixed point then involve eigenperturbations that do not appear in the spectrum of the pure $\phi^4$ theory in 2 dimensions less. In particular, scale-free avalanches are present in the RFIM at criticality for $d\geq d_{\rm DR}$, a feature that does not alter the SUSY/DR sector but is not described by it. The fixed point which disappears in $d<d_{\rm DR}$ is the full RFIM fixed point and this is due to the sector associated with nonequal field arguments in the cumulants; the fixed point of the pure $\phi^4$ theory in 2 dimensions less of course continues to exist.

- Reply to Comment in PRL now cited as Ref. [59].

---

## Editorial Decision

accepted_in_target_journal